# Interaction of chikungunya virus glycoproteins with macrophage factors controls virion production

Zhenlan Yao [1], Sangeetha Ramachandran[1], Serina Huang [2], Erin Kim[3], Yasaman Jami-Alahmadi [4], Prashant Kaushal [1,5,6], Mehdi Bouhaddou [1,5,6], James A Wohlschlegel[4] & Melody MH Li [1,6 ✉]

## Abstract

**Despite their role as innate sentinels, macrophages can serve as cellular reservoirs of chikungunya virus (CHIKV), a highly-pathogenic arthropod-borne alphavirus that has caused large outbreaks among human populations. Here, with the use of viral chimeras and evolutionary selection analysis, we define CHIKV glycoproteins E1 and E2 as critical for virion production in THP-1 derived human macrophages. Through proteomic analysis and functional validation, we further identify signal peptidase complex subunit 3 (SPCS3) and eukaryotic translation initiation factor 3 subunit K (eIF3k) as E1-binding host proteins with anti-CHIKV activities. We find that E1 residue V220, which has undergone positive selection, is indispensable for CHIKV production in macrophages, as its mutation attenuates E1 interaction with the host restriction factors SPCS3 and eIF3k. Finally, we show that the antiviral activity of eIF3k is translation-independent, and that CHIKV infection promotes eIF3k translocation from the nucleus to the cytoplasm, where it associates with SPCS3. These functions of CHIKV glycoproteins late in the viral life cycle provide a new example of an intracellular evolutionary arms race with host restriction factors, as well as potential targets for therapeutic intervention.**

**Keywords** Alphavirus E1 Glycoprotein; Chikungunya Virus; eIF3k; Evolutionary Selection; Macrophage
**Subject Categories** Evolution & Ecology; Microbiology, Virology & Host Pathogen Interaction

## Introduction

Macrophages are phagocytic innate immune cells with critical functions in first-line defense against virus infection, inflammation, and priming of the adaptive immune system (Murray and Wynn, 2011). The sensing of viral infection by pattern recognition receptors in macrophages rapidly establishes an antiviral state through activation of the interferon (IFN) response (McNab et al, 2015). However, some viruses, such as highly-pathogenic avian influenza H5N1 viruses can breach this antiviral immunity (Cline et al, 2017; Marvin et al, 2017; Short et al 2012), highlighting productive macrophage infection as an important determinant for viral virulence. Moreover, in individuals infected with human immunodeficiency virus (HIV)(Kruize and Kootstra, 2019; Brown and Mattapallil, 2014), macrophages are potential reservoirs for rebound viremia upon cessation of antiretroviral therapy (Kruize and Kootstra, 2019; Kumar et al, 2014). Therefore, targeting viral infection of macrophages is an attractive therapeutic strategy for virus eradication.

Chikungunya virus (CHIKV) is a highly-pathogenic arthropod-borne alphavirus that has expanded worldwide with emerging lineages in recent decades (Weaver et al, 2020; Gould and Higgs, 2009). The unprecedented outbreaks from the Indian Ocean islands to Southeast Asia were caused by the novel CHIKV Indian Ocean lineage (IOL), characterized primarily by the E1-A226V mutation that adapted the virus from its principal vector *Aedes aegypti* to *Aedes albopictus* (Tsetsarkin et al, 2007, 2014; Chen et al, 2021). Although CHIKV infection is typically cleared in a few days, a significant percentage of individuals develop incapacitating arthralgia for up to 20 months (Schwartz and Albert, 2010; Pialoux et al, 2007; Gunn et al, 2012). Interestingly, CHIKV RNA and proteins persist in monocyte-derived macrophages (MDMs) in the spleen or synovial tissue for months in macaques and humans suffering from chronic arthralgia (Dupuis-Maguiraga et al, 2012; Labadie et al, 2010; Hoarau et al, 2010). These studies propose a role for macrophages as a cellular reservoir for CHIKV persistence and a niche for inflammation that is recurrently activated by viral components (Dupuis-Maguiraga et al, 2012; Kril et al, 2021). However, it is not clear what mechanism drives CHIKV persistence and whether this pathogenic role of macrophages is found in all arthritogenic alphavirus infections.

In contrast, o'nyong'nyong virus (ONNV), an arthritogenic alphavirus that shares the most genetic identity with CHIKV, is confined to periodic outbreaks in Africa (Weaver et al, 2020; Cottis et al, 2023). ONNV causes similar symptoms in humans, but is less virulent in mouse models, requiring a higher dose than

[1]Department of Microbiology, Immunology and Molecular Genetics, University of California, Los Angeles, Los Angeles, CA, USA. [2]Department of Human Genetics, David Geffen School of Medicine, University of California, Los Angeles, Los Angeles, CA, USA. [3]Department of Chemistry and Biochemistry, University of California, Los Angeles, CA, USA. [4]Department of Biological Chemistry, University of California, Los Angeles, Los Angeles, CA, USA. [5]Institute for Quantitative and Computational Biosciences, University of California, Los Angeles, Los Angeles, CA, USA. [6]Molecular Biology Institute, University of California Los Angeles, Los Angeles, CA, USA. ✉E-mail: Manhingli@mednet.ucla.edu

CHIKV to reach the same level of mortality (Seymour et al, 2013). The evolutionary similarities yet epidemiological differences make CHIKV-ONNV chimeras excellent molecular tools for probing viral determinants for host adaptation. However, these studies so far have mostly focused on their differential uses of mosquito vectors, such as transmission of ONNV by *Anopheles gambiae* (Saxton-Shaw et al, 2013; Vanlandingham et al, 2006), while little is known about the molecular mechanisms underlying infection of human cells relevant for viral dissemination, such as macrophages.

Viral infection is mostly abortive in macrophages as host restriction factors either basally expressed or amplified by the IFN response suppress specific viral life cycle stages (Tenthorey et al, 2022). Even though CHIKV replication is active in human MDMs, it is more restricted in MDMs than in epithelial cells and fibroblasts (Sourisseau et al, 2007), suggesting viral suppression by macrophage restriction factors. Host antiviral immunity can impose evolutionary selective pressures on viral proteins, propelling viruses to evade or antagonize these blockades, such as the arms race between myeloid-cell-specific SAMHD1 (SAM and HD domain containing deoxynucleoside triphosphate triphosphohydrolase 1) and HIV-2 Vpx (Hrecka et al, 2011; Laguette et al, 2012; Daugherty et al, 2014). This prompted us to question how and to what extent the evolutionary pressure brought on by the host-virus arms race has selected for increased CHIKV survival in human macrophages.

Here, we found that human primary monocyte and THP-1-derived macrophage infection with CHIKV (vaccine strain 181/clone 25) is much more efficient than that of ONNV at a step following genome replication. By utilizing a repertoire of CHIKV-ONNV chimeras, we mapped the viral determinant for efficient virion production in macrophages to the CHIKV E2 and E1 glycoproteins. Interestingly, evolutionary analysis of 397 CHIKV structural polyprotein sequences isolated from infected individuals uncovered signatures of positive selection mostly in E2 and E1 proteins. Mutating two of the positively selected residues in CHIKV to the homologous ones in ONNV (E2-V135L, E1-V220I) attenuates virion production in 293T and BHK-21 cells while the E1-V220I mutation completely abolishes virion production in macrophages. We further performed affinity purification-mass spectrometry (AP-MS) to identify macrophage interactors of CHIKV glycoproteins that are involved in CHIKV production. We discovered and validated that E1 interacts with signal peptidase complex subunit 3 (SPCS3) and eukaryotic translation initiation factor 3 subunit K (eIF3k), which block CHIKV production in macrophages. Importantly, the E1-V220I mutation significantly reduces E1 binding to both SPCS3 and eIF3k, suggesting that the positive selection signature is driven by these host restriction factors. Despite its role as a translation initiation factor, eIF3k exhibits both cytoplasmic and nuclear localization. Interestingly, we observed translocation of eIF3k from the nucleus to the cytoplasm and increased colocalization with SPCS3 upon CHIKV infection. Interrogation of eIF3k anti-CHIKV mechanism in CRISPR-Cas9 knockout (KO) cells showed that eIF3k specifically inhibits CHIKV production through its HAM protein domain in a translation-independent manner. Taken together, we found that, in addition to their critical function in viral entry and egress, CHIKV glycoproteins may interfere with cellular restrictions to facilitate virion production and spread in macrophages.

# Results

## CHIKV infects human macrophages more efficiently than other arthritogenic alphaviruses

To evaluate the susceptibility of macrophages to different arthritogenic alphaviruses, we infected human primary monocyte-derived macrophages with EGFP-expressing Sindbis virus (SINV), Ross river virus (RRV), ONNV, and CHIKV, and quantified infection levels at 24 h post infection (h.p.i.) by flow cytometry (Fig. 1A). Despite generally low infection rates with these alphaviruses (<1%), we observed a small percentage (0.76%) of macrophages highly infected with CHIKV, according to intracellular EGFP expression that spans 3 logs. We then compared the growth kinetics of CHIKV and its closest relative, ONNV, in infected human monocytic cell line THP-1-derived macrophages by quantifying virion production in the supernatant (Fig. 1B). We found that CHIKV produces two to three logs higher titers than ONNV throughout the infection time-course (up to $1.05 \times 10^7$ pfu/ml for CHIKV compared to $3.75 \times 10^4$ pfu/ml for ONNV), with the titers of both viruses peaking at 24 h.p.i. We also compared ONNV SG650 infection to infections with pathogenic CHIKV La Réunion (LR2006 OPY1) and Asian (AF15561) strains in THP-1 derived macrophages (Fig. EV1A). We demonstrated that pathogenic CHIKV infections result in higher levels of virion production in comparison to ONNV infection. These results suggest that a small number of CHIKV-infected macrophages are extremely efficient at producing viral progeny.

We asked whether the high level of CHIKV production is achieved by enhanced viral replication in macrophages. To bypass viral entry, we directly transfected in vitro transcribed genomic viral RNAs (vRNAs) of CHIKV and ONNV into THP-1-derived macrophages (Fig. 1C). We measured intracellular negative-sense viral RNA ((-) vRNA), the replicative intermediate, by TaqMan RT-qPCR assays. To our surprise, the (−) vRNA levels of ONNV are significantly higher than those of CHIKV following vRNA transfection, suggesting CHIKV infection is enhanced at a step after genome replication in macrophages. Nevertheless, virion production of CHIKV is dramatically more robust than that of ONNV and could be detected as early as 8 h post transfection (h.p.t.) (Fig. 1D). Taken together, human macrophage infection with CHIKV drives more superior virion production than that with ONNV.

## CHIKV E2 and E1 synergize to mediate efficient virion production in THP-1-derived human macrophages

To identify the viral determinants for CHIKV infection of human macrophages, we constructed several CHIKV-ONNV chimeras (Fig. 2A) and assessed their infection levels in THP-1 derived macrophages, compared to parental CHIKV and ONNV. Alphaviruses express four nonstructural proteins (nsP1-4) for viral replication, and five structural proteins from a subgenomic mRNA (capsid, E3, E2, 6K/TF, E1) for viral particle assembly and host cell entry (Kafai et al, 2022). These proteins are proteolytically processed from the nonstructural and structural polyproteins. Given the genome organization, we generated Chimera I that contains ONNV nsP1 to capsid in a CHIKV backbone, and Chimera III, that contains CHIKV nsP1 to capsid in an ONNV

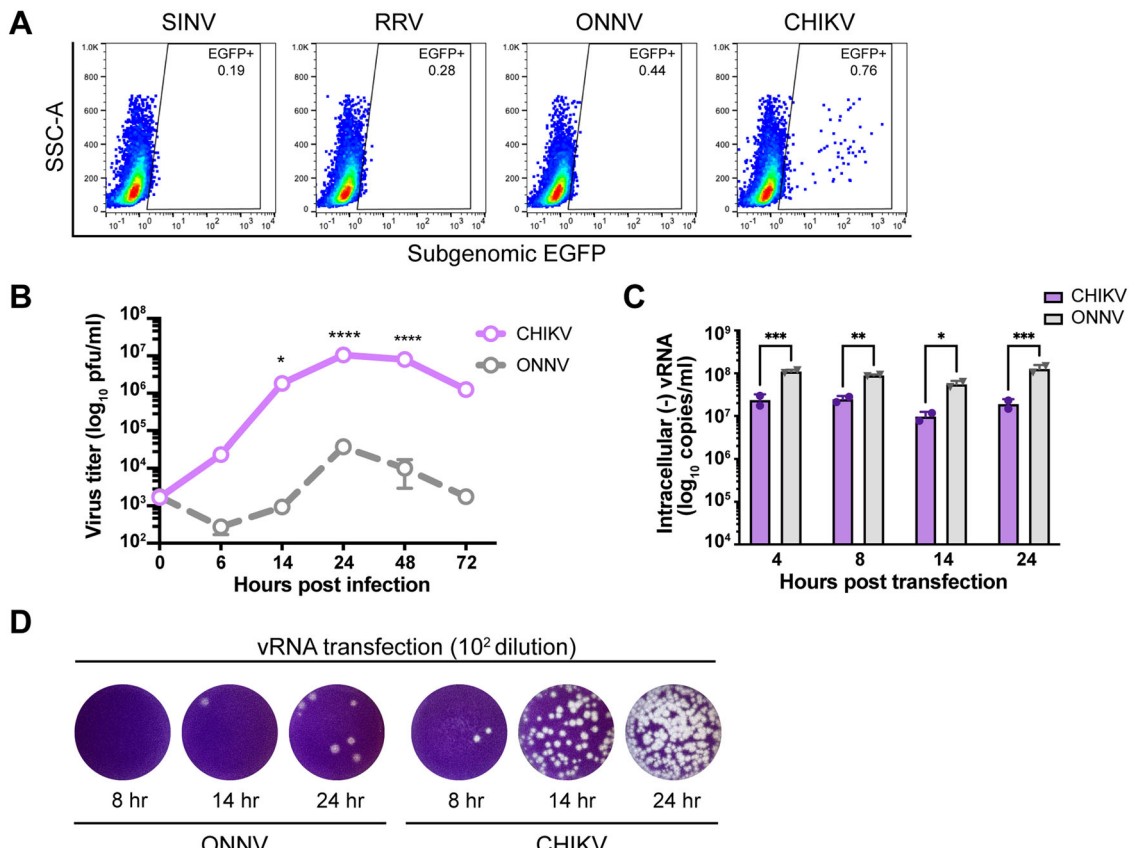

**Figure 1. Efficient CHIKV infection in human macrophages depends on a high level of virion production.**

(A) Human peripheral monocyte-derived macrophages were infected with EGFP-labeled alphaviruses (SINV TE/5′2 J, RRV strain T48, ONNV strain SG650, and CHIKV vaccine strain 181/clone 25) at MOI of 5 for 24 h. Levels of infection with different alphaviruses were determined by percent EGFP-positive cells evaluated by flow cytometry. Data are representative of 2 independent experiments performed in biological duplicates. (B) THP-1-derived macrophages were infected with CHIKV 181/clone 25 or ONNV SG650 at MOI 5. Titration of supernatant virus samples was performed at 0, 6, 14, 24, 48, and 72 h.p.i by plaque assay on BHK-21 cells. Data were representative of two independent experiments. Mean values of biological duplicates were plotted with SD. Asterisks indicate statistically significant differences as compared to ONNV (two-way ANOVA and Šidák's multiple comparisons test: 14 h *$p = 0.0128$; 24 h and 48 h ****$p < 0.0001$). (C) Levels of intracellular ($-$) vRNAs, the viral replicative intermediate, at 4, 8, 14, and 24 h post transfection of THP-1 derived macrophages with CHIKV 181/clone 25 or ONNV SG650 viral RNAs (vRNAs) were quantified through RT-qPCR with specific TaqMan probes. Data were representative of two independent experiments. Mean values of biological duplicates measured in technical duplicates were plotted with SD (Two-way ANOVA and Šidák's multiple comparisons test: 4 h ***$p = 0.0006$; 8 h **$p = 0.004$; 14 h *$p = 0.0299$; 24 h ***$p = 0.0001$). (D) CHIKV and ONNV titers of supernatant samples collected from transfected THP-1 derived macrophages in (C) were determined by plaque assay. The incubation period for plaque assay took 40 h. Representative plaques of CHIKV and ONNV from two independent experiments (1:100 dilution) are shown. Source data are available online for this figure.

backbone. To account for potential discrepancies associated with mismatched subgenomic promoters located at the 3′ end of nsP4 and structural proteins, we also generated Chimeras II and IV, where the swapping of viral genes starts with the subgenomic promoters in CHIKV and ONNV nsP4. We found comparable levels of virion production of Chimeras I and II as CHIKV in the supernatant of infected macrophages, while Chimeras III and IV recapitulate ONNV production (Fig. 2B). These data demonstrate that the viral determinants for effective macrophage infection lie in the CHIKV E3-E2-6K-E1 structural polyprotein region.

To investigate the role of CHIKV structural proteins in virion production, we transfected vRNAs of CHIKV, ONNV, and Chimeras I-IV into THP-1-derived macrophages to bypass viral entry. We compared viral replication and production among the transfected cells at 24 h.p.t. based on intracellular positive-sense viral RNA ((+) vRNA) levels and supernatant titers (Fig. 2C).

Consistent with Fig. 2B, transfection of viral genomes without CHIKV E3-E2-6K-E1 (ONNV, Chimera III, and Chimera IV) led to lower levels of virion production.

To further narrow down the viral determinants for CHIKV infection in macrophages, we constructed three additional chimeras in the context of Chimera III to include CHIKV E3 (Chimera III-I), E3-E2 (Chimera III-II), or E3-E2-6K (Chimera III-III) (Fig. 2D). Upon macrophage infection with CHIKV, ONNV, and the chimeras, we found that only Chimera III-II and Chimera III-III, both possessing CHIKV E2, partially enhance virion production at 24 and 48 h.p.i. although not significantly (Fig. 2E), suggesting that E2 alone is not sufficient. Chimera III-III with all the CHIKV structural proteins except E1 fails to fully rescue virion production in macrophages. Taken together, this supports the involvement of both CHIKV E2 and E1 in virion production.

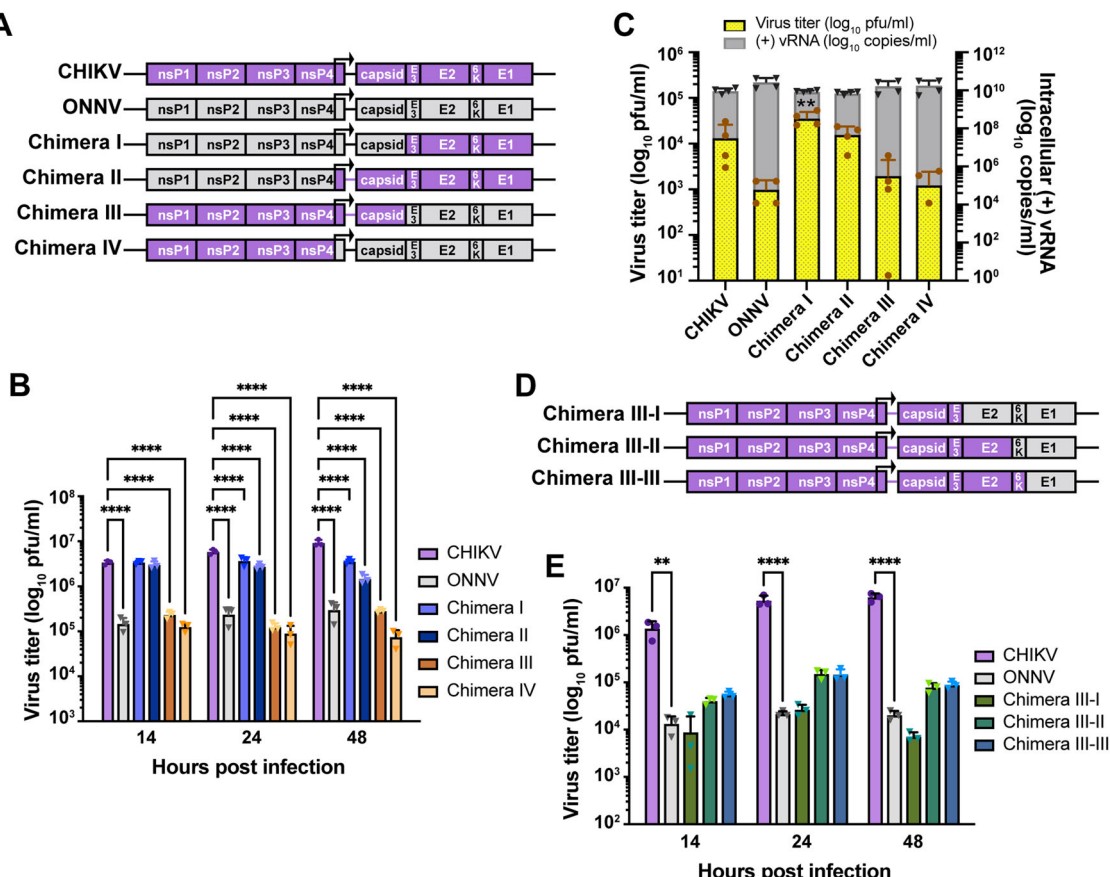

**Figure 2. Viral glycoproteins are critical determinants for macrophage tropism of CHIKV.**

(A) Schematic representation of CHIKV, ONNV, Chimera I, II, III, and IV. These chimeras consist of genomes from CHIKV vaccine strain 181/clone 25 and ONNV SG650 in different ratios: Chimera I contains the ONNV genome from nsP1 to capsid and the CHIKV genome from E3 to E1. Chimera II contains the CHIKV genome from nsP1 to the region prior to the subgenomic promoter in nsP4 and the CHIKV genome from the subgenomic promoter to E1. Chimera III contains the CHIKV genome from nsP1 to capsid and the ONNV genome from E3 to E1. Chimera IV contains the CHIKV genome from nsP1 to the region prior to the subgenomic promoter in nsP4 and the ONNV genome from the subgenomic promoter to E1. (B) Titration of supernatant samples from THP-1 derived macrophages infected with CHIKV 181/clone 25, ONNV SG650, and 4 chimeras (I, II, III, IV). The macrophages were inoculated with the virus at MOI 5, and the supernatant samples were collected at 14, 24, and 48 h.p.i for plaque assay analysis. The incubation period for plaque assay took 28 h. Data were representative of three independent experiments. Mean values of biological triplicates were plotted with SD. Asterisks indicate statistically significant differences as compared to CHIKV (Two-way ANOVA and Dunnett's multiple comparisons test: 14, 24, and 48 h ****$p < 0.0001$). (C) THP-1-derived macrophages were transfected with 0.5 µg RNA of CHIKV 181/clone 25, ONNV SG650, or chimeras (I, II, III, IV). Virion productions were determined by intracellular (+) vRNA transcript levels and supernatant infectious particle titers through RT-qPCR and plaque assay, respectively. The incubation period for plaque assay took 40 h. Data were plotted with the mean value of four biological replicates from two independent experiments. The error bar represents SD. Asterisks indicate statistically significant differences as compared to CHIKV (two-way ANOVA and Dunnett's multiple comparisons test: viral titer of CHIKV vs Chimera I **$p = 0.0036$). (D) Schematic representation of Chimera III-I, III-II, and III-III. Chimera III-I contains the CHIKV genome from nsP1 to E3 and the ONNV genome from E2 to E1. Chimera III-II contains the CHIKV genome from nsP1 to E2 and the ONNV genome from 6K to E1. Chimera III-III contains the CHIKV genome from nsP1 to 6 K and ONNV E1. (E) Titration of supernatant samples from THP-1 derived macrophages infected with CHIKV 181/clone 25, ONNV SG650, or chimeras (III-I, III-II, III-III) for 14, 24, and 48 h. The infection conditions and virus titer assessments were performed as previously described in (B). Data were representative of two independent experiments. Mean values of biological triplicates measured were plotted with SD. Asterisks indicate statistically significant differences as compared to ONNV (two-way ANOVA and Dunnett's multiple comparisons test: 14 h CHIKV vs ONNV **$p = 0.0092$; 24 and 48 h CHIKV vs ONNV ****$p < 0.0001$). Source data are available online for this figure.

To pinpoint the impact of CHIKV E2 and E1 on virion production, we generated three chimeras in the ONNV backbone with CHIKV replacement of E2 (ONNV/CHIKV E2), E1 (ONNV/CHIKV E1), or both E2 and E1 (ONNV/CHIKV E2 + E1) (Fig. 3A). Neither single replacement of CHIKV E2 nor E1 rescues ONNV infection of macrophages to comparable levels as CHIKV (Fig. 3B). Surprisingly, macrophage infection with ONNV/CHIKV E1 is more attenuated than that with ONNV. In contrast, the simultaneous replacement of E2 and E1 with CHIKV homologs (ONNV/CHIKV E2 + E1) increased the supernatant titers to levels even higher than those of CHIKV. We

then transfected vRNAs into macrophages to evaluate viral replication and production (Fig. 3C). All of the transfected vRNAs launched productive viral replication in macrophages; however, only the transfection of ONNV/CHIKV E2 + E1 RNA led to significantly enhanced virion production, albeit at levels lower than those for transfection of CHIKV RNA (Fig. 3C).

In order to further characterize the viral particles released by infected macrophages, we compared the particle-to-PFU ratios among ONNV, CHIKV, Chimera I (Fig. 2A), and ONNV/CHIKV E2 + E1 (Fig. 3A). Importantly, we found ONNV and CHIKV to

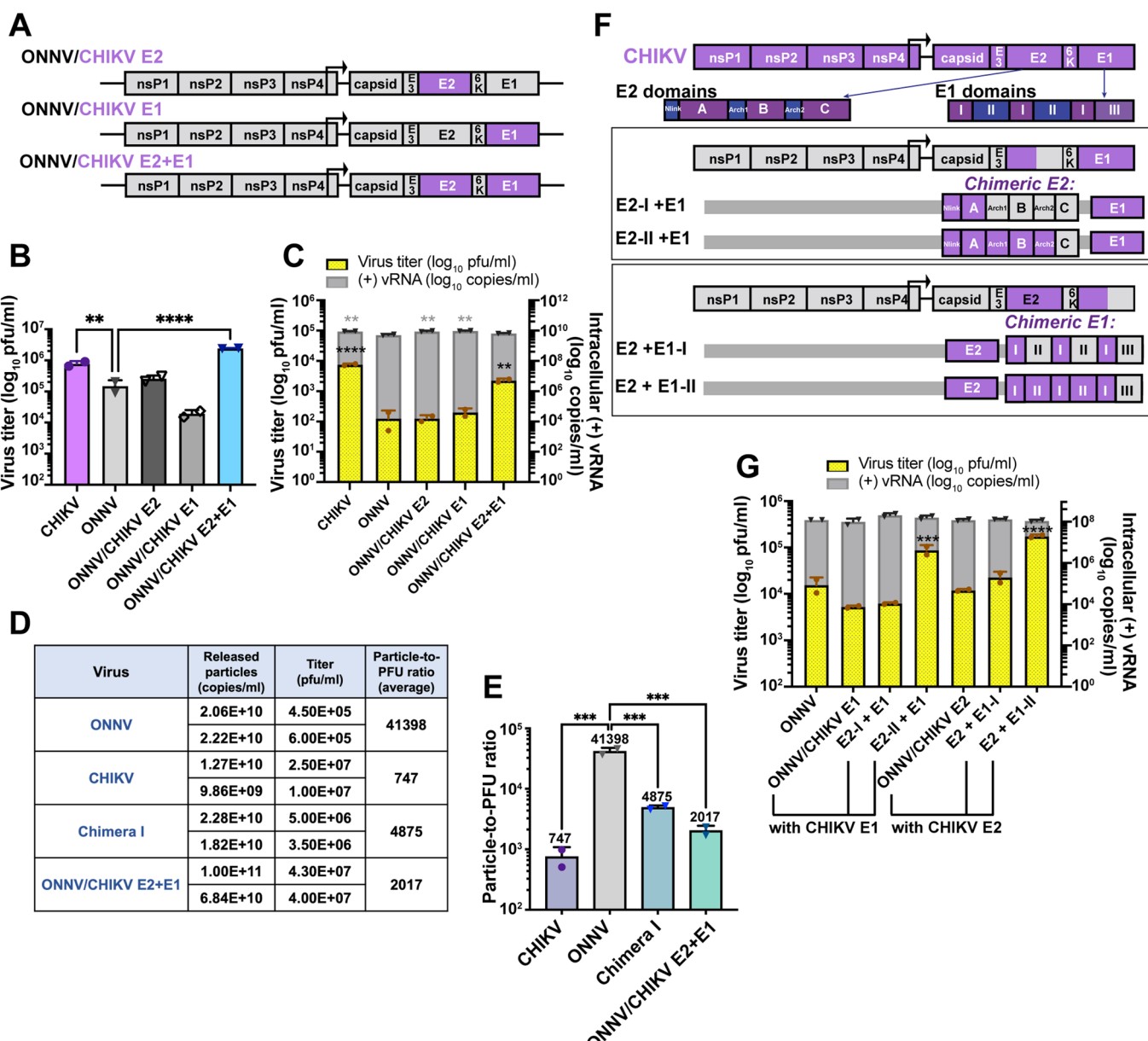

have the highest (41398) and lowest (747) particle-to-PFU ratios, respectively (Fig. 3D,E). Consistent with that, either replacing the entire glyco-polyprotein or just E2 and E1 with the CHIKV homologs significantly decreased the particle-to-PFU ratios to 4875 and 2017, respectively, highlighting increased infectivity mediated by CHIKV glycoproteins. Since alphaviruses utilize the host secretory pathway for glycoprotein processing and maturation, we questioned whether the secretory pathway confers an advantage on CHIKV E2 and E1 proteins during the late stage of the viral life cycle in macrophages. We infected THP-1-derived macrophages, which had been treated with FLI-06 and Golgicide A (GCA), with ONNV, CHIKV, Chimera I, and ONNV/CHIKV E2 + E1 (Fig. EV1B). Golgicide A is a reversible inhibitor of Golgi-specific brefeldin A-resistance guanine-nucleotide exchange factor 1 (GBF1), an ARF-GEF (guanine-nucleotide exchange factors for ADP-ribosylation factor GTPases) in cis-Golgi, which leads to

rapid disassembly of the Golgi and trans-Golgi network (TGN) (Saenz et al, 2009). FLI-06 interferes with cargo recruitment to ER-exit sites and disrupts Golgi without depolymerizing microtubules or interfering with GBF1 (Krämer et al, 2013). The plaque assay result shows that all the viruses are sensitive to secretory pathway disruption, however, the production of viruses containing CHIKV glycoproteins (CHIKV, Chimera I, and ONNV/CHIKV E2 + E1) is significantly more attenuated by FLI-06 and GCA. This demonstrates the greater dependence of CHIKV glycoproteins on the host secretory pathway for productive infection in macrophages.

In addition, we found that infections with ONNV, CHIKV, and chimeric viruses are less restricted in 293 T cells resulting in more robust virion production (over $10^8$ pfu/ml) (Fig. EV3A). There is no significant difference in virion production in 293T cells between CHIKV and ONNV, between CHIKV and Chimera I (Fig. 2A; ONNV with CHIKV poly-glycoproteins), between CHIKV and

**Figure 3. CHIKV E2 and E1 dramatically increase the specific infectivity of viral particles secreted from human macrophages without affecting viral RNA replication.**

(A) Schematic representation of chimera ONNV/CHIKV E2, ONNV/CHIKV E1 and ONNV/CHIKV E2 + E1. These three chimeric viruses were built on ONNV backbone with the replacement of CHIKV E2 (ONNV/CHIKV E2), E1 (ONNV/CHIKV E1), or both E2 and E1 (ONNV/CHIKV E2 + E1). (B) Titration of supernatant samples from THP-1 derived macrophages infected with CHIKV vaccine strain 181/clone 25, ONNV SG650, ONNV/CHIKV E2, ONNV/CHIKV E1, and ONNV/CHIKV E2 + E1. Macrophages were inoculated with the viruses at MOI 5, and the supernatant samples were collected at 24 h.p.i for plaque assay analysis. The incubation period for plaque assay took 28 h. Data were representative of two independent experiments. Mean values of biological duplicates were plotted with SD. Asterisks indicate statistically significant differences as compared to ONNV (two-way ANOVA and Dunnett's multiple comparisons test: ONNV vs CHIKV **$p$ = 0.004; ONNV vs ONNV/CHIKV E2 + E1****$p$ < 0.0001). (C) THP-1 derived macrophages were transfected with 0.5 µg RNA of CHIKV 181/clone 25, ONNV SG650, ONNV/CHIKV E2, ONNV/CHIKV E1, or ONNV/CHIKV E2 + E1. Virion production was determined by intracellular (+) vRNA transcript levels and supernatant infectious particle titers through RT-qPCR and plaque assay, respectively. The incubation period for plaque assay took 40 h. Data were representative of three independent experiments. Mean values of biological duplicates were plotted with SD. Asterisks indicate statistically significant differences as compared to ONNV (one-way ANOVA and Dunnett's multiple comparisons test: viral titer of ONNV vs CHIKV ****$p$ < 0.0001; viral titer of ONNV vs ONNV/CHIKV E2 + E1**$p$ = 0.0058; viral copies of ONNV vs CHIKV **$p$ = 0.0031; viral copies of ONNV vs ONNV/CHIKV E2 **$p$ = 0.0034; viral copies of ONNV vs ONNV/CHIKV E1 **$p$ = 0.0019). (D, E) Particle-to-PFU ratios of ONNV, CHIKV, and chimeric viruses containing CHIKV glycoproteins. THP-1-derived macrophages were infected with ONNV SG650, CHIKV 181/clone 25, Chimera I (refer to Fig. 2A schematic), and ONNV/CHIKV E2 + E1 at MOI 5 for 24 h. The viral particle numbers in the supernatant were quantified by TaqMan qPCR assay with specific probes targeting nsP1 in (+) RNA. Virus titers were determined by plaque assay on BHK-21 cells. The incubation period for plaque assay took 28 h. Data were representative of two independent experiments, each of which has biological duplicate samples. The viral copy numbers and titers of each duplicate and their averaged values are shown in (D) and summarized as bar charts in (E). Asterisks indicate statistically significant differences as compared to ONNV (one-way ANOVA and Dunnett's multiple comparisons test: ONNV vs CHIKV***$p$ = 0.0005; ONNV vs Chimera I***$p$ = 0.0008; ONNV vs ONNV/CHIKV E2 + E1***$p$ = 0.0006). (F) Schematic representation of modified chimeras based on parental ONNV/CHIKV E1 + E2 that contain hybrid E2 or E1. E2 has three domains: A and B connected to A and C by two flanking β-ribbon arches, and C. E1 has three domains: I, II, and III, with a fusion loop in II. Chimera containing hybrid E2 that has arch-B-arch-C (E2-I + E1), or only domain C (E2-II + E1) from ONNV. Chimera containing hybrid E1 has domains II and III (E2 + E1-I), or only domain III (E2 + E1-II) from ONNV. (G) THP-1 derived macrophages were transfected with 0.5 µg RNA of ONNV SG650, ONNV/CHIKV E1 and chimeras (E2-I + E1, E2-II + E1), ONNV/CHIKV E2 and chimeras (E2 + E1-I, E2 + E1-II). Virion production was determined through RT-qPCR and plaque assays as described in (C). Data were representative of four independent experiments. Mean values of biological duplicates were plotted with SD. Asterisks indicate statistically significant differences as compared to ONNV (one-way ANOVA and Dunnett's multiple comparisons test: viral titer of ONNV vs E2-II + E1***$p$ = 0.0008; viral titer of ONNV vs E2 + E1-II****$p$ < 0.0001). Source data are available online for this figure.

Chimera III (Fig. 2A; CHIKV with ONNV poly-glycoproteins). Interestingly, infection of 293 T cells with ONNV/CHIKV E2 + E1 is significantly more productive than that with the parental CHIKV and ONNV viruses. These results clearly demonstrate that ONNV infection is not as attenuated in 293T cells as in macrophages, and hence the requirement for CHIKV structural proteins is highly specific to macrophage infection.

To map the viral determinants for virion production to specific domains, we strategically swapped in ONNV E2 or E1 domains in the context of ONNV/CHIKV E2 + E1. Alphavirus E2 comprises three domains (A, B, and C) connected by β-ribbon arches, with domains A and B functioning in receptor binding and cellular attachment (Li et al, 2010). Alphavirus E1 consists of three β-barrel domains: I, II, and III, with the fusion peptide embedded in domain II critical for viral fusion and uncoating. We generated two chimeras that contain total CHIKV E1 and partial domains of CHIKV E2 in the ONNV backbone (Fig. 3F, E2-I + E1 and E2-II + E1). We also constructed two chimeras that contain total CHIKV E2 and partial domains of CHIKV E1 in the ONNV backbone (Fig. 3F, E2 + E1-I and E2 + E1-II). We transfected macrophages with vRNAs of these chimeras in comparison with ONNV, ONNV/CHIKV E1, and ONNV/CHIKV E2 to measure virion production (Fig. 3G). We found that only the chimeras containing CHIKV E2 without domain C or E1 without domain III restore virion production to significantly high levels. These results suggest that glycoprotein determinants crucial for virion production in macrophages may lie in CHIKV E2 domain B and flanking β-ribbon arches, and E1 domain II.

## Positively selected residues in E2 and E1 are essential for CHIKV production in THP-1-derived human macrophages

Recent SARS-CoV-2 studies have harnessed the power of complementary selection analyses to reveal residues under positive

selection that might promote virus adaptation and expansion in human hosts (MacLean et al, 2021; Maher et al, 2022; Kistler et al, 2022). CHIKV, like SARS-CoV-2, is a zoonotic virus well-adapted to humans. Therefore, we asked whether residues in the CHIKV glycoproteins have been under positive selection to overcome antiviral immunity and productively replicate in macrophages. We applied the same methodology from a highly cited SARS-CoV-2 study (MacLean et al, 2021) to analyze the evolutionary selection sites in the CHIKV structural proteins from patient isolates. The analysis pipeline is depicted in Fig. 4A: 397 CHIKV sequences isolated from infected individuals globally were obtained from the NCBI virus (Hatcher et al, 2017) database. The structural polyprotein sequences of these isolates were aligned through MUSCLE (Edgar, 2004) and built into a phylogenetic tree with IQ-TREE (Minh et al, 2020; Trifinopoulos et al, 2016) (Fig. EV2A) for positive selection site detection. The positively selected residues in CHIKV structural proteins were finally identified by the fixed effects likelihood (Kosakovsky Pond and Frost, 2005) (FEL, $p < 0.05$) and mixed effects model of evolution (Murrell et al, 2012) (MEME, $p < 0.05$). FEL identified four amino acid residues in E2, 6K, and E1 under pervasive positive selection; MEME identified 14 residues in the capsid, E2, 6K, and E1 under pervasive and episodic positive selection, including all four residues identified by FEL (Figs. 4B,C and EV2B).

Interestingly, the positively selected sites identified by MEME were concentrated in E2 and E1 (Fig. 4B). We found three residues in E2 (E2-V135, E2-A164, E2-A246) and three in E1 (E1-E211, E1-V220, E1-R366) to be different between our experimental strains, CHIKV vaccine strain 181/clone 25 and ONNV strain SG650 (Fig. EV2B). Next, we compared these six evolutionary sites in E2 and E1 of additional ONNV and CHIKV strains (Fig. EV2D,E). Four of these sites (E2-135, E2-246, E1-220, E1-366) are conserved among all ONNV strains (E2-135L, E2-246S, E1-220I, E1-366K) and all CHIKV strains (E2-135V, E2-246A, E1-220V, E1-366R). On

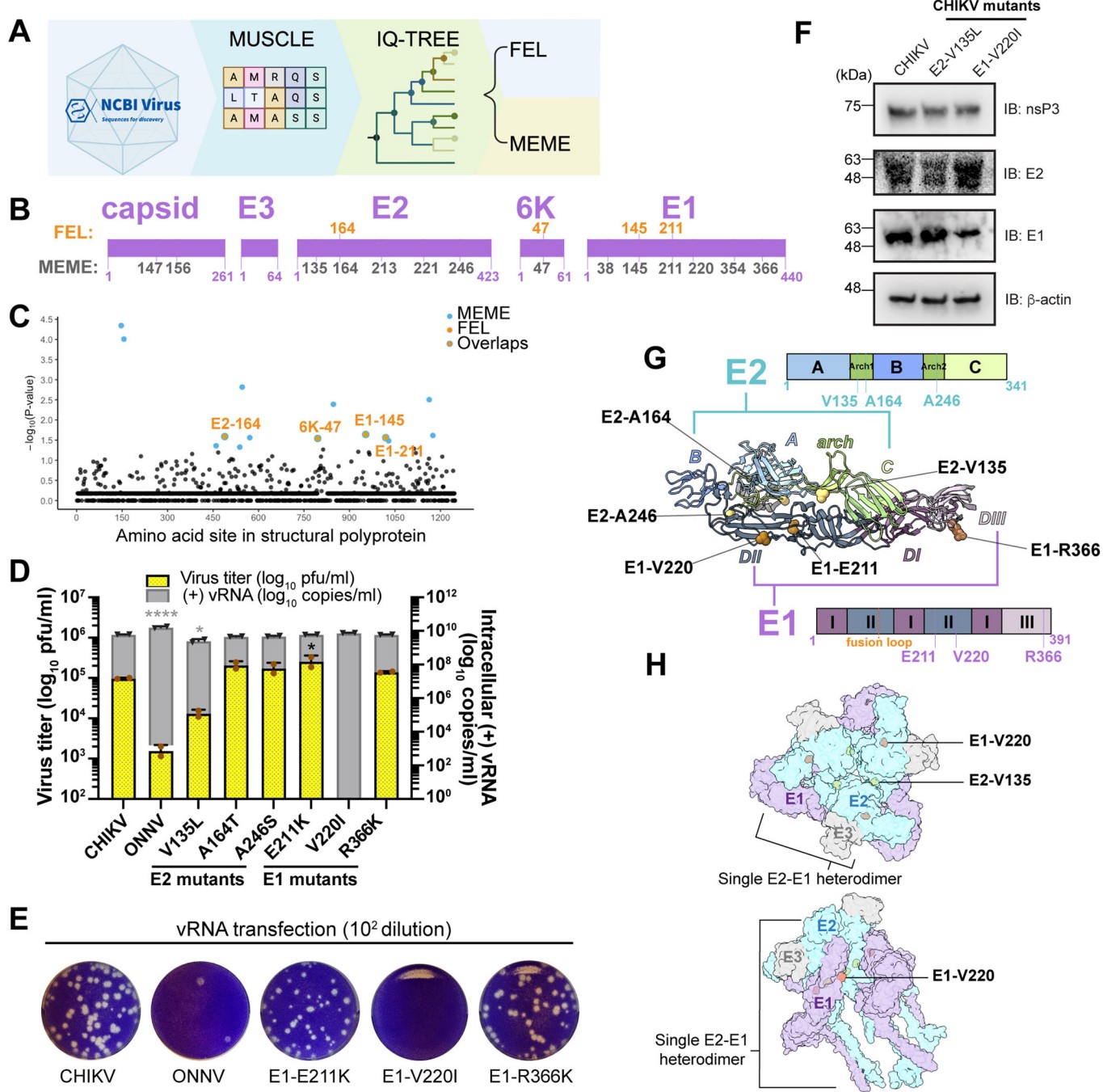

the other hand, the E2-164 and E1-211 sites encode for two different amino acids (E2-164T/A, E1-211K/E) that can be found in either one of the two viruses or both. These alignments suggest that most of the positively selected sites are conserved across different CHIKV strains.

To interrogate if these positively selected residues affect CHIKV production, we mutated them individually into the homologous residues in ONNV. We compared viral replication and production of these mutants (E2-V135L, E2-A164T, E2-A246S, E1-E211K, E1-V220I, and E1-R366K) with that of parental CHIKV in vRNA-transfected THP-1 derived macrophages (Fig. 4D). The E2-V135L

mutation decreases virus titers by about 1 log and significantly reduces intracellular (+) vRNA levels. Strikingly, the E1-V220I mutation completely abrogates virion production in macrophages without affecting viral replication (Fig. 4D,E), suggesting a defect in the viral life cycle after genome replication. In contrast, both E2-V135L and E1-V220I mutations attenuate viral replication and production in 293T and BHK-21 cells (Fig. EV3B,C). In addition, the E2-V135L and E1-V220I mutations do not affect expression of the viral nonstructural protein nsP3 and only slightly reduce the expression of viral glycoproteins E2 and E1, respectively, further supporting a defect in virion assembly and/or exit after structural

**Figure 4.  CHIKV E2 and E1 residues under positive selection are essential for virion production in human macrophages.**

(A) The pipeline for analyzing natural selection in the evolution of CHIKV structural proteins in human hosts. 397 CHIKV sequences isolated from infected individuals globally were downloaded from the NCBI virus database, and structural polyprotein sequence alignment was performed by MUSCLE(Edgar, 2004). The phylogenetic tree of CHIKV was constructed based on the maximum-likelihood (ML) optimality criterion with IQ-TREE (Minh et al, 2020; Trifinopoulos et al, 2016). The sites under positive selection were identified using mixed effects model of evolution (MEME) (Murrell et al, 2012) and fixed effects likelihood (FEL) (Kosakovsky Pond and Frost, 2005). (B) The positively selected sites identified by FEL or MEME are annotated in each CHIKV structural protein. The sites identified by MEME are colored in dark gray. Four of these sites (E2-164, 6K-47, E1-145, and E1-211) were identified with both methods and are colored in orange. (C) The positively selected sites in CHIKV structural proteins are plotted with the y-axis of $-\log_{10} P$ values (determined by MEME or FEL) and the x-axis of the amino acid locations in the full-length structural polyprotein (from the beginning of capsid to the end of E1). The $P$ values are generated by the FEL or MEME algorithm and adjusted with Benjamini–Hochberg correction. The statistically significant sites identified by MEME ($p < 0.05$) are in blue, the ones identified by FEL (FEL $p < 0.05$) are in orange, and the ones identified by both methods (MEME $p < 0.05$ and FEL $p < 0.05$) are in orange with blue circles. (D) Comparison of virion production of CHIKV positive selection site mutants in THP-1 derived macrophages. The positive selection site in E2 or E1 of CHIKV 181/clone 25 was mutated to the homologous residue in ONNV, respectively, to generate six CHIKV mutants (E2-V135L, E2-A164T, E2-A246S, E1-E211K, E1-V220I, and E1-R366K). Macrophages were transfected with 0.5 µg RNA of CHIKV, ONNV, or CHIKV positive selection site mutants, and virion productions were determined by intracellular (+) vRNA transcript levels and supernatant infectious particle titers as previously described. Data were representative of three independent experiments. Mean values of biological duplicates were plotted with SD. Asterisks indicate statistically significant differences as compared to CHIKV (One-way ANOVA and Dunnett's multiple comparisons test: viral titer of CHIKV vs E211K *$p = 0.0414$; viral copies of CHIKV vs ONNV ****$p < 0.0001$; viral copies of CHIKV vs V135L *$p = 0.0228$). (E) Representative plaque images of CHIKV E1 positive selection site mutants (E1-E211K, E1-V220I, E1-R366K) in comparison with CHIKV and ONNV. Plaque assays were performed on supernatant samples from transfected THP-1-derived macrophages as mentioned in (D). The incubation period for plaque assay is 40 h. The representative plaques from the 1:100 dilution are shown here. (F) The expression levels of viral nonstructural and structural proteins of CHIKV wild-type, E2-V135L, and E1-V220I mutants in THP-1 derived macrophages. The THP-1-derived macrophages were transfected with viral RNAs of CHIKV, E2-V135L, or E1-V220I mutant for 48 h. The expression levels of viral nsP3, E2, and E1 proteins were evaluated through immunoblotting. (G) Visualization of positively selected sites in single E2/E1 heterodimer with the presence of E3 from infectious CHIKV 181/clone 25 virus particle. The heterodimer structure was downloaded from PDB (6NK7)(Basore et al, 2019) and visualized in Chimera X (Pettersen et al, 2021). The positively selected sites E2-V135, E2-A164, and E2-A246 are located in β-ribbon arches flanking domain B in E2. The positively selected sites E1-E211 and E1-V220 are located in domain II in E1. The positively selected site E1-R366 is in domain III in E1. (H) The locations of E2-V135 (yellow nodes) and E1-V220 (orange nodes) in trimerized E2/E1 heterodimers (PDB: 6NK7). The E2 (cyan), E1 (purple), and E3 (gray) were annotated to show a single heterodimer unit. Source data are available online for this figure.

protein translation (Fig. 4F). Given the importance of E2-V135 and E1-V220 in CHIKV production, we analyzed the amino acid heterogeneity at these two sites in the original 397 CHIKV primary isolates from NCBI Virus database (Fig. EV2C). Most of the amino acids at E2-135 and E1-220 are valine. This suggests that the valine residues at the positively selected sites E2-135 and E1-220 are crucial for CHIKV fitness and strongly selected during viral evolution.

While all six unique CHIKV residues are on the exterior of a single E2/E1 heterodimer (with E3) (Fig. 4G), E2-V135 and E1-V220 also interface with E2 from the neighboring heterodimer in trimerized E2/E1 heterodimer configuration (with E3) (Fig. 4H), according to the recently solved CHIKV vaccine strain 181/clone 25 structure (Basore et al, 2019). Meanwhile, E1-V220 is partially embedded in the groove formed by E1 and the neighboring E2, which may provide additional docking sites for host interactors (Fig. 4H). Interestingly, E2-V135 and E1-V220 are in the E2 β-ribbon arch and E1 domain II (Fig. 4G), respectively, that were identified to be critical for virion production in Fig. 3G. Taken together, the positively selected residue E1-V220 mediates efficient virion production likely by facilitating host factor binding in macrophages.

## Identification of cellular factors that interact with CHIKV glycoproteins in macrophages

Successful virion production requires the maturation of E2/E1 heterodimer for proper virion assembly which involves proteolytic processing of the precursor (E3-E2-6K-E1) to an intermediate form (p62/E1), and finally to the E2/E1 heterodimer in the secretory pathway (Brown et al, 2018; Helenius, 1995; Ren et al, 2022). To investigate intracellular macrophage factors that interact with the uncleaved precursors or mature glycoproteins to affect CHIKV production, we inserted a myc tag in the genome of CHIKV vaccine

strain 181/clone 25 to label E2 N-terminally (CHIKV/myc-E2) that can also label the precursors in addition to E2/E1 heterodimers. We infected THP-1-derived macrophages in two independent experiments with either CHIKV/myc-E2 or untagged CHIKV vaccine strain 181/clone 25 (WT, negative control). We performed myc immunoprecipitation to enrich for uncleaved polyprotein E3-myc-E2-6K-E1, E3-myc-E2 in p62/E1 heterodimer and myc-E2 in mature E2/E1 heterodimer, followed by MS analysis of the resultant protein mixtures to identify interactors (Fig. 5A).

We identified 1157 proteins (Log2FC >0, $p < 0.05$) in the second experiment to be significantly enriched in CHIKV/myc-E2-infected cells compared to CHIKV WT infected cells (Fig. EV4A; Dataset EV1). Most of the candidate interactors showed more than twofold abundance, with the top enriched protein being S100 calcium-binding protein A9 (S100A9) (Log2FC = 7.89) (Fig. 5B). In addition to the bait protein E2 (Log2FC = 5.84, $p = 2.11E-5$) that was significantly pulled down in CHIKV/myc-E2-infected macrophages, we also detected E1 (Log2FC = 4.09, $p = 1.59E-3$) and E3 (Log2FC = 2.99, $p = 3.08E-2$) as expected. We then used CORUM, an experimentally confirmed, high-confidence protein–protein interaction database, to decipher multiprotein complexes among the host proteins co-immunoprecipitated in CHIKV/myc-E2-infected macrophages (Fig. 5C). The predominantly identified protein complexes that strongly interact with CHIKV glycoprotein precursors and E2/E1 heterodimers include the respiratory chain complex I, SNARE complex, spliceosome, mediator complex, signal peptidase complex, emerin complex I, oligosaccharyltransferase OSTC-III complex, and eIF3 complex. These results suggest that CHIKV glycoproteins may intersect with or co-opt different protein complexes involved in diverse biological processes. Interestingly, signal peptidases are hijacked for polyprotein maturations of several viruses, including alphavirus (Neufeldt et al, 2018; Zimmerman et al, 2023), however, the exact peptidases involved are still largely unknown.

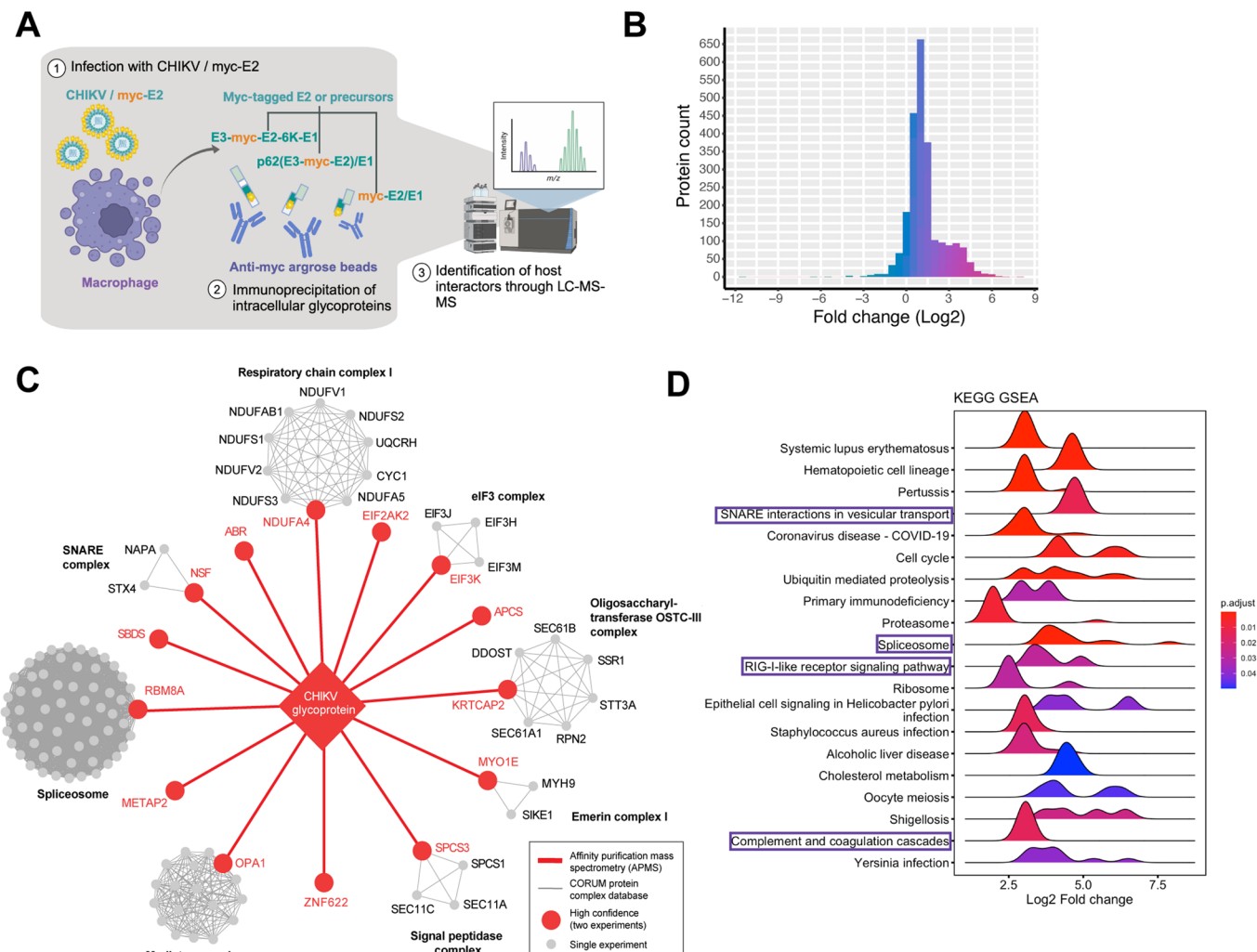

**Figure 5. Identifying host factors interacting with CHIKV glycoproteins in infected human macrophages by affinity purification-mass spectrometry (AP-MS).**

(A) The workflow of AP-MS analysis to identify host factors in THP-1 derived macrophages that interact with CHIKV glycoproteins. After 48 h infection with CHIKV/myc-E2, different forms of myc-tagged glycoproteins (polyprotein E3-myc-E2-6K-E1, E3-myc-E2/E1, myc-E2/E1) were pulled down by anti-myc agarose beads from the infected macrophage lysates and submitted to LC-MS/MS analysis to identify co-immunoprecipitated host factors. The co-immunoprecipitated proteins from untagged CHIKV vaccine strain 181/clone 25 (CHIKV WT)-infected macrophages serve as negative controls for proteomic analysis (not elaborated in this diagram). (B) The histogram of fold change distribution of all the identified macrophage proteins in the second independent AP-MS experiment that interact with myc-tagged glycoproteins (poly-glycoprotein E3-myc-E2-6K-E1, E3-myc-E2/E1, myc-E2/E1). (C) The interaction network between CHIKV glycoproteins and macrophage proteins. CHIKV glycoproteins interacting partners in THP-1 derived macrophages that were significantly enriched in both independent mass spectrometry experiments are depicted in red. Candidate interactors significantly enriched in at least one mass spectrometry experiment that belongs to existing red protein complexes are colored in gray. The protein complexes were identified through the CORUM database. (D) Gene set enrichment analysis (GSEA) of top 20 KEGG pathways in identified host factors summarized in ridgeplot. All the identified host factors are ranked according to the log2 expression fold change of proteins co-immunoprecipitated from CHIKV/myc-E2 infected macrophages with respect to proteins from CHIKV 181/clone 25 infected macrophages (x-axis). The significance of the KEGG enrichment is shown in a continuous color scale based on the adjusted P values, which are generated by Fisher's exact test and corrected by Benjamini–Hochberg. The histogram in each KEGG term is defined by the number of genes with a specific log2 fold change value. Created with BioRender.com.

Moreover, we visualized the overall biological processes of enriched host proteins through EnrichmentMap (Fig. EV4B). Consistent with the identified protein complexes, CHIKV glyco-protein interactors are mostly enriched in RNA processes (transcription regulation, pre-mRNA splicing), and secretory pathway (ER-Golgi transportation, intracellular vesicle transport, negative regulation of endopeptidase activity, signal peptide processing). Immune responses (type I IFN pathway/complement

activation, antigen presentation) are also among the biological processes targeted by CHIKV glycoproteins. Consistent with the enriched protein complexes and biological processes, the KEGG analysis (Subramanian et al, 2005; Wu et al, 2021; Kanehisa et al, 2016) identified similar pathways (Fig. 5D, framed), suggesting that RNA processes, secretory pathway, and immune responses are critical for CHIKV glycoprotein interactions with macrophage factors.

## CHIKV E1 binding proteins exhibit potent anti-CHIKV activities

We next inquired whether the host factors interacting with the CHIKV glycoproteins are proviral or antiviral. We selected 13 host factors for further investigation, including ten hits identified in both AP-MS experiments, classical ISGs (APOBEC3F, OAS3), and a myeloid-specific gene (S100A9) which is an endogenous ligand for toll-like receptor 4 (TLR4) (Foell et al, 2007; Vogl et al, 2007) (Fig. 6A). We knocked down these genes with pooled siRNAs (Fig. EV5A) in THP-1 derived macrophages, followed by CHIKV infection (Fig. 6B). We included nontargeting (NT) siRNA as negative control and siRNAs targeting pro-CHIKV factors G3BP stress granule assembly factor 1 (G3BP1) and 2 (G3BP2) (Scholte et al, 2015; Kim et al, 2016) as a positive control. Knockdown of most of the host factors led to elevated CHIKV titers in macrophages compared to NT-transfected cells, except for G3BP1 + 2 knockdown, indicating that many of the candidate E2 interactors have antiviral activities. In addition to the previously reported anti-CHIKV restriction factors OAS3 and PKR (Bréhin et al, 2009; Gorchakov et al, 2004; Ryman et al, 2005), knockdown of the host genes SPCS3 and EIF3K significantly restores virion production by about fivefold. To confirm that the antiviral activities observed in Fig. 6B are specific to a step after viral entry, we knocked down the same host factors in THP-1-derived macrophages followed by transfection of CHIKV vRNA (Fig. 6C). We found that silencing of most of the genes enhances virion production in vRNA-transfected macrophages. CHIKV production in macrophages with OAS3, SPCS3, and EIF3K knockdown is significantly higher than that in NT-transfected cells, despite similar intracellular vRNA levels.

To confirm the interaction of CHIKV glycoproteins with host proteins demonstrating antiviral activities (OAS3, SPCS3, eIF3k, APOBEC3F, and PKR, Fig. 6B,C), we transfected 293T cells with plasmids expressing 3xflag-tagged host factors, followed by transfection with CHIKV vRNA (Fig. 6D) or CHIKV poly-glycoprotein (E3-myc-E2-6K-E1) expressing plasmid (Fig. EV5B). The host factors were pulled down to probe for glycoproteins in precursor or mature forms. We consistently detected strong binding of E1 and moderate binding of E3-E2-6K-E1 to SPCS3 and eIF3k. However, it is surprising that neither SPCS3 nor eIF3k binds to E2 or p62, which is presumed to interact with E1 in heterodimer forms. To confirm the specific binding of SPCS3 and eIF3k to E1, we performed reciprocal immunoprecipitation. We transfected 293T cells with plasmids expressing the CHIKV poly-glycoprotein and SPCS3/eIF3k followed by E1 or E2 pulldown (Fig. EV5C,D). The reciprocal immunoprecipitation validated the specific interaction of SPCS3 and eIF3k with E1, respectively, while we did not observe consistent pulldown of the host factors with E2.

Given the unexpected absence of E2 in both host factor and E1/E2 pulldown, it is possible that a group of free E1 proteins unassociated with E2 has distinct functions in interfering with cytoplasmic host factors for efficient virion production. It would be interesting to determine whether the E1 proteins that interact with SPCS3 and eIF3k localize to a different cellular compartment away from E2. To address this question, we applied confocal laser-scanning microscopy with an Airyscan detector to identify colocalization among E2, E1, and host factors (SPCS3 or eIF3k) (Fig. 7A,B). We found that the majority of E2 accumulates at the plasma membrane, while E1 mostly localizes to the region adjacent to the nucleus, potentially the endoplasmic reticulum (ER). According to the Pearson correlation coefficient analysis (Fig. 7A,B, violin plots), E1 colocalizes more with the host factors (SPCS3, eIF3k) than with E2. These results suggest that E2 and E1 may not always be together in heterodimer forms, and the cytoplasmic pool of E1 associates more with SPCS3 and eIF3k.

To further solidify the role of E1 interaction with macrophage restriction factors during viral evolution, we investigated the effects of replacement of the positively selected site E1-V220 in CHIKV with the isoleucine found in the homologous ONNV site. We previously showed that the E1-V220I mutation completely abrogates virion production in THP-1-derived macrophages (Fig. 4D). Consistent with that, E1-V220I dramatically reduces E1 binding to SPCS3 and to eIF3k (Fig. 7C). This implies that these host restriction factors may be involved in genetic conflict with CHIKV glycoproteins through the same E1 interface, which may engage with both SPCS3 and eIF3k in a complex.

To determine whether SPCS3 and eIF3k work together, we quantified their colocalization in CHIKV-infected 293T cells through Airyscan microscopy (Fig. 7D). SPCS3 is mostly localized to the cytoplasm while eIF3k is found in both the cytoplasm and nucleus, consistent with previous reports on the strong nuclear localization of eIF3k (Salsman et al, 2013). Some SPCS3 and eIF3k colocalize in mock-infected cells, and upon CHIKV infection their colocalization significantly increases (Pearson correlation coefficient increases from 0.4 to 0.6). This suggests that SPCS3 and eIF3k may work together to inhibit CHIKV upon infection. Interestingly, we observed eIF3k translocation from the nucleus to the cytoplasm in some CHIKV-infected cells (white arrows in Fig. 7D). This supports a novel cytoplasmic function of eIF3k upon CHIKV infection. Taken together, the viral E1 glycoprotein has been engaged in an evolutionary arms race with host restriction factors in human macrophages which is distinct from its conventional role in E2/E1 heterodimer formation.

## E1-binding protein eIF3k inhibits CHIKV production through its HAM domain in a translation-independent manner

Since neither SPCS3 nor eIF3k was previously reported as a restriction factor before, we further characterized their roles in CHIKV infection by validating their antiviral activities in CRISPR-Cas9 knockout (KO) 293T cells. Although we failed to generate a complete KO of SPCS3 consistent with a previous report (Zhang et al, 2016), we successfully obtained single-cell clones of eIF3k KO in 293T cells (Fig. 8A; Appendix Fig. S1B). We compared different arthritogenic alphavirus infections in eIF3k KO 293T cells (Fig. 8B), including SINV that shows similarly low infection levels in primary monocyte-derived macrophages as ONNV (Fig. 1A). We found that eIF3k KO leads to increased CHIKV titer by ~2.5-fold while having no effects on virion production of ONNV and SINV, confirming the specificity of CHIKV inhibition by eIF3k.

Next, we investigated how eIF3k antagonizes CHIKV production. Previously, we showed that eIF3k blocks virion production without affecting viral genome replication (Fig. 6C). Since viral structural polyprotein expression from the subgenomic mRNA precedes virion assembly, we asked whether eIF3k acts at the step of viral structural protein translation. We transfected a CHIKV

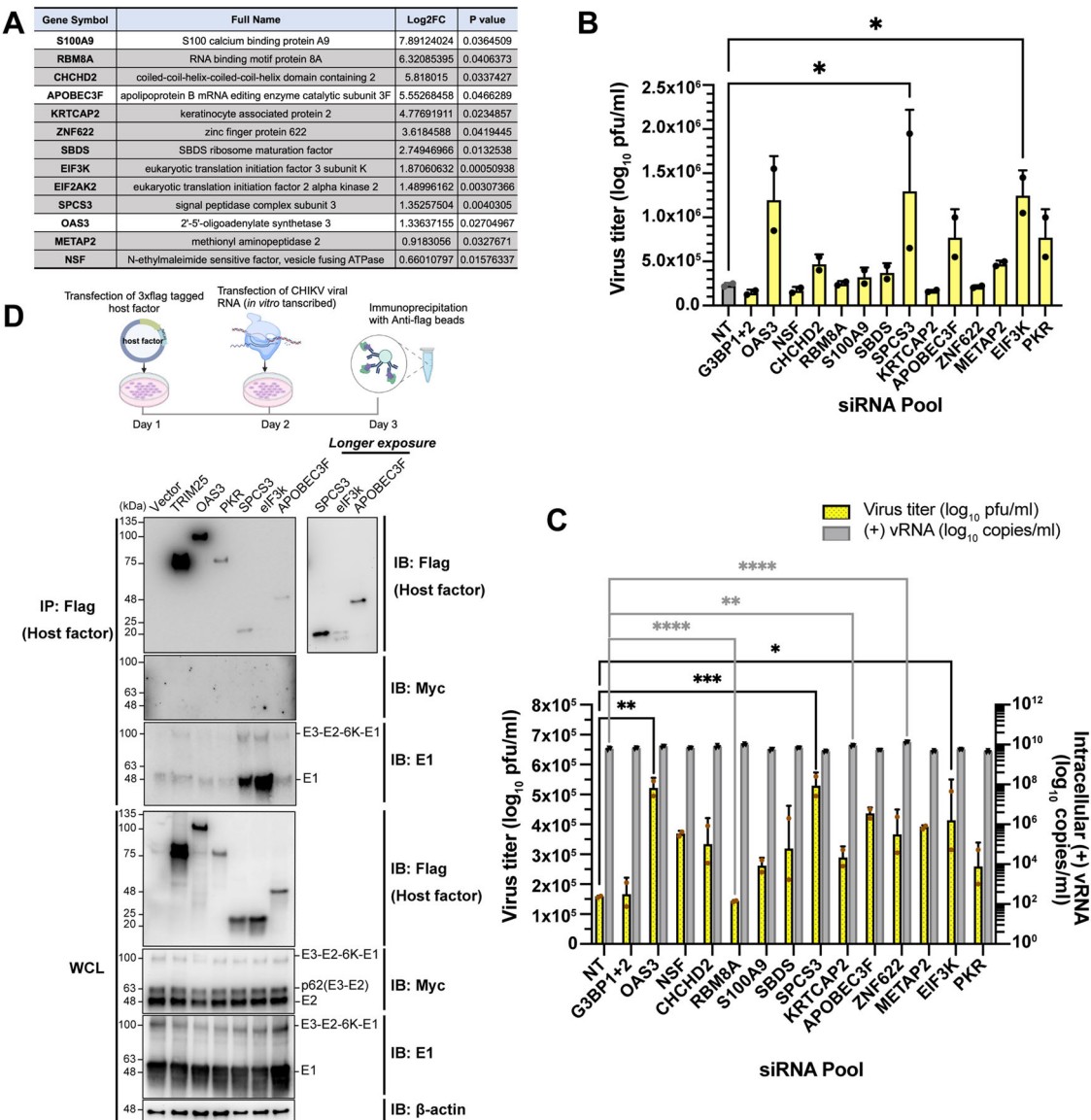

**Figure 6. CHIKV E1 interacts with macrophage host factors that block virion production.**

(A) Table of identified host factors that were chosen for siRNA knockdown assays in (B, C). Statistical analysis for protein differential expression is a moderated *t*-test from R package ArtMS3. The *P* values are adjusted with Benjamini–Hochberg for the multiple hypothesis correction. The gray-highlighted genes are significantly detected in 2 independent AP-MS experiments. (B, C) Evaluation of CHIKV infection (B) and production (C) in human macrophages with OAS3, NSF, CHCHD2, RBM8A, S100A9, SBDS, SPCS3, KRTCAP2, APOBEC3F, ZNF622, METAP2, EIF3K, or PKR knocked down. THP-1-derived macrophages were transfected with pooled siRNAs targeting specific host factors or nontargeting siRNAs (NT) for 48 h. The cells were then infected with CHIKV 181/clone 25 (MOI 5) (B) or transfected with CHIKV vRNA (C) for 24 h. The supernatant virus titers from cells treated with siRNAs targeting host factors were determined by plaque assay and compared to the titers from cells treated with NT siRNA to assess the anti- or proviral effects of specific host genes on CHIKV production. G3BP1 and G3BP2 (G3BP1 + 2) known to be proviral for CHIKV replication were knocked down together as control. For (B), data were representative of two independent experiments. The mean values of biological duplicates were plotted with SD (one-way ANOVA and Dunnett's multiple comparisons test: si-NT vs si-SPCS3 *$p = 0.031$; si-NT vs si-EIF3K *$p = 0.0421$.) For (C), data were representative of two independent experiments. The plaque assay results were plotted from biological duplicates with the mean values (one-way ANOVA and Dunnett's multiple comparisons test: viral titer of si-NT vs si-OAS3 **$p = 0.01$; viral titer of si-NT vs si-SPCS3 ***$p = 0.0008$; viral titer of si-NT vs si-EIF3K *$p = 0.0194$). The qPCR results were plotted from biological triplicates with the mean values (one-way ANOVA and Brown-Forsythe test: viral copy of si-NT vs si-RBM8A ****$p < 0.0001$; viral copy of si-NT vs si-KRTCAP2 **$p = 0.0054$; viral copy of si-NT vs si-ZNF622 ****$p < 0.0001$). (D) 293T cells were transfected with plasmids expressing 3xflag-tagged host factors (TRIM25, OAS3, PKR, SPCS3, eIF3k, and APOBEC3F) or empty vector control for 24 h and later transfected with vRNA of CHIKV/myc-E2. The cells were lysed and immunoprecipitated by anti-flag agarose beads. Immunoblot was probed to check for E2/E1 binding to these host factors. 3xflag tagged TRIM25 (tripartite motif containing 25) was transfected into 293T cells for immunoprecipitation control. Data were representative of three independent experiments. Source data are available online for this figure.

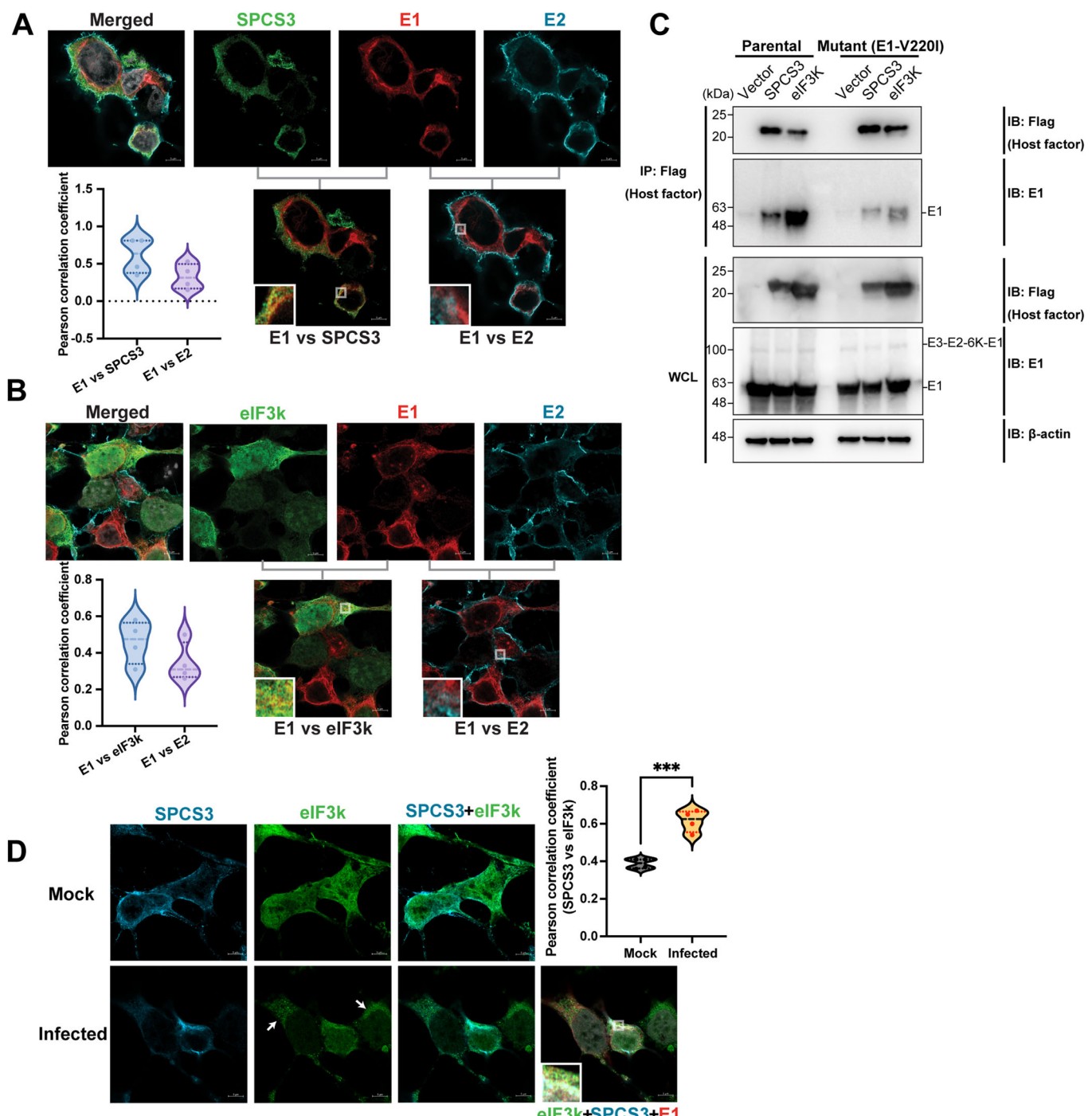

replicon where we replaced the viral structural polyprotein with EGFP into eIF3k KO 293T cells with or without overexpression of exogeneous 3xflag-tagged eIF3k (Fig. 8C). We found that restoration of eIF3k slightly reduces subgenomic promoter-driven EGFP expression while expression of the viral nonstructural protein (nsP3) is unaffected. We also transfected CHIKV vaccine strain 181/clone 25 into eIF3k KO 293T cells with or without 3xflag-tagged eIF3k expression (Fig. 8D). Again, eIF3k overexpression does not alter structural (E2 and E1) and nonstructural (nsP3) protein expression. These results suggest

that eIF3k has no impact on CHIKV subgenomic or genomic RNA translation.

To further characterize the antiviral activity of eIF3k, we dissected the involvement of eIF3k protein domains. eIF3k contains two major domains: HEAT repeat-like HAM (HEAT analogous motif) domain, a winged-helix-like WH domain, and a C terminal long tail region (Wei et al, 2004; Chen et al, 2022) (Fig. 8E). Highly conserved hydrophobic residues from four of the helices in the HAM domain and the first helix from the WH domain form a hydrophobic core between the HAM and WH domains (Wei et al,

◄

**Figure 7.    A pool of free E1 separate from E2 associates with SPCS3 and eIF3k through the positively selected site E1-V220.**

(**A**) Colocalization analysis of CHIKV E1 with E2 or with SPCS3 through immunofluorescence. 293T cells were transfected with plasmids expressing 3xflag-SPCS3 and infected with CHIKV 181/clone 25 for 24 h 1 day later. flag-tagged SPCS3 (green), and CHIKV E1 (red) and E2 (cyan) were labeled through indirect staining with primary antibodies against flag, E1, and E2. The representative colocalization regions are enlarged on the bottom left of the overlaid images. Colocalization between CHIKV E1 and SPCS3 (E1 vs SPCS3) and between CHIKV E1 and E2 (E1 vs E2) are compared through Pearson correlation analysis and shown as violin plots. Pearson correlation coefficient values range from 1 to −1, where 1 is a total positive correlation, −1 is a total negative correlation, and 0 is no correlation. Scale bar: 5 μm. Representative results from two independent are shown here. Two field images were taken for each sample in each independent experiment, and four cells from one independent experiment were designated as region of interests (ROIs) for colocalization analysis. (**B**) Colocalization analysis of CHIKV E1 with E2 or with eIF3k through immunofluorescence. 293T cells were transfected with plasmids expressing 3xflag-eIF3k and infected with CHIKV 181/clone 25 for 24 h one day later. 3xflag-tagged eIF3k (green), and CHIKV E1 (red) and E2 (cyan) were labeled as previously described. The representative colocalization regions are shown in the bottom left of the overlaid images. Colocalization between CHIKV E1 and eIF3k (E1 vs eIF3k) and between CHIKV E1 and E2 (E1 vs E2) is compared through Pearson correlation analysis (refer to 7A) and shown as violin plots. Scale bar: 5 μm. Representative results from two independent are shown here. Two field images were taken for each sample in each independent experiment, and four cells from one independent experiment were designated as region of interests (ROIs) for colocalization analysis. (**C**) 293T cells were transfected with plasmids expressing 3xflag-tagged host factors (SPCS3, eIF3k) or empty vector control for 24 h followed by transfection with a plasmid expressing parental or E1-V220I-containing CHIKV poly-glycoprotein (E3-myc-E2-6K-E1). The cells were lysed for immunoprecipitation with anti-flag agarose beads. Immunoblot was probed for parental or mutant E1 binding to host factors. Data are representative of 2 independent experiments. (**D**) Colocalization analysis of SPCS3 and eIF3k in uninfected and CHIKV-infected cells through immunofluorescence. 293T cells were co-transfected with plasmids expressing 3xflag-eIF3k and V5-SPCS3 followed by mock or CHIKV infection for 24 h 1 day later. V5-SPCS3 (cyan), 3xflag-eIF3k (green), and CHIKV E1 (red) were labeled through indirect staining with antibodies against V5, flag, and E1. The representative colocalization regions are shown on the bottom left of the overlaid images. Colocalization between SPCS3 and eIF3k (SPCS3 vs eIF3k) is compared in mock and CHIKV-infected 293T cells through Pearson correlation analysis (refer to 7A) and shown as violin plots (unpaired t-test: Mock vs Infected ***$p$ = 0.0004). Scale bar: 5 μm. Representative results from two independent are shown here. Two field images were taken for each sample in each independent experiment, and four cells from one independent experiment were designated as region of interests (ROIs) for colocalization analysis. Source data are available online for this figure.

2004). To identify the domain(s) required for the antiviral activity of eIF3k, we constructed several 3xflag-tagged truncation mutants: HAM + WH mutant, Core mutant, and HAM mutant (Fig. 8F). The HAM + WH mutant contains the two eIF3k domains without the C terminal tail. The Core mutant contains HAM and the first helix from WH domain to include the hydrophobic core structure (Wei et al, 2004) (Fig. 8F). Except for the Core truncation, the HAM + WH and HAM truncations can be well expressed in eIF3k KO cells after transient transfection (Fig. 8G). To investigate the antiviral activities of HAM and WH domains, we transfected eIF3k KO 293T cells with HAM + WH and HAM truncation mutants and later infected the cells with CHIKV. The plaque assay results from all the independent experiments showed that the HAM alone is sufficient to inhibit CHIKV production though statistically insignificant (Fig. 8H).

Contrary to the dogma that viral glycoproteins only play essential roles in entry, assembly, and egress, our study provides one of the first comprehensive evidence that CHIKV glycoproteins actively interfere with intracellular blockades for efficient virion production and spread in human macrophages.

## Discussion

Macrophages are important cellular reservoirs for persistent CHIKV infection; however, the underlying mechanisms are largely unexplored. In this study, we interrogated the CHIKV proteins that hijack macrophages to produce and spread new infectious virus particles. We first demonstrated that both CHIKV glycoproteins E2 and E1 mediate efficient virion production from infected macro-phages through comparative infection with CHIKV-ONNV chimeras. By performing evolutionary selection analysis on sequences of human CHIKV isolates from NCBI Virus (Hatcher et al, 2017), we identified E2-V135 and E1-V220 to be associated with elevated CHIKV production. We then uncovered two new host factors, SPCS3 and eIF3k, with inhibitory effects on CHIKV production that specifically interact with CHIKV E1. Unlike other

translation initiation factors involved in virus infection, the anti-CHIKV activity of eIF3k is mediated by its HAM protein domain in a translation-independent manner. Mutating the positively selected site at CHIKV E1-V220 into the ONNV homologous residue attenuates its interaction with SPCS3 and eIF3k, respectively. Our results suggest that the evolutionary selection of CHIKV glycopro-teins driven by intracellular antiviral host factors, including SPCS3 and eIF3k, contributes to efficient CHIKV production in macrophages.

According to previous studies (Brown et al, 2018; Voss et al, 2010), CHIKV E2 and E1 are always interacting with each other from single heterodimer formation in the ER to heterodimer trimerization before viral particle assembly. We found the CHIKV positively selected sites E2-V135 and E1-V220 on the exterior of a single E2-E1 heterodimer, suggesting that they may be involved in interactions with host factors, but they appear to not be engaged in the E2-E1 interaction in a single heterodimer (Fig. 4G). Interest-ingly, in trimerized spike structure, both of these residues are located at the interaction surface between two adjacent E2-E1 heterodimers (Fig. 4H) and may play a role in trimer formation. Unlike E2-V135, that is fully embedded in the center of the trimerized spike, E1-V220 is partially exposed and protruding into the groove formed by E1 and the E2 of the neighboring heterodimer, accessible to host factors. As such, mutating CHIKV E1-V220 to the ONNV residue (E1-V220I) may not only disrupt E2-E1 trimerization but also interfere with viral glycoprotein interaction with host factors. Similarly, swapping E2 or E1 with CHIKV glycoprotein in the ONNV backbone may also affect the interaction between neighboring E2 and E1 during trimerization, which may explain why neither ONNV/CHIKV E2 nor ONNV/CHIKV E1 rescues virion production in macrophages (Fig. 3B,C). Taken together, E2-V135 in a single heterodimer and E1-V220 in a single or trimerized heterodimer are all likely to interface with intracellular restriction factors in macrophages, driving positive selection at these sites to increase viral fitness and production.

While we found no impact of other positively selected sites on CHIKV production, it is possible that they are involved in adaptive

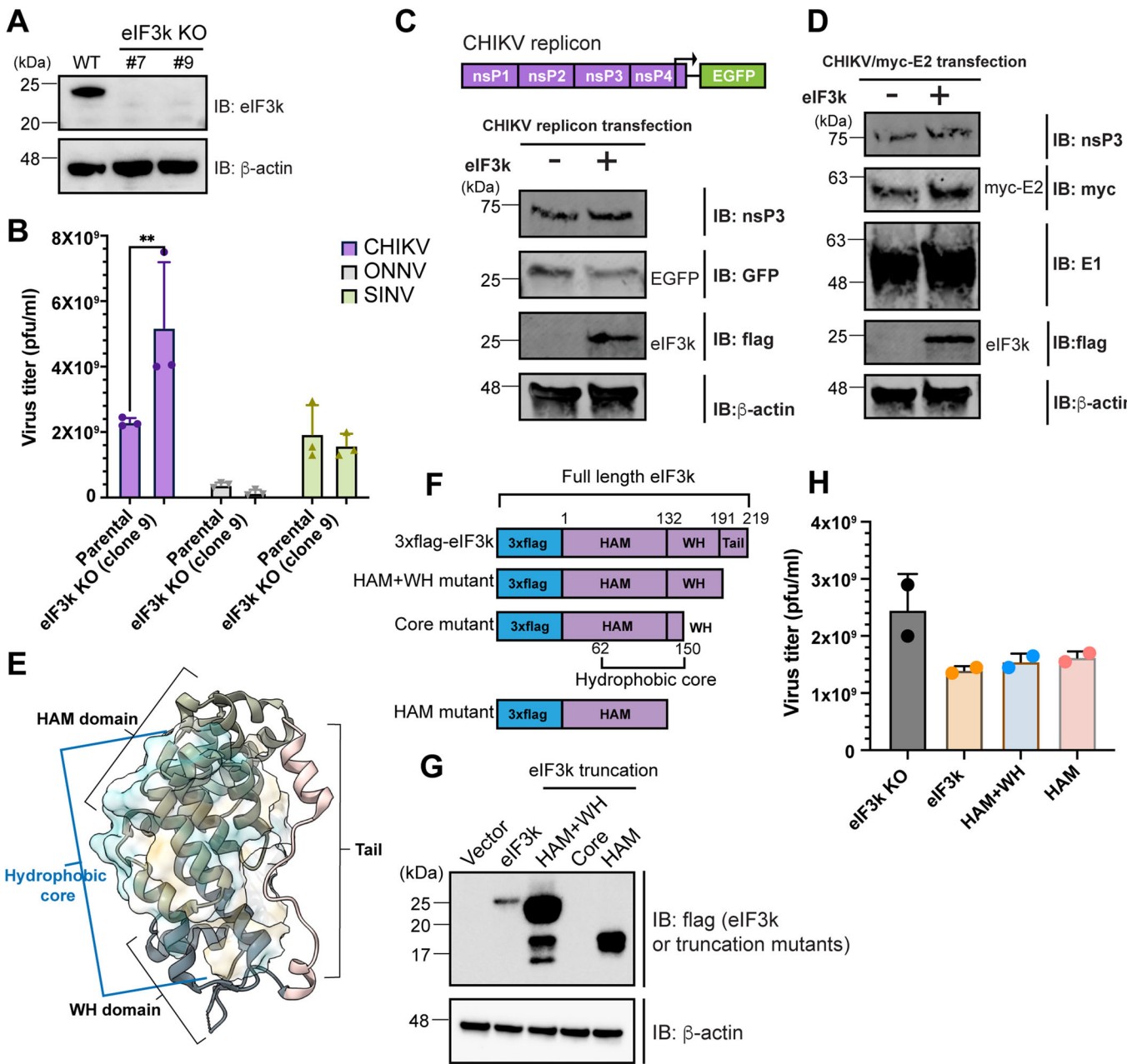

immune response, given that they are evolutionarily selected in the viral glycoproteins. Components of the adaptive immune response, such as virus-specific antibodies and T cells, can also select for escape mutations in viral glycoproteins (Tenthorey et al, 2022). We looked into the epitopes of currently characterized human CHIKV-neutralizing antibodies or broadly anti-alphavirus antibodies. They are mainly mapped to E2 domains A and B, responsible for receptor binding and cellular attachment, and E1 domain II, proximal to or within the fusion loop (Kim and Diamond, 2022; Kim et al, 2021; Pal et al, 2013). None of these reported antibodies target the six differential selection sites in CHIKV 181/clone 25 (Fig. EV2B, amino acids in red), suggesting that these residues are more likely selected by intracellular host restriction factors.

Among the candidate CHIKV glycoprotein interactors we identified, SPCS3 and eIF3k have inhibitory activities against CHIKV production in THP-1-derived macrophages (Fig. 6B,C). Surprisingly, the interaction with SPCS3 and eIF3k only engages CHIKV E1 but not E2. In infected cells, E1 is mostly localized to the cytoplasmic region adjacent to the nucleus, likely the ER, while E2 is predominantly found on the plasma membrane. The distinct localization patterns of E2 and E1 challenge previous knowledge that E2 and E1 always act together in heterodimer forms. Importantly, a previous study showed that alphavirus nsP3 (Götte et al, 2018), thought to always associate with membrane-bound viral replication complexes, can be freed to form large cytoplasmic aggregates. These findings support a model where a separate pool of

◄

**Figure 8. The specific anti-CHIKV activity of eIF3k is translation-independent and mediated by its HAM domain.**

(A) Immunoblot validation of *EIF3K* CRISPR KO in 293T clones 7 and 9. (B) eIF3k KO 293T cells (clone 9) were infected with CHIKV 181/clone 25, ONNV SG650, or SINV Toto1101 at MOI 1 for 24 h. Virion production was evaluated by titering the supernatant infectious particles through plaque assay. Data were representative results of three independent experiments. The mean values of biological triplicates were plotted with SD (two-way ANOVA and Šidák's multiple comparisons test: Parental vs eIF3k KO for CHIKV infection **$p = 0.0066$). (C) Translation of CHIKV replicon in eIF3k KO 293T cells (clone 7) with or without eIF3k reconstitution. The schematic for the CHIKV replicon is shown on top. Viral structural polyprotein downstream of the subgenomic promoter is replaced with the EGFP reporter. The eIF3k KO 293T cells were first transfected with an empty vector or plasmid expressing 3xflag-eIF3k followed by transfection with the CHIKV replicon RNA one day later. Twenty-four hours after the second transfection, protein expression of nsP3, EGFP, and 3xflag-eIF3k was detected by immunoblotting. As a GFP variant, EGFP was detected by a GFP antibody. Data were representative of three independent experiments. (D) The translation of CHIKV/myc-E2 in eIF3k KO 293T cells (clone 7) with or without eIF3k reconstitution. As mentioned in Fig. 5A, the myc tag is inserted at the N-terminal end of E2. The eIF3k KO 293T cells were first transfected with an empty vector or plasmid expressing 3xflag-eIF3k followed by transfection with CHIKV/myc-E2 RNA one day later. Protein expression of nsP3, myc (E2), E1, and (flag) eIF3k was detected by immunoblotting 24 h after the second transfection. Data were representative of three independent experiments. (E) The structure of human eIF3k with the protein domains labeled. The eIF3k crystal structure is downloaded from PDB (1RZ4)(Wei et al, 2004) and visualized in Chimera X. eIF3k consists of a HAM domain (khaki), WH domain (blue), and a long C-terminal tail region with α-helix at both ends (pink). The HAM domain contains a leading α-helix and 3 HEAT analogous repeats followed by a short helix. The WH domain contains three α-helices and three β-strands. (F) The diagram of eIF3k truncation mutants. The HAM + WH mutant that lacks the C-terminal tail terminates after residue S191 of full-length eIF3k. The Core mutant with a truncated WH domain terminates after residue Y150 of full-length eIF3k and includes the hydrophobic core formed by the highly conserved hydrophobic residues from HAM and the first helix of WH. The HAM-only mutant terminates after residue T132 of full-length eIF3k. All the eIF3k truncation mutants are tagged with an N-terminal 3xflag. (G) Validation of expression of eIF3k truncation mutants in eIF3k KO 293 T cells through immunoblotting. Since the Core mutant cannot be expressed, it is not followed up in (8H). (H) The anti-CHIKV activities of eIF3k truncation mutants. The eIF3k KO 293T cells (clone 7) were transfected with plasmids expressing full-length eIF3k or different truncation mutants. The cells were then infected with CHIKV at MOI 1 for 24 h 1 day following transfection. Levels of infectious particle production in supernatant samples were determined by plaque assay on BHK-21 cells. The incubation period for plaque assay is 28 h. Data were representative of five independent experiments. The mean values of biological duplicates were plotted with SD. Source data are available online for this figure.

free E1 interferes with cytoplasmic host restriction factors, further highlighting the enigmatic roles of alphavirus proteins in the viral life cycle.

To build on that model. we showed that mutation of the positively selected site in E1 (E1-V220) completely abrogates virion production in THP-1 derived macrophages and reduces E1 interaction with SPCS3 and eIF3k (Figs. 4D and 7C). These results clearly demonstrate that E1-V220 is a critical interaction site that has been evolutionarily selected by multiple restriction factors, including SPCS3 and eIF3k, to drive increased virion production in macrophages. To better elucidate the mechanism of these macrophage restriction factors, further studies need to be performed to determine the functional consequences of E1 binding to these anti-CHIKV factors, such as protein degradation or sequestration.

SPCS3 is one of the core components of the endoplasmic reticulum-associated signal peptidase complex (SPC) (Gemmer and Förster, 2020), which cleaves signal peptides during the translocation of protein precursors in the ER (Böhni et al, 1988, 11; Shelness et al, 1993). The signal peptidases are presumably usurped by flaviviruses, bunyaviruses, and alphaviruses for poly-glycoprotein cleavage (Zhang et al, 2016; Neufeldt et al, 2018; Zimmerman et al, 2023). However, it is unknown what exact peptidase releases p62, 6K, and E1 from the alphavirus poly-glycoprotein precursor (Frolov et al, 1996). A previous genome-wide CRISPR KO screen uncovered both SPCS1 and SPCS3 as proviral factors for flavivirus infection, and depletion of SPCS1 led to inefficient polyprotein cleavage disrupting flavivirus production (Zhang et al, 2016). Unexpectedly, we found that SPCS3 exhibits anti-CHIKV activity and strongly associates with CHIKV E1. SPCS3 overexpression does not affect CHIKV poly-glycoprotein cleavage (Fig. 6D), suggesting novel peptidase-independent antiviral activities. For the first time, we demonstrated the functional dualities of SPC proteins in different virus infection systems.

On the other hand, eIF3k is a subunit of the eukaryotic translation initiation factor 3 (eIF3) complex, which is the most complex and least characterized among the mammalian translation initiation factors containing at least 12 subunits (eIF3a-m)(Gomes-Duarte et al, 2017). eIF3 binds the small ribosomal subunit (40 S) and is involved in almost all steps of translation initiation (Wei et al, 2004; Aitken et al, 2016). Not as an essential component of the eIF3 complex, eIF3k is located on the outside of the eIF3 structure and can be easily dissociated from the complex (Gomes-Duarte et al, 2017).

The role of eIF3k in a viral context has not been explored previously. The most well-known example of translation shutoff as a general antiviral mechanism is mediated by protein kinase R (PKR). PKR, also known as eukaryotic translation initiation factor 2 alpha kinase 2 (EIF2AK2), senses double-stranded vRNAs in the cytoplasm leading to eIF2α phosphorylation and suppression of viral and host gene expression (Fros and Pijlman, 2016). Interestingly, PKR and other eIF3 subunits (eIF3h, eIF3j, eIF3m) were also identified in our AP-MS results (Figs. 5C, 6A). Therefore, it led us to hypothesize that the specific anti-CHIKV activity of eIF3k might involve viral translation inhibition. However, we showed that eIF3k neither inhibits CHIKV nonstructural nor structural protein translation, suggesting that eIF3k antiviral activity is not translation dependent (Fig. 8C,D). Although eIF3k is normally known for its role in translation initiation in the cytoplasm, it also interacts with promyelocytic leukemia protein (PML) and is associated with PML nuclear bodies in the nucleus (Salsman et al, 2013). Our results confirmed the strong nuclear localization of eIF3k (Fig. 7B,D mock infection) and revealed the translocation of eIF3k from nucleus to cytosol induced by CHIKV infection. Notably, eIF4E, which also has nuclear localization, was previously reported to mediate nuclear-cytoplasmic export of select transcripts (Osborne and Borden, 2015). It will be interesting to determine in future studies whether eIF3k also affects RNA export leading to modulation of the host antiviral response. Meanwhile, we also found that the anti-CHIKV activity of eIF3k potentially lie in the HAM domain (Fig. 8H). Previous structure analysis demonstrated that the eIF3k HAM domain consists of three HEAT analogous which can provide an interaction surface for protein–protein interaction (Wei et al, 2004). Further investigations are required to elucidate whether the eIF3k HAM domain recruits other antiviral host proteins to mediate anti-CHIKV activity.

Finally, although macrophages are widely recognized as persistent CHIKV reservoirs, most of the evidence came from the detection of viral components in nonhuman primates or patient samples. Here we used interdisciplinary approaches to uncover the advantage conferred by CHIKV glycoproteins in virion production in an in vitro macrophage model system. Future validations with ONNV-CHIKV chimeric virus infection in mammalian hosts will benefit the mechanistic understanding of how CHIKV glycoproteins facilitate virus dissemination through infected macrophages in a more physiologically relevant environment. Especially, while macrophages in vivo comprise heterogenous cell subsets, including both monocyte-derived macrophages and tissue-resident macrophages, the in vitro THP-1 derived macrophage model only represents monocyte-derived macrophages. Using in vivo models will address the question of whether resident macrophages, especially Langerhans cells from the skin and synovial macrophages from the joints, also exhibit greater susceptibility to CHIKV.

In summary, our study has unraveled a novel role of CHIKV glycoproteins in virion production in macrophages that is driven by an evolutionary arms race with intracellular antiviral factors, SPCS3 and eIF3k. Overall, this research not only challenges the prevailing paradigm that viral glycoproteins mainly play a role in entry, but also provides promising targets for therapeutic intervention to strengthen the antiviral status of macrophages in order to eliminate CHIKV reservoirs.

# Methods

### Reagents and tools table

| Reagent/Resource | Reference or source | Identifier or catalog number |
| --- | --- | --- |
| **Experimental Models** | | |
| THP-1 cells (*H. sapiens*) | ATCC | Cat#TIB-202 |
| Human primary PBMCs | UCLA/CFAR Virology Core Lab | N/A |
| HEK-293T cells (*H. sapiens*) | ATCC | Cat#CRL-3216 |
| BHK-21 cells (*Mesocricetus auratus*) | ATCC | Cat#CCL-10 |
| Lenti-X 293 T cells | Takara | Cat#631294 |
| **Recombinant DNA** | | |
| CHIKV 181/clone 25 | Scott Weaver, The University of Texas Medical Branch at Galveston (Gorchakov et al, 2012) | GenBank: AAA53256.3 |
| CHIKV 181/clone 25-EGFP | Nguyen et al, 2023 | N/A |
| pONN.AP3 | Stephen Higgs, Kansas State University (Brault et al, 2004) | GenBank: AF079456 |
| p5'dsONNic-foy | Stephen Higgs, Kansas State University (Brault et al, 2004) | N/A |
| SINV Toto1101 | Charles M. Rice, The Rockefeller University (Rice et al, 1987) | N/A |
| SINV TE/5'2 J/GFP | Charles M. Rice, The Rockefeller University (Pierro et al, 2003) | N/A |
| RRV T48-EGFP | Mark Heise, The University of North Carolina at Chapel Hill (Morrison et al, 2006) | N/A |

| Reagent/Resource | Reference or source | Identifier or catalog number |
| --- | --- | --- |
| CHIKV LR2006 OPY1 | Stephen Higgs, Kansas State University (Tsetsarkin et al, 2006) | GenBank: DQ443544 |
| CHIKV AF15561 | Scott Weaver, The University of Texas Medical Branch at Galveston (Gorchakov et al, 2012) | GenBank: EF452493 |
| Chimera I | This study | N/A |
| Chimera II | This study | N/A |
| Chimera III | This study | N/A |
| Chimera IV | This study | N/A |
| Chimera III-I | This study | N/A |
| Chimera III-II | This study | N/A |
| Chimera III-III | This study | N/A |
| ONNV/CHIKV E2 | This study | N/A |
| ONNV/CHIKV E1 | This study | N/A |
| ONNV/CHIKV E2 + E1 | This study | N/A |
| E2-I + E1 | This study | N/A |
| E2-II + E1 | This study | N/A |
| E2 + E1-I | This study | N/A |
| E2 + E1-II | This study | N/A |
| CHIKV/myc-E2 | This study | N/A |
| CHIKV-EGFP replicon | This study | N/A |
| pcDNA-3xflag-SPCS3 | This study | N/A |
| pcDNA-3xflag-EIF3K | This study | N/A |
| pcDNA-3xflag-APOBEC3F | This study | N/A |
| pcDNA-3xflag-PKR | This study | N/A |
| pcDNA-3xflag-OAS3 | This study | N/A |
| pcDNA-E3-myc-E2-6k-E1 | This study | N/A |
| pcDNA-3xflag-HAM | This study | N/A |
| pcDNA-3xflag-HAM + WH | This study | N/A |
| pcDNA-3xflag-Core | This study | N/A |
| CHIKV E2-V135L | This study | N/A |
| CHIKV E2-A164T | This study | N/A |
| CHIKV E2-A246S | This study | N/A |
| CHIKV E1-E211K | This study | N/A |
| CHIKV E1-V220I | This study | N/A |
| CHIKV E1-R366K | This study | N/A |
| lentiCRISPRv2 puro | Addgene | Cat#98290 |
| pMD2.G | Addgene | Cat#12259 |
| psPAX2 | Addgene | Cat#12260 |
| **Antibodies** | | |
| Anti-myc, mouse monoclonal | Cell Signaling Technology | Cat#2276 S |
| Anti-myc, rabbit monoclonal | Cell Signaling Technology | Cat#2272 S |
| Anti-flag, mouse monoclonal | Sigma-Aldrich | Cat#F1084 |
| Anti-flag, rabbit monoclonal | Cell Signaling Technology | Cat#14793 S |

| Reagent/Resource | Reference or source | Identifier or catalog number |
|---|---|---|
| Anti-CHIKV E1, rabbit polyclonal | GeneTex | Cat#GTX135187 |
| ChromoTek GFP antibody, rabbit polyclonal | Proteintech | Cat#pabg1 |
| Anti-β-actin-HRP, mouse monoclonal | Sigma-Aldrich | Cat#A3854 |
| Goat-anti-mouse HRP | Jackson ImmunoResearch | Cat#115-035-146 |
| Goat-anti-rabbit HRP | Thermo Fisher Scientific | Cat#31462 |
| Anti-V5, mouse monoclonal | Millipore Sigma | Cat#V8012 |
| Anti-CHIKV E2, mouse monoclonal | BEI Resources | Cat#NR-44002 |
| Anti-flag, Alexa Fluor 488, rat monoclonal | Invitrogen | Cat#MA1-142-A488 |
| Goat-anti-rabbit Alexa Flour 594 | Invitrogen | Cat#A-11012 |
| Goat-anti-mouse Cy5 | Invitrogen | Cat#A10524 |
| **Oligonucleotides and other sequence-based reagents** | | |
| Primers for overlap & normal PCR | This study | Dataset EV2A |
| Primers for NEBuilder HiFi assembly | This study | Dataset EV2B |
| Primers for CHIKV mutants | This study | Dataset EV2C |
| qPCR primers | This study | Dataset EV2D |
| siRNAs | Thermo Fisher Scientific | Ambion Silencer siRNA |
| **Chemicals, Enzymes and other reagents** | | |
| MEM | Gibco | Cat#11095098 |
| DMEM, high glucose | Gibco | Cat#11965092 |
| RPMI 1640 (ATCC modification) | Gibco | Cat#A1049101 |
| FBS | VWR | Cat#89510 |
| Penicillin/streptomycin | Fisher Scientific | Cat#SV30010 |
| Non-essential amino acids | Gibco | Cat#11140050 |
| β-mercaptoethanol | Sigma-Aldrich | Cat#M3148 |
| DPBS | HyClone | Cat#SH30378 |
| Human AB serum | Omega Scientific | Cat#HS-20 |
| PMA | Sigma-Aldrich | Cat#P1585 |
| RosetteSep Human Monocyte Enrichment Cocktail | STEMCELL Technologies | Cat#15068 |
| Human recombinant M-CSF | STEMCELL Technologies | Cat#78057 |
| ImmunoCult-SF Macrophage Medium | STEMCELL Technologies | Cat#10961 |
| ACCUMAX | STEMCELL Technologies | Cat#07921 |
| TranIT-X2 transfection kit | Mirus Bio | Cat#MIR6004 |
| TransIT-mRNA transfection kit | Mirus Bio | Cat#MIR2225 |
| X-tremeGENE9 | Roche | Cat# 6365787001 |

| Reagent/Resource | Reference or source | Identifier or catalog number |
|---|---|---|
| Complete EDTA-free protease inhibitor mixture tablet | Roche | Cat# 11873580001 |
| FLI-06 | MCE | Cat# HY-15860 |
| Golgicide A | MCE | Cat# HY-100540 |
| NEBuilder HiFi DNA assembly kit | NEB | Cat#E5520 |
| Q5 site-directed Mutagenesis Kit | NEB | Cat# E0552S |
| Protoscript II First Strand cDNA Synthesis Kit | NEB | Cat# E6560L |
| TRIzol reagent | Thermo Fisher Scientific | Cat#E6560L |
| Direct-zol RNA Microprep Kit | Zymo Research | Cat#R2060 |
| MAXIscript SP6/T7 Transcription Kit | Thermo Fisher Scientific | Cat#AM1320 |
| Luna qPCR Dye | NEB | Cat#E6560 |
| Luna Universal Probe qPCR Master Mix | NEB | Cat#M3004 |
| PrimeTime One-Step RT-qPCR master mix | IDT | Cat# 10007065 |
| Nonidet P 40 Substitute (NP40) | VWR | Cat#M158 |
| EZview Red Anti-c-Myc Affinity Gel | Sigma-Aldrich | Cat#E6654 |
| EZview Red Anti-flag M2 Affinity Gel | Sigma-Aldrich | Cat#F2426 |
| Dynabeads Protein G | Invitrogen | Cat#10004D |
| 4–15% precast Mini-PROTEAN TGX Gel | Bio-Rad | Cat#4561086 |
| Laemmli Sample Buffer | Bio-Rad | Cat# 1610747 |
| Trans-Blot Turbo RTA Midi 0.2 μm PVDF Transfer Kit | Bio-Rad | Cat# 1704273 |
| ProSignal Pico ECL Reagents | Genesee Scientific | Cat#20-300B |
| Collagen-coated coverslips | Corning BioCoat | Cat#354089 |
| Formaldehyde (37% W/M) | Fisher Scientific | Cat# BP531-500 |
| Triton-X100 | Sigma-Aldrich | Cat# T8787 |
| Glycine | Sigma-Aldrich | Cat# 50046 |
| PpuMI | NEB | Cat#R0506 |
| NotI-HF | NEB | Cat#R3189 |
| ApaI | NEB | Cat#R0114 |
| PspXI | NEB | Cat#R0656 |
| BamHI-HF | NEB | Cat#R3136 |
| MfeI-HF | NEB | Cat#R3589 |
| EcoRI-HF | NEB | Cat#R3101 |
| NdeI | NEB | Cat#R0111 |
| SpeI-HF | NEB | R3133 |
| XbaI | NEB | R0145 |
| NheI-HF | NEB | R3131 |
| SacI-HF | NEB | R3156 |

| Reagent/Resource | Reference or source | Identifier or catalog number |
|---|---|---|
| BspEI | NEB | R0540 |
| EcoRV-HF | NEB | R3195 |
| **Software** | | |
| Graphpad Prism v9 | https://www.graphpad.com | N/A |
| FlowJo v10 | https://www.flowjo.com | N/A |
| ImageJ2 v2.14.0 | https://imagej.net | N/A |
| R Studio v2023.09 | https://posit.co/products/open-source/rstudio/ | N/A |
| Cytoscape v3.9 | https://cytoscape.org/ | N/A |
| **Other** | | |
| CFX96 OPUS | Bio-Rad | N/A |
| ChemiDoc | Bio-Rad | N/A |
| Dionex Ultimate 3000 UHPLC | Thermo Fisher Scientific | N/A |
| Nimbus electrospray ionization source | Phoenix S&T | N/A |
| Orbitrap Fusion Lumos Tribrid mass spectrometer | Thermo Fisher Scientific | N/A |
| ZEISS LSM 880 with (Airyscan) | ZEISS | N/A |
| MACSQuant Analyzer | Miltenyi Biotec | N/A |
| R package: ggplot2 | https://ggplot2.tidyverse.org/ | N/A |
| R package: ClusterProfile | Wu et al, 2021 | N/A |
| R package: ggmsa | Zhou et al, 2022 | N/A |
| R package: Biostring | https://rdrr.io/bioc/Biostrings/ | N/A |
| R package: ArtMS3 | https://bioconductor.org/packages/release/bioc/html/artMS.html | N/A |
| Cytoscape plugin: EnrichmentMap | Merico et al, 2010 | N/A |
| Image J plugin: Coloc 2 | https://imagej.net/plugins/coloc-2 | N/A |
| CRAPome database | crapome.org | N/A |
| CORUM database | Tsitsiridis et al, 2023 | N/A |
| DAVID database | david.ncifcrf.gov | N/A |
| KEGG database | kegg.jp | N/A |
| NCBI Virus database | https://www.ncbi.nlm.nih.gov/labs/virus/vssi/#/ (Hatcher et al, 2017) | N/A |
| MUSCLE v3.8.31 | https://drive5.com/muscle/ | N/A |
| HyPhy | https://hyphy.org/ | N/A |
| IQ-Tree v1.6.12 ModelFinder | http://iqtree.cibiv.univie.ac.at (Minh et al, 2020; Trifinopoulos et al, 2016) | N/A |

## Methods and Protocols

### Cell culture, viruses, and infections

BHK-21 cells (American Type Culture Collection (ATCC)) were maintained in Minimum Essential Media (MEM, Gibco) supplemented with 7.5% fetal bovine serum (FBS, VWR). HEK-293T cells (ATCC) were maintained in Dulbecco's Modified Eagle Medium (DMEM, VWR) supplemented with 10% FBS. THP-1 human monocytes (ATCC) were maintained in Roswell Park Memorial Institute 1640 Medium (RPMI 1640, Gibco) supplemented with 10% FBS, 1X penicillin/streptomycin (P/S, Fisher Scientific), 1X non-essential amino acids (NEAA, Gibco), and 0.05 mM β-mercaptoethanol (Sigma-Aldrich).

The infectious clone plasmids of enhanced GFP (EGFP)-expressing or unlabeled CHIKV vaccine strain 181/clone 25, EGFP-expressing (p5′dsONNic-foy) or unlabeled (pONN.AP3) ONNV strain SG650, EGFP-expressing SINV (TE/5′2 J/GFP) or unlabeled (pToto1101) SINV, and EGFP-expressing RRV (strain T48) have been previously reported(Pierro et al, 2003; Brault et al, 2004; Gorchakov et al, 2012; Morrison et al, 2006; Kuhn et al, 1991; Rice et al, 1987; Nguyen et al, 2023). The EGFP-expressing CHIKV, ONNV, and SINV have a 5′ duplicated subgenomic promoter that controls EGFP expression, while the EGFP-expressing RRV has a 3′ duplicated subgenomic promoter that controls EGFP expression. The infectious clone plasmids of pathogenic CHIKV La Réunion strain (LR2006 OPY1) (Tsetsarkin et al, 2006) and Asian strain (AF15561)(Gorchakov et al, 2012) are kind gifts from Stephen Higgs (Kansas State University) and Scott Weaver (The University of Texas Medical Branch at Galveston), respectively. Propagations and titrations of virus stocks were generated in BHK-21 cells as previously described (Yang et al, 2022; Luu et al, 2021). The pathogenic CHIKV stocks were prepared and titrated in a biosafety level 3 lab. To infect THP-1-derived macrophages or primary monocyte-derived macrophages, viruses were diluted in Dulbecco's phosphate buffered saline (DPBS) supplemented with 1% human AB serum (Omega Scientific) and 1% P/S, and added to cells at a multiplicity of infection (MOI) of 5 plaque-forming units (pfu)/cell. Typically, infection was carried out in a 12-well or 24-well plate with $5 \times 10^5$ or $2.5 \times 10^5$ macrophages seeded per well. Cells were incubated with the virus for 1 h and washed twice with PBS to remove the virus. Freshly made media was then added to cells, and supernatant samples were collected at the indicated timepoints for plaque assay as previously described.

### Monocyte differentiation and transfection

THP-1 human monocytes were differentiated into macrophages through a 24-h stimulation with 50 ng/ml phorbol 12-myristate 13-acetate (PMA, Sigma-Aldrich) in RPMI 1640 supplemented with 10% human AB serum, 1X NEAA, 1X P/S followed by a 24 h rest in human-serum containing RPMI 1640.

Human primary peripheral blood mononuclear cells (PBMCs) were obtained from donors through the UCLA/CFAR Virology Core Lab. The RosetteSep™ Human Monocyte Enrichment Cocktail (STEMCELL Technologies) was used to purify monocytes from the PBMCs. To differentiate the purified monocytes from macrophages, the monocytes were cultured in ImmunoCult™-SF Macrophage Medium (STEMCELL Technologies) supplemented with 50 ng/ml Human Recombinant M-CSF (STEMCELL Technologies) for 4 days. After differentiation, the macrophages were infected as described in the previous section.

### Generation of EIF3K Cas9-CRISPR KO clones

The designed guide RNAs (gRNAs) target exons 3 and 7 of *EIF3K* (Appendix Fig. S1A): sgRNA1: 5′-GTGCAAGTGCATGATC-GACC-3′; sgRNA2: 5′-GAAGATCTGCCCCGACTCGT-3′. The gRNAs were ligated into lentiCRISPRv2 puro vector (Addgene,

#98290). Lenti-X 293T cells (Takara) were transfected with lentiCRISPRv2, pMD2.G (Addgene, # 12259), and psPAX2 (Addgene, #12260) to generate CRISPR/Cas9 lentiviruses. 293T cells were transduced with lentiviruses and selected with 1 µg/ml puromycin for 5 days. The surviving cells were seeded at the density of 0.3 cell/well in a 96-well plate and expanded in DMEM supplemented with 10% FBS and 0.1 µg/ml puromycin. Clones 7 and 9 were verified by genomic DNA sequencing (Appendix Fig. S1B) and immunoblotting (Fig. 8A), and chosen for validation studies.

### siRNA and viral RNA transfection

For gene silencing, three unique Ambion Silencer siRNAs (Thermo Fisher Scientific) targeting 13 host factors identified by AP-MS were pooled and transfected into THP-1 macrophages at a final concentration of 25 nM. To simultaneously knock down G3BP1 and G3BP2 as a positive control, two unique Ambion Silencer siRNAs, respectively, targeting G3BP1 and G3BP2 were pooled (25 nM) and transfected into THP-1 macrophage. The same amount of nontargeting siRNA (Thermo Fisher Scientific) was transfected into THP-1 macrophages as negative control. siRNA transfections were performed with TransIT-X2 Transfection Kit (Mirus Bio) following the manufacturer's instructions. Downstream assays were conducted 48 h after transfection.

To observe viral production in transfected macrophages, 500 ng of viral genomic RNA was transfected per well in 12-well plates through the TransIT®-mRNA Transfection Kit (Mirus Bio) following manufacturer's instructions.

### Inhibition of secretory pathways

The secretory inhibitors FLI-06 and Golgicide A were purchased from MedChemExpress (MCE). The THP-1-derived macrophages were pretreated with 10 µM FLI-06 or 10 µM Golgicide A in RPMI 1640 containing 10% human serum for 30 min. The macrophages were then inoculated with ONNV, CHIKV, and ONNV/CHIKV E2 + E1 at MOI of 5 in DPBS containing 1% human AB serum for 1 h. After two washes with DPBS, the macrophages were again cultured in RPMI 1640 supplemented with 10% human AB serum and 10 mM secretory inhibitors. The macrophage supernatants were collected for plaque assay titration 24 h post infection.

### Construction of CHIKV-ONNV chimeras, positively selected site mutants, myc-tagged CHIKV, and CHIKV-EGFP replicon

All the primers and restriction sites used in chimeras, mutants, reporter virus, and replicon constructions mentioned below are listed in Tables A, B, and C in Dataset EV2, respectively.

To construct Chimera I, gene regions amplified from the parental CHIKV vaccine strain 181/clone 25 and ONNV SG650 strains were fused into two chimeric fragments, Fragment 1 and Fragment 2, through PCR overlap extension (Appendix Fig. S2). Fragment 1 was inserted into the CHIKV 181/clone 25 backbone to generate an intermediate chimera with parts of nsP4 and capsid from ONNV. The fragment from the ONNV subgenomic promoter to the end of the CHIKV poly(A) tail was digested from the intermediate chimera and inserted into the ONNV backbone with Fragment 2 to obtain Chimera I.

To generate Chimera III, we first used overlapping PCR to generate Fragment 3 to replace the equivalent region in CHIKV 181/clone 25 to obtain the CHIKV/ONNV 5'UTR backbone. We

then used the NEBuilder HiFi DNA Assembly Kit (New England Biolabs, NEB) to ligate the CHIKV/ONNV 5'UTR backbone with CHIKV subgenomic promoter and capsid (Fragment 4) and ONNV E3 to the end of the poly(A) tail (Fragment 5) (Appendix Fig. S2). Both Fragments 4 and 5 contained overlapping overhangs for HiFi ligation.

The cloning of Chimera II was based on Chimera I. We amplified the region from the CHIKV subgenomic promoter to the PspXI site in E2 with overlapping overhangs and used the NEBuilder HiFi DNA Assembly Kit to ligate the amplified product to the digested Chimera I backbone. To generate Chimera IV, we amplified the region from the ONNV subgenomic promoter to the intrinsic BamHI site in ONNV E2. We then used T4 ligase (NEB) to ligate the amplified fragment with a digested Chimera III backbone.

The other chimera clone plasmids (Chimera III-I, III-II, III-III, ONNV/CHIKV E1, ONNV/CHIKV E2, ONNV/CHIKV E2 + E1, E2-I + E1, E2-II + E1, E2 + E1-I, and E2 + E1-II) were generated in a similar fashion through multiple fragment ligations with the NEBuilder HiFi DNA Assembly Kit.

To construct the CHIKV positively selected site mutants (V135L, A164T, A246S, E211K, V220I, and R366 K), the region containing E2 or E1 was amplified from CHIKV 181/clone 25 and inserted into pCR-Blunt II-TOPO vector (Thermo Fisher Scientific) according to manufacturer's instructions. Corresponding site-directed mutagenesis was conducted on the intermediate TOPO constructs with specific mutation primers by using the Site-Directed Mutagenesis Kit (NEB). The mutated E2- or E1-containing fragments were digested from the TOPO constructs through intrinsic viral restriction sites and inserted back into CHIKV through T4 ligation.

To construct CHIKV with myc-tagged E2 (CHIKV/myc-E2), the myc tag was inserted between E3 and E2 through the NEBuilder HiFi DNA Assembly Kit. Fragment 6 was amplified from parental CHIKV 181/clone 25, containing the region from the subgenomic promoter in nsp4 to the end of E3. The segment of E2, from the start of E2 to the second NdeI site, was amplified from CHIKV 181/clone 25 as Fragment 7. The reverse primer of Fragment 6 and forward primer of Fragment 7 incorporates the myc tag into CHIKV 181/clone 25 through three-fragment assembly (Appendix Fig. S2).

To construct the CHIKV-EGFP replicon in which the structural genes in the genome of CHIKV vaccine strain 181/clone 25 were replaced with EGFP, the second subgenomic promoter, downstream structural genes, 3'UTR region, and polyA tail were removed from EGFP-CHIKV infectious clone plasmid through digestion at SpeI and NotI sites. The 3'UTR region and polyA tail were amplified from the CHIKV vaccine strain 181/clone 25 genome and reintroduced into the digested EGFP-CHIKV infectious clone plasmid through two-fragment assembly with NEBuilder HiFi DNA Assembly Kit.

### Construction of host factor and CHIKV structural polyprotein plasmids

All the primers and restriction sites used in the construction of host factor and CHIKV structural polyprotein plasmids are listed in Table B in Dataset EV2, respectively.

The cellular mRNA from THP-1 cells was reverse transcribed with oligo-dT primer through the Protoscript II First Strand cDNA

Synthesis Kit (NEB) after TRIzol (Thermo Fisher Scientific) extraction. The host genes OAS3, PKR, SPCS3, EIF3K, and APOBEC3F were amplified with specific primers containing regions overlapping the pcDNA3.1-3xflag vector. The cDNAs of host factors were then incorporated into the NotI and XbaI sites of pcDNA3.1-3xflag vector through NEBuilder HiFi DNA Assembly Kit to transiently express N-terminally 3xflag-tagged host factors.

To construct the plasmid for CHIKV structural glycoprotein following capsid cleavage (pcDNA3.1-E3-myc-E2-6K-E1), the sequence spanning the beginning of E3 to the end of E1 was amplified from CHIKV/myc-E2 with primers containing overlapping regions with the pcDNA3.1 vector and incorporated into pcDNA3.1 through NEBuilder HiFi DNA Assembly Kit. To construct the plasmid expressing pcDNA3.1-E3-myc-E2-6K-E1-3xflag, the amplified E3-myc-E2-6K-E1 fragment was incorporated into the pcDNA3.1-3xflag vector, which transiently expresses the CHIKV poly-glycoprotein with a C-terminally 3xflag-tagged E1.

### Quantitative PCR

For intracellular viral RNA detection, cells were lysed with TRIzol reagent (Thermo Fisher Scientific) followed by extraction of total RNAs through the Direct-zol RNA Microprep Kit (Zymo Research) according to the manufacturer's instructions. For quantifying viral copy number, viral RNAs from secreted particles in the cell culture supernatant samples were extracted through PureLink Viral RNA/DNA Kit (Invitrogen) according to the manufacturer's instructions. To enhance assay specificity, tagged reverse transcription primers targeting viral genes were used to synthesize viral cDNAs from total RNAs. The transcribed cDNAs were then quantified by SYBR Green or TaqMan qPCR.

The SYBR Green assay was used to evaluate the copy number of intracellular $(+)$ vRNAs in the samples. To generate standard curve transcripts, full-length CHIKV E1 and partial ONNV E1 (SG650 bp 10092-11361) sequences were amplified with reverse primers containing the SP6 promoter and inserted into the pcDNA3.1 vector with an inherent T7 promoter at the 5′ terminal end. The $(+)$ and $(-)$ standard curve transcripts were synthesized with T7 polymerase using HindIII-linearized plasmid and Sp6 polymerase using NheI-linearized plasmid, respectively, through the MAXIscript™ SP6/T7 Transcription Kit (Thermo Fisher Scientific). The cDNAs of $(+)$ standard curve transcripts and viral RNA in the samples were reverse transcribed with a reverse E1 primer containing a nongenomic tag sequence (Pinto et al, 2006) 5′-CAGACAGCACTCGTTCGTACAC-3′ through the Protoscript II First Strand cDNA Synthesis Kit (NEB). The $(+)$ standard curve cDNAs were then serially diluted ten-fold from $10^{-1}$ to $10^{-8}$ and run through the SYBR Green assay (NEB) together with sample cDNAs. Specific forward primer targeting E1 and a reverse primer targeting the nongenomic tag were used in 20 ul SYBR Green reaction with 1x Luna qPCR Dye (NEB) according to the manufacturer's instructions. The reactions were run under the cycling conditions as previously reported (Luu et al, 2021).

The TaqMan assay was performed to determine the copy number of intracellular $(-)$ vRNAs and supernatant $(+)$ vRNAs from infected cells. The standard curve of $(-)/(+)$ strand nsP1 from CHIKV or ONNV, tagged reverse transcription primers, qPCR primers, and TaqMan probes were designed and generated as previously described (Plaskon et al, 2009). Briefly, a portion of

CHIKV or ONNV nsP1 was cloned into pcDNA3.1(+) with a T7 promoter at the 5′ terminus and an SP6 promoter at the 3′ terminus. The $(-)$ and $(+)$ transcripts of nsP1 were transcribed through SP6 and T7 promoters with MAXIscript SP6/T7 Transcription Kit (Invitrogen) respectively. For intracellular $(-)$ viral RNA detection, the partial nsP1 cDNAs of $(-)$ viral RNA in the samples were synthesized with a forward nsP1 primer containing a unique tag sequence 5′-GGCAGTATCGTGAATTC-GATGC-3′ by the Protoscript II First Strand cDNA Synthesis Kit. The appropriate reverse nsP1 primer, tag-specific forward primer, and FAM-labeled TaqMan probe (synthesized by Integrated DNA Technologies, IDT) were used in viral negative-strand quantification with Luna Universal Probe qPCR Master Mix (NEB). The reactions were run under the cycling conditions as follows: initial denaturation step at 95 °C for 1 min followed by 40 cycles of 95 °C for 15 s and 60 °C for 30 s. Data collection occurs during the 60 °C extension step. For supernatant $(+)$ viral RNA detection, $(+)$ nsP1 partial transcripts generated from standard curve plasmids were 10-fold serially diluted in water to create qPCR standard curves. The partial $(+)$ nsP1 transcripts in the supernatant samples were amplified with specific primers but detected by the same FAM-labeled TaqMan probe that was used in $(-)$ nsP1 qPCR with PrimeTime One-Step RT-qPCR Master Mix (IDT) according to the manufacturer's protocol.

Both SYBR Green and TaqMan reactions were performed in technical duplicates of cDNA/vRNA samples from biological replicates. All qPCR reactions were run on the CFX96 OPUS (Bio-Rad). The total copy number of viral RNA was determined by using the standard curve method. All the primers used in qPCR assays are listed in Table D in Dataset EV2.

### Positive selection analysis, E2 and E1 alignments

Chikungunya virus (taxid: 37124) structural polyprotein sequences were downloaded from the NCBI Virus database. Sequences that were not isolated from a human host, less than 10,000 nucleotides in length, or had more than 0.5% of ambiguous characters were excluded; 556 sequences remained.

To guide the nucleotide alignment, the sequences were first translated to amino acids with HyPhy's Codon-aware MSA program (pre-msa). The amino acids were aligned with MUSCLE and used to align the nucleotide sequences with HyPhy's Codon-aware MSA program (post-msa). A maximum-likelihood phylogenetic tree was constructed by IQ-TREE (Minh et al, 2020). By using HyPhy's FEL (Kosakovsky Pond and Frost, 2005) and MEME (Murrell et al, 2012) methods, positive selection analyses were performed on 397 sequences after the exclusion of duplicates from the original 556 sequences.

To visualize the positively selected sites in E2 and E1 proteins of different CHIKV and ONNV strains. The structural polyprotein sequences of CHIKV 181/clone 25 (GenBank: AAA53256.3), CHIKV Asian strain (GenBank: ABO38821.1), CHIKV Caribbean strain (GenBank: AUS84054.1), CHIKV SL15649 strain (GenBank: ACZ72971.1), CHIKV LR2006 OPY1 strain (GenBank: ABD95938.1), CHIKV West African 37997 strain (GenBank: AAU43881.1), ONNV SG650 strain (GenBank: AAC97205.1), ONNV Gulu strain (GenBank: AAA46785.1), and ONNV Ahero strain (GenBank: AOS52786.1) were downloaded from NCBI database. The sequences were aligned with MUSLE, formatted with Biostrings and visualized with ggmsa (Zhou et al, 2022).

### Co-immunoprecipitation and immunoblot

To prepare samples for AP-MS, THP-1 monocytes were differentiated into macrophages in 36 15-cm dishes with $2 \times 10^7$ cells per dish. Half of the dishes were either infected with CHIKV vaccine strain 181/clone 25 or CHIKV/myc-E2 at an MOI of 5 pfu/cell. Forty-eight hours later, cells in each dish were lysed with 2 mL NP40 lysis buffer (100 mM Tris-HCl (pH 8.0), 5 mM EDTA, 150 mM NaCl, 0.1% NP40, 5% glycerol) supplemented with 1X PMSF, 2X PPI, 1 uM DTT, and Complete EDTA-free protease inhibitor mixture tablet (Roche). Cell lysates from every six dishes under the same treatment were combined and further centrifuged at $14000 \times g$ for 15 min. The clarified supernatants were incubated with anti-myc agarose beads (EZview™ Red Anti-c-Myc Affinity Gel, Millipore) for 4 h at 4 °C. After washing with NP40 lysis buffer four times, proteins were eluted with urea buffer (8 M urea, 100 mM Tris-HCl (pH 8)) for mass spectrometry analysis.

To validate CHIKV glycoprotein interactions with host factors identified by AP-MS, 293T cells were seeded in six-well plates at a starting density of $1.5 \times 10^5$ cells/well, followed by transient transfection with plasmids expressing 3xflag-tagged host factors (OAS3, PKR, SPCS3, eIF3k, and APOBEC3F), empty vector, or control plasmid expressing 3xflag-tagged TRIM25 through X-tremeGENE9 (Roche). Immunoprecipitation of flag-tagged host factors in clarified supernatants with anti-flag agarose beads (EZview™ Red Anti-flag M2 Affinity Gel, Sigma-Aldrich) was performed at 4 °C for 45 min. After 4x washing, proteins were directly eluted with Laemmli Sample Buffer (Bio-Rad) containing 5% 2-mercaptoethanol and denatured by 99 °C 10-min incubation.

For E2 and E1 reciprocal immunoprecipitation (IP), CHIKV E2 antibody (CHK-48), or CHIKV E1 antibody (GeneTex) was conjugated to Dynabeads Protein G (Invitrogen) through 20 min room temperature incubation with rotation at the ratio of 2.2 μg antibodies per mg beads. The E2 antibody-conjugated Dynabeads were incubated with the lysates of 293T cells that were transfected with plasmids expressing 3xflag-SPCS3/eIF3k and CHIKV E3-myc-E2-6k-E1 for 20 min at room temperature to pull down E2. The pulldown efficiency of E2 was verified by myc antibody (Cell Signaling Technology) through immunoblot. For E1 immunoprecipitation, 293T cells were transfected with plasmids expressing 3xflag-SPCS3/eIF3k and E3-myc-E2-6k-E1-3xflag, followed by incubation of cell lysates with E1 antibody-conjugated beads. The pulldown of E1 was evaluated with flag antibody (Millipore Sigma) through immunoblot.

Proteins were resolved by SDS-PAGE in 4–15% precast Mini-PROTEAN TGX Gels (Bio-Rad) in conventional Tris/Glycine/SDS buffer. Proteins were blotted to the PVDF membrane (Bio-Rad) and detected with primary antibodies and HRP-conjugated secondary antibodies listed in the Reagent and tools table. Immunoblots were imaged by chemiluminescence with the ProSignal Pico ECL Reagents (Genesee Scientific) on a ChemiDoc (Bio-Rad).

### Mass spectrometry

Two independent AP-MS experiments were performed to identify macrophage proteins that interact with CHIKV glycoproteins. For mass spectrometry, protein disulfide bonds were subjected to reduction using 5 mM Tris (2-carboxyethyl) phosphine for 30 min, and free cysteine residues were alkylated by 10 mM iodoacetamide for another 30 min. Samples were diluted with 100 mM Tris-HCl at

pH 8 to reach a urea concentration of less than 2 M, and then digested sequentially with Lys-C and trypsin at a 1:100 protease-to-peptide ratio for 4 and 12 h, respectively. The digestion reaction was terminated by the addition of formic acid to 5% (vol/vol) with centrifugation. Finally, samples were desalted using C18 tips (Thermo Scientific, 87784), dried in a SpeedVac vacuum concentrator, and reconstituted in 5% formic acid for LC-MS/MS processing.

Tryptic peptide mixtures were loaded onto a 25 cm long, 75-μm inner diameter fused-silica capillary, packed in-house with bulk 1.9 μM ReproSil-Pur beads with 120 Å pores as described previously (Jami-Alahmadi et al, 2021). Peptides were analyzed using a 140 min water-acetonitrile gradient delivered by a Dionex Ultimate 3000 UHPLC (Thermo Fisher Scientific) operated initially at 400 nL/min flow rate with 1% buffer B (acetonitrile solution with 3% DMSO and 0.1% formic acid) and 99% buffer A (water solution with 3% DMSO and 0.1% formic acid). Buffer B was increased to 6% over 5 min at which time the flow rate was reduced to 200 nl/min. A linear gradient from 6–28% B was applied to the column over the course of 123 min. The linear gradient of buffer B was then further increased to 28–35% for 8 min followed by a rapid ramp-up to 85% for column washing. Eluted peptides were ionized via a Nimbus electrospray ionization source (Phoenix S&T) by application of a distal voltage of 2.2 kV.

All label-free mass spectrometry data were collected using data-dependent acquisition on Orbitrap Fusion Lumos Tribrid mass spectrometer (Thermo Fisher Scientific) with an MS1 resolution of 120,000 followed by sequential MS2 scans at a resolution of 15,000. Data generated by LC-MS/MS were searched using the Andromeda search engine integrated into the MaxQuant 2 bioinformatic pipelines against the UniProt *Homo sapiens* reference proteome (UP000005640 9606) and then filtered using a "decoy" database-estimated false discovery rate (FDR) <1%. Label-free quantification (LFQ) was carried out by integrating the total extracted ion chromatogram (XIC) of peptide precursor ions from the MS1 scan. These LFQ intensity values were used for protein quantification across samples. Statistical analysis of differentially expressed proteins was done using the Bioconductor package ArtMS3. Samples were normalized by median intensity.

### Bioinformatic analysis of mass spectrometry data

Due to higher protein abundance, results from our second AP-MS experiment were visualized by histogram and volcano plot (ggplot2.tidyverse.org) to show the fold change distribution of host factors that were significantly enriched by myc pulldown in CHIKV/myc-E2 infected macrophages. To perform gene ontology analysis, candidate host interactors were first filtered by the cut-offs of $p$ value <0.05 and Log$_2$ fold change >0, based on the comparison of CHIKV/myc-E2 treatment group to CHIKV 181/clone 25 treatment group. We then used the CRAPome (contaminant repository for affinity purification) database (crapome.org) to remove potential contaminant proteins by a cutoff of $\geq$ 200 appearances in 716 recorded experiments. The filtered host factors were submitted to The Database for Annotation, Visualization and Integrated Discovery (DAVID: david.ncifcrf.gov) to analyze the enriched biological process (BP) categories. To have an intuitive view of all the BP categories, EnrichmentMap (Merico et al, 2010) in Cytoscape (Shannon et al, 2003) was used to generate the network of all the BP enrichment results. The KEGG pathway

analysis on host factors was performed by the latest online KEGG database (kegg.jp) downloaded in ClusterProfiler (Wu et al, 2021) in R, and the distribution of core enriched host factors for KEGG categories were visualized through ridgeplot in ClusterProfiler.

For the CORUM protein–protein interaction network, we recovered 37 hits from experiment 1 and 1157 hits from experiment 2 ($p$ value <0.05 and Log2 fold change >0). There were 14 hits overlapping between the two experiments. We first attempted to identify known host protein complexes among the overlapping hits using the CORUM database (Tsitsiridis et al, 2023), a manually curated database of high-confidence protein complexes, but found none. Next, we searched for protein complexes (again, using the CORUM database) in either experiment 1 or experiment 2, reasoning that although indirect protein–protein interactions may be lower abundance in the affinity purification they may be recovered in at least one of the experiments. Our final visualization (Fig. 5C) required that (1) protein complex members pass additional stringency criteria of $Log_2$ fold change >2 from either experiment 1 or 2 and (2) protein complexes possess at least one protein member that was an overlapping hit between experiments 1 and 2.

*Immunofluorescence staining and Airyscan microscopy*

To analyze the colocalization of CHIKV E1 with E2 or with host factors (SPCS3, eIF3k), 293T cells were grown on collagen-coated coverslips (Corning BioCoat) and transfected with plasmids expressing 3xflag-tagged SPCS3 or eIF3k followed by CHIKV infection one day later. Twenty-four-hours post infection, the 293T cells were fixed with 4% formaldehyde (Fisher Scientific) in PBS (v/v) for 15 min and sequentially washed with 300 mM glycine (Fisher Scientific) in PBS to quench unreacted formaldehyde residues. The cells were then permeabilized in 0.1% Triton-X100 (Sigma-Aldrich) in PBS (v/v) and blocked in 3% FBS/PBS at room temperature for 1 h. The cells were incubated with primary antibodies targeting CHIKV E1 (GeneTex) and E2 (CHK-48) (Fox et al, 2015) (BEI Resources) diluted in blocking buffer at 4 degrees overnight, followed by 2-h incubation with secondary antibodies and Alexa Fluor 488 conjugated flag antibody (Invitrogen). Coverslips were mounted on glass slides with Fluoromount-G (Invitrogen) and imaged by ZEISS LSM 880 with an Airyscan detector.

To evaluate the colocalization of SPCS3 and eIF3k, 293T cells were co-transfected with plasmids expressing 3xflag-tagged eIF3k and V5-tagged SPCS3 and infected with CHIKV 1 day after. Primary antibodies targeting CHIKV E1 (GeneTex) and V5 (Millipore Sigma) were diluted in the blocking buffer and incubated with the cells at 4 degrees overnight, followed by 2-hour incubation with secondary antibodies and Alexa Fluor 488 conjugated flag antibody. Mounting and imaging were performed as previously described. The antibodies used in the immunofluorescence staining are listed in the Reagent and tools table.

We used the Fiji Coloc2 plugin to analyze the colocalization of E1 and E2, E1 and host factors, and SPCS3 and eIF3k. For each analysis, we selected four cells that were identified to express all the relevant proteins as regions of interest (ROI) and calculated average Pearson correlation coefficient values.

*Flow cytometry*

After 24 h incubation with EGFP-labeled alphaviruses, primary human monocyte-derived macrophages were detached from 12-well plates by using ACCUMAX (Stemcell Technologies). Digested macrophages were washed with PBS two times in 96-well plates and fixed in fixation buffer (1% paraformaldehyde (PFA), 1% FBS in PBS). The intracellular EGFP expressions were detected by MACSQuant Analyzer (Miltenyi Biotec) with a minimum collection of 20,000 events per sample. The results were analyzed through FlowJo (Tree Star).

## Data availability

The datasets and computer code produced in this study are available in the following databases:

Protein interaction AP-MS data:

MassIVE repository

Accession number: MSV000094494 (https://massive.ucsd.edu/ProteoSAFe/dataset.jsp?task=db9adf314352491a8bbc20ca5291a838)

The source data of this paper are collected in the following database record: biostudies:S-SCDT-10_1038-S44318-024-00193-3.

## Peer review information

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

## Acknowledgements

We thank Dr. Stephen Higgs (Kansas State University) for providing the infectious clone constructs of CHIKV (La Réunion strain) and ONNV (SG650) and for his help with CHIKV-ONNV chimera constructions. We thank Dr. Scott Weaver (The University of Texas Medical Branch at Galveston) and Dr. Mark Heise (The University of North Carolina at Chapel Hill) for providing the infectious clone constructs of CHIKV (AF15561, 181/clone 25) and RRV (T48). We thank Dr. Charles M Rice for providing the infectious clone constructs of SINV (Toto1101, TE/5'2J/GFP). We thank Dr. Joyce Jose and her student Zeinab Elmasri (Pennsylvania State University), as well as Dr. Graham Simmons and Dr. Jing Jin (Vitalant Research Institute) for their help with the construction of the CHIKV E2 reporter virus. We express our gratitude to Dr. Sergei L Kosakovsky Pond (Temple University), and his students Jordan Zehr and Alexander Lucaci for their insights on positive selection analysis. We also thank Dr. Oliver Fregoso (UCLA) for his comments on this study. We thank the UCLA Proteome Research Center for their services. We thank the instructions and supports from Microscopy Core of UCLA Broad Stem Cell Reseaerch Center for setting up the Airyscanning. We thank the qPCR and flow cytometry platforms provided by UCLA AIDS Institute which is supported by the James B Pendleton Charitable Trust and the McCarthy Family Foundation. We also thank UCLA/CFAR Virology Core Lab (grant number 5P30 AI028697) for providing human primary monocytes. The following reagent was obtained through BEI Resources, NIAID, NIH: Monoclonal Anti-Chikungunya Virus E2 Envelope Glycoprotein, Clone CHK-48 (produced in vitro), NR-44002. This work was funded in part by NIH R01AI158704 (MMHL) and UC Cancer Research Coordinating Committee Faculty Seed Grant (CRN-20-637544; MMHL). JAW was supported by NIH GM089778 and R35GM153408. MB was supported by an HIV Accessory and Regulatory Complexes (HARC) Collaborative Development Award and a Center for AIDS Research Pilot Grant. ZY was supported by the Sydney Finegold Post-Doctoral Fellow Award. EK was supported by Ruth L. Kirschstein National Research Service Award AI007323.

## Author contributions

**Zhenlan Yao**: Conceptualization; Data curation; Software; Formal analysis; Validation; Investigation; Visualization; Methodology; Writing—original draft; Writing—review and editing. **Sangeetha Ramachandran**: Data curation; Formal analysis; Validation; Investigation; Visualization; Writing—original draft; Writing—review and editing. **Serina Huang**: Data curation; Software; Formal analysis; Visualization; Writing—original draft; Writing—review and editing. **Erin Kim**: Data curation; Investigation; Writing—review and editing. **Yasaman Jami-Alahmadi**: Data curation; Software; Formal analysis; Investigation; Writing—original draft. **Prashant Kaushal**: Data curation; Software; Formal analysis; Investigation. **Mehdi Bouhaddou**: Resources; Software; Formal analysis; Visualization; Writing—original draft. **James A Wohlschlegel**: Resources; Methodology; Writing—review and editing. **Melody MH Li**: Conceptualization; Resources; Data curation; Supervision; Funding acquisition; Methodology; Project administration; Writing—review and editing.

Source data underlying figure panels in this paper may have individual authorship assigned. Where available, figure panel/source data authorship is listed in the following database record: biostudies:S-SCDT-10_1038-S44318-024-00193-3.

## Disclosure and competing interests statement

The authors declare no competing interests.

# Expanded View Figures

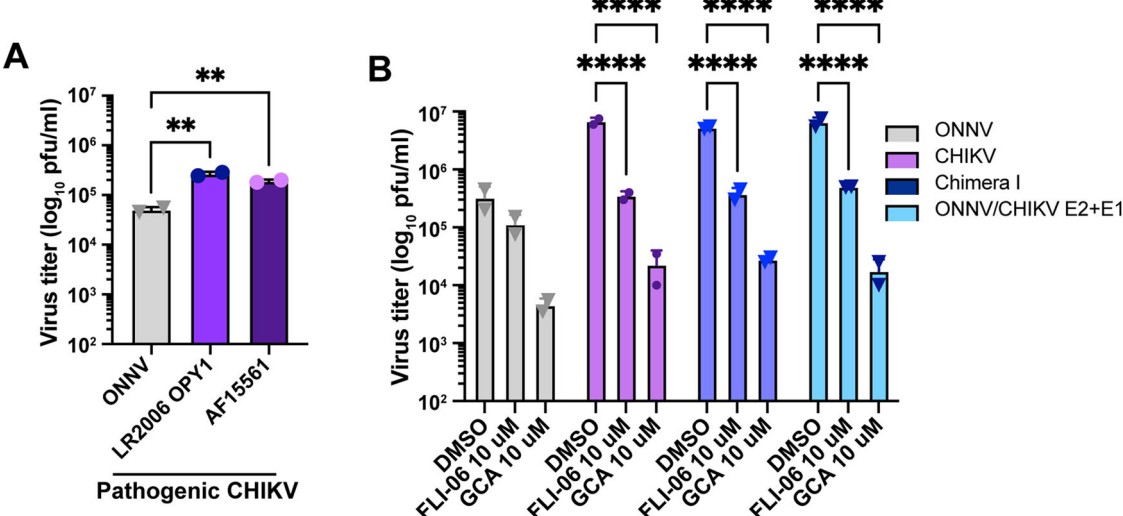

**Figure EV1.  The advantage of virus production in macrophages is also recapitulated by pathogenic CHIKV and depends more on the host secretory pathway.**

(A) THP-1-derived macrophages were infected with ONNV SG650, CHIKV La Réunion strain (LR2006 OPY1), and CHIKV Asian strain (AF15561) at MOI 5. Titration of supernatant infectious particles was performed at 24 h.p.i by plaque assay on BHK-21 cells. The incubation period for plaque assay takes 28 h. Data were representative of three independent experiments. Mean values of biological duplicates were plotted with SD. Asterisks indicate statistically significant differences as compared to ONNV (One-way ANOVA and Dunnett's multiple comparisons test: ONNV vs LR2006 OPY1 \*\*$p = 0.0024$; ONNV vs AF15561 \*\*$p = 0.0082$). (B) The influence of secretary pathway inhibition on the infections of ONNV, CHIKV, Chimera I, and ONNV/CHIKV E2 + E1. The THP-1-derived macrophages were pretreated with 10 μM FLI-06 or GCA for 30 min prior to 1-h inoculation with ONNV, CHIKV, Chimera I, or ONNV/ CHIKV E2 + E1. The cells were then cultured with the inhibitors at the same concentration (10 μM) for 24 h. The virus titers from supernatants were analyzed by plaque assay as previously described. Data were representative of two independent experiments. Mean values of biological duplicates were plotted with SD. Asterisks indicate statistically significant differences as compared to ONNV (one-way ANOVA and Dunnett's multiple comparisons test: DMSO vs FLI-06/GCA with the infection of CHIKV, Chimera I, or ONNV/CHIKV E2 + E1 \*\*\*\*$p < 0.0001$).

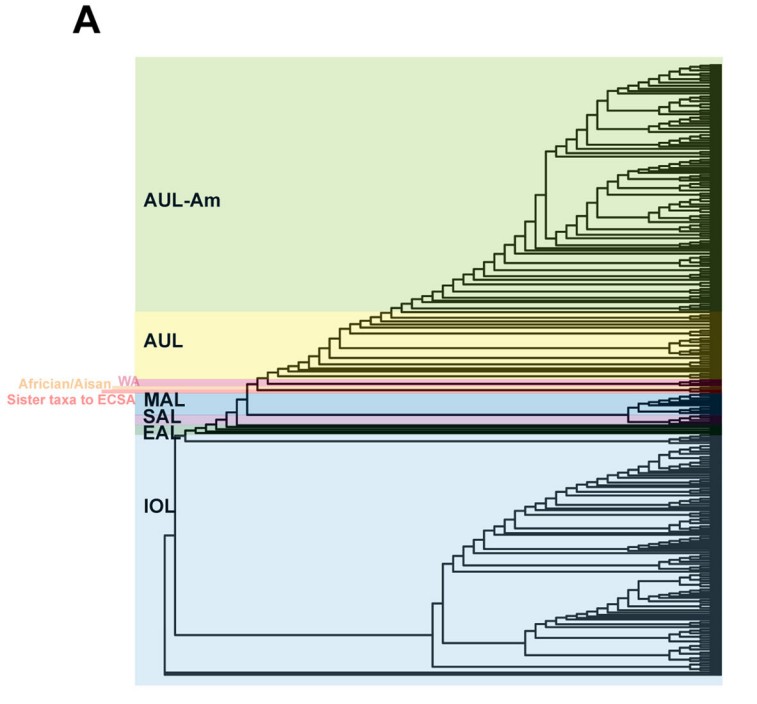

**B**

| AA location (structural polyprotein) | FEL (p<0.05) | MEME (p<0.05) | CHIKV 181/clone 25 | ONNV SG650 |
|---|---|---|---|---|
| 147 | | C-147 | Ala | Ala |
| 156 | | C-156 | Arg | Arg |
| 460 | | E2-135 | Val | Leu |
| 489 | E2-164 | E2-164 | Ala | Thr |
| 538 | | E2-213 | Thr | Thr |
| 546 | | E2-221 | Lys | Lys |
| 571 | | E2-246 | Ala | Ser |
| 795 | 6K-47 | 6K-47 | Thr | Thr |
| 847 | | E1-38 | Leu | Leu |
| 954 | E1-145 | E1-145 | Ser | Ser |
| 1020 | E1-211 | E1-211 | Glu | Lye |
| 1029 | | E1-220 | Val | Ile |
| 1163 | | E1-354 | Ile | Ile |
| 1175 | | E1-366 | Arg | Lys |

**C**

| | No. of sequences at E2-135 | No. of sequences at E1-220 |
|---|---|---|
| Valine (V) | 394 | 392 |
| Alanine (A) | 1 | 3 |
| Methionine (M) | 1 | 0 |
| Glutamic acid (E) | 0 | 1 |
| Glycine (G) | 1 | 0 |
| Isoleucine (I) | 0 | 1 |

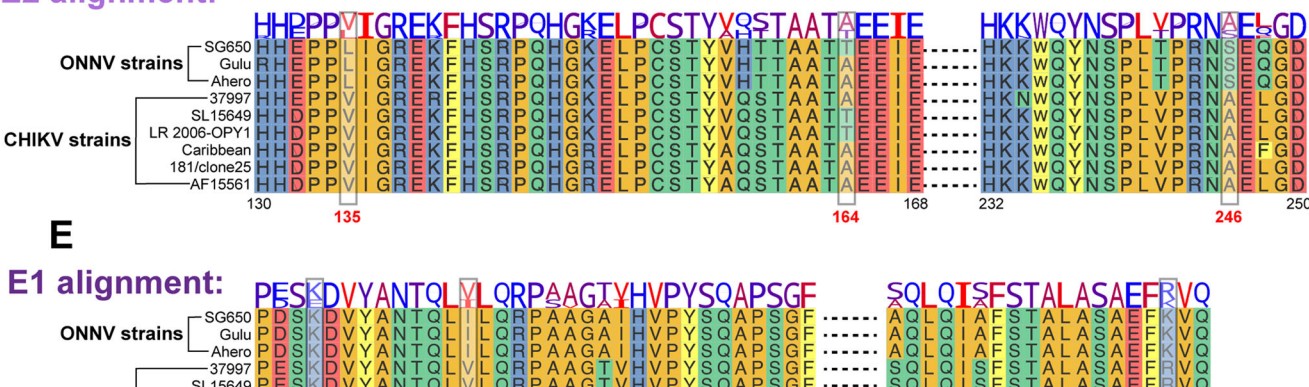

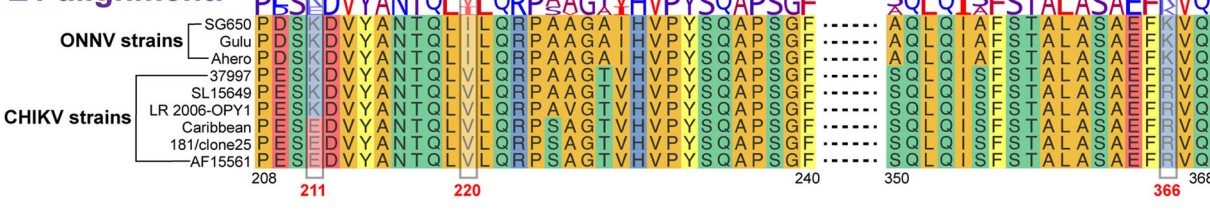

**Figure EV2.  Evolutionary selection analysis on CHIKV structural proteins.**

(A) Phylogenetic tree constructed by IQ-tree (Minh et al, 2020) using an alignment of the CHIKV structural polyprotein. The tree was visualized by ggtree (Yu et al, 2017). Tree branches were colored according to the latest CHIKV lineage classification (de Bernardi Schneider et al, 2019) used in CHIKVnext v3 (nextstrain.org/groups/ViennaRNA/CHIKVnext/v3.0). AUL-Am Asian urban + American lineage, AUL Asian urban lineage, EAL Eastern African lineage, IOL Indian Ocean lineage, MAL Middle African lineage, SAL South American lineage, WA Western African lineage. (B) Comparison of CHIKV positively selected sites with homologous sites on ONNV. MEME and FEL were used to analyze the positively selected sites in CHIKV structural proteins and generate *P* values. The *P* values are corrected with Benjamini–Hochberg. The positively selected CHIKV amino acids that are different from the homologous residues in ONNV were colored in red and highlighted in gray. (C) The heterogeneity of residues at E2-135 and E1-220 in 397 CHIKV patient isolates from NCBI Virus database. (D) The E2 alignment of ONNV and CHIKV strains to compare the amino acid residues at E2-135, E2-164, and E2-246. CHIKV 37997 belongs to the West African lineage. CHIKV LR2006 OPY1 and CHIKV SL15649 belong to the East/Central/South African (ECSA) lineage. CHIKV Caribbean and CHIKV AF15561 belong to the Asian lineage. CHIKV AF15561 is the parental strain of CHIKV vaccine strain 181/clone 25. The alignment is visualized through ggmsa (Zhou et al, 2022). (E) The E1 alignment of different ONNV and CHIKV strains to compare the amino acid residues at E1-211, E1-220, and E1-366.

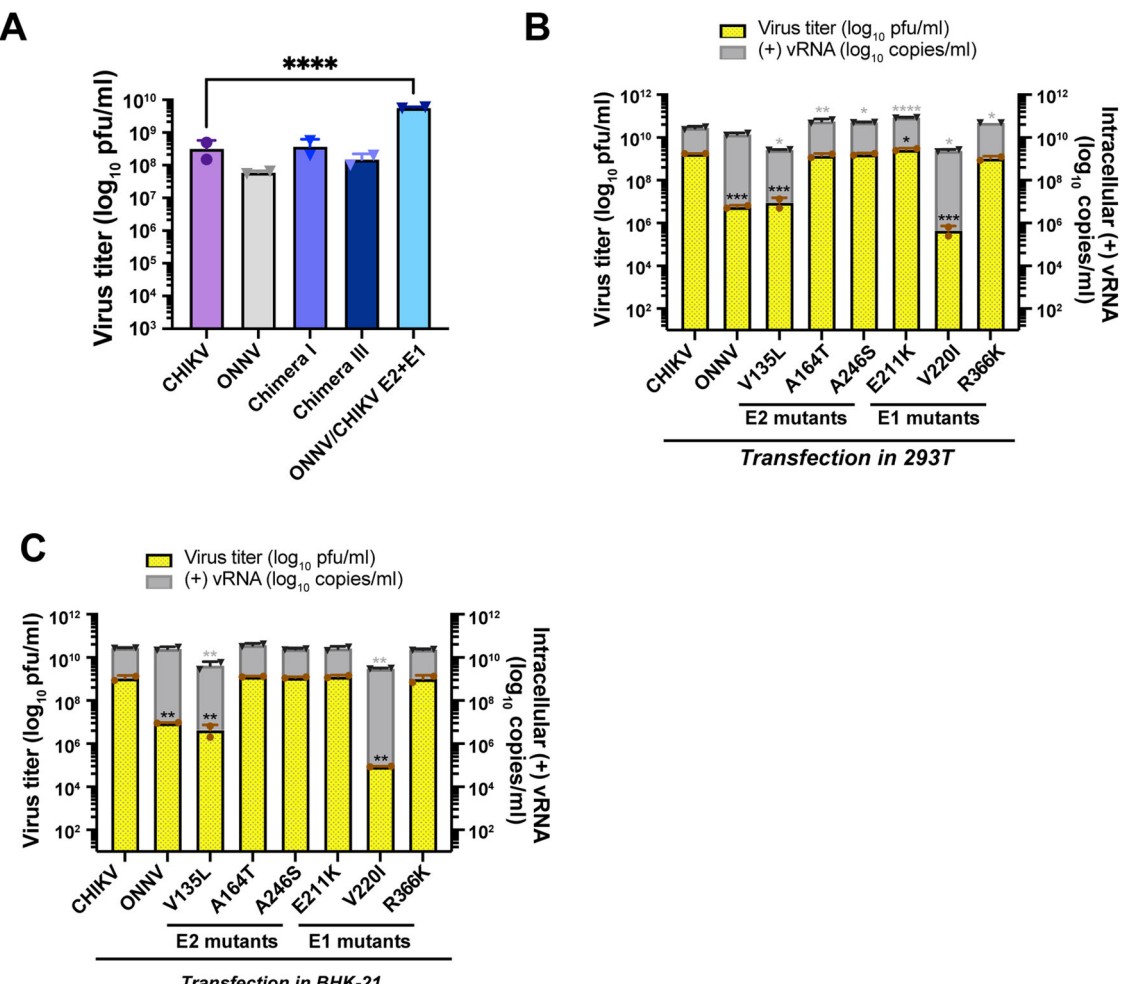

**Figure EV3. The superior virus production conferred by CHIKV structural proteins is macrophage-specific.**

(A) CHIKV, ONNV, Chimera I, Chimera III, and ONNV/CHIKV E2 + E1 infection in 293T cells. Virion production in the supernatant of infected 293T cells was titrated through plaque assay on BHK-21 cells as previously described. Mean values of biological duplicates were plotted with SD. Data were representative of two independent experiments. Asterisks indicate statistically significant differences as compared to CHIKV (one-way ANOVA and Dunnett's multiple comparisons test: CHIKV vs ONNV/ CHIKV E2 + E1 ****$p < 0.0001$). (B, C) Infection of 293T (B) and BHK-21 (C) cells with CHIKV vaccine strain 181/clone 25 positive selection site mutants. Viral replication and production of positive selection site mutants (E2-V135L, E2-A164T, E2-A246S, E1-E211K, E1-V220I, and E1-R366K) were determined by levels of intracellular (+) vRNAs and secreted infectious particles as previously described. For EV3B, data were representative of two independent experiments. The plaque assay results were plotted from biological duplicates with the mean values. Error bars represent SD (one-way ANOVA and Dunnett's multiple comparisons test: viral titer of CHIKV vs ONNV ***$p = 0.0004$; viral titer of CHIKV vs E2-V135L ***$p = 0.0004$; viral titer of CHIKV vs E1-E211K *$p = 0.017$; viral titer of CHIKV vs E1-V220I ***$p = 0.0004$). The qPCR results were plotted from biological duplicates with the mean values. Error bars represent SD (one-way ANOVA and Brown-Forsythe test: viral copies of CHIKV vs E2-V135L *$p = 0.0116$; viral copies of CHIKV vs E2-A164T **$p = 0.0036$; viral copies of CHIKV vs E2-A246S *$p = 0.0156$; viral copies of CHIKV vs E1-E211K ****$p < 0.0001$; viral copies of CHIKV vs E1-V220I *$p = 0.011$; viral copies of CHIKV vs E1-R366K *$p = 0.0274$). For EV3C, data were representative of two independent experiments. The plaque assay results were plotted from biological duplicates with the mean values. Error bars represent SD (one-way ANOVA and Dunnett's multiple comparisons test: viral titer of CHIKV vs ONNV **$p = 0.006$; viral titer of CHIKV vs E2-V135L **$p = 0.0058$; viral titer of CHIKV vs E1-V220I **$p = 0.0057$). The qPCR results were plotted from biological duplicates with the mean values. Error bars represent SD (one-way ANOVA and Brown-Forsythe test: viral copies of CHIKV vs E2-V135L **$p = 0.0027$; viral copies of CHIKV vs E1-V220I **$p = 0.0019$).

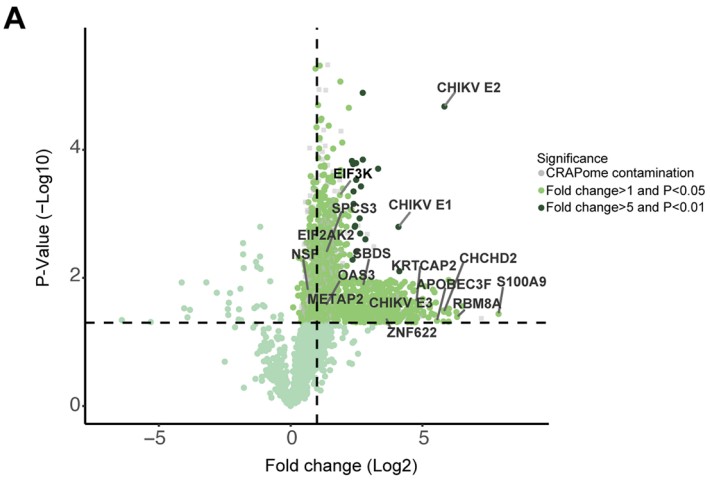

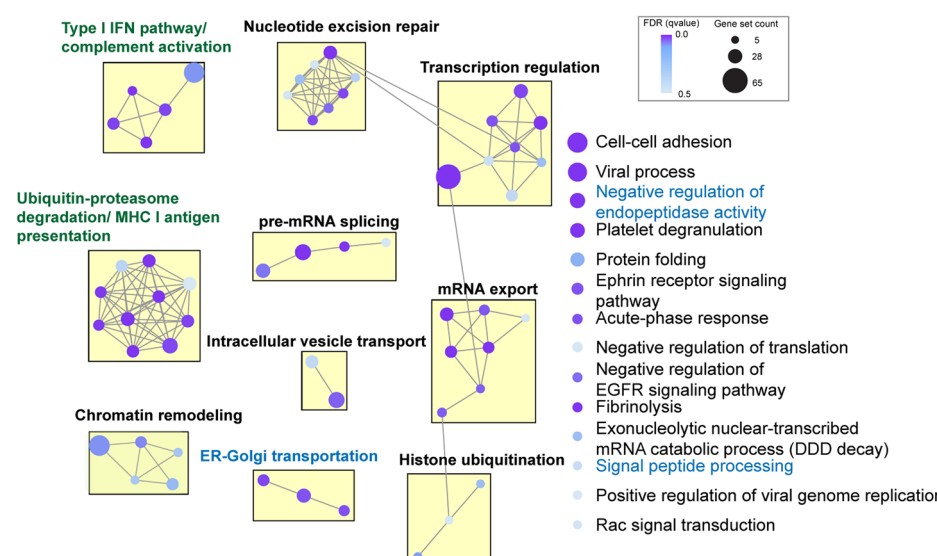

**Figure EV4. The macrophage host factors identified by AP-MS and representative biological processes of significantly enriched host factors.**

(**A**) Volcano plot depicting cellular interactors of CHIKV glycoproteins identified by mass spectrometry. A moderated *t*-test from R package ArtMS3 was used to generate the *P* values which were adjusted with Benjamini–Hochberg for the multiple hypothesis correction. The volcano plot is scattered by $-\log_{10}$ *P* value (y-axis) and $\log_2$ expression fold change (FC) of proteins co-immunoprecipitated from CHIKV/myc-E2 infected cells with respect to the proteins from CHIKV WT infected cells (x-axis). The dashed cut-offs of the adjusted *P* value and expression fold change are 0.05 ($-\log_{10}$ *P* value $= 1.30103$) and 2 ($\log_2$FC $= 1$), respectively. CHIKV glycoproteins (E3, E2, E1) and host factors for further investigation in Fig. 6A are annotated here. (**B**) Enrichment map that summarizes over-represented biological processes of identified host factors in groups. The enriched proteins identified by mass spectrometry were clustered by biological processes and organized into a network with edges connecting overlapping gene sets to reveal functional modules.

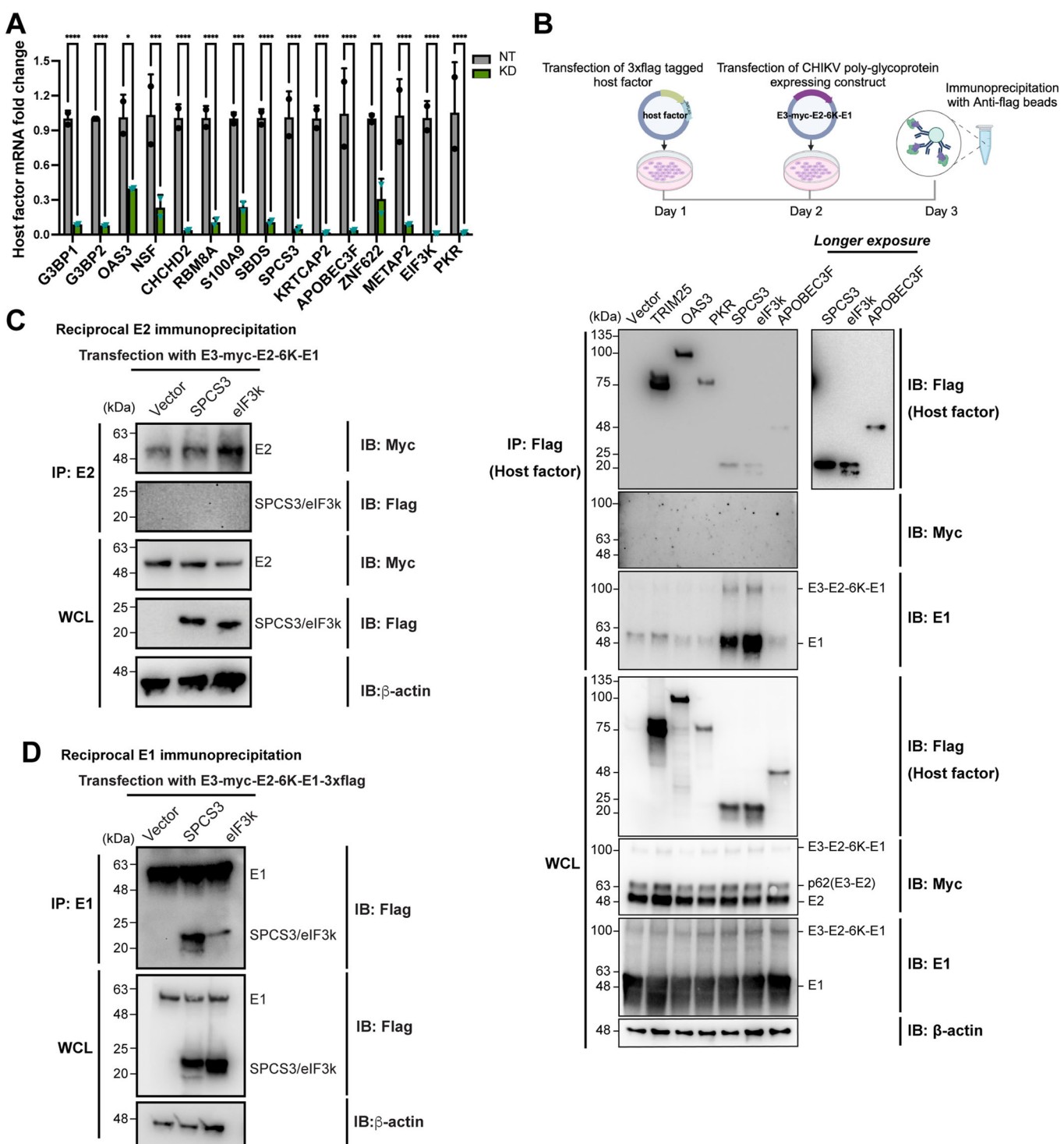

**Figure EV5.　New antiviral host factors, SPCS3 and eIF3k, specifically interact with CHIKV E1.**

(A) The macrophages were transfected with 25 nM nontargeting siRNAs (NT) or pooled siRNAs targeting host factors (G3BP1, G3BP2, OAS3, NSF, CHCHD2, RBM8A, S100A9, SBDS, SPCS3, KRTCAP2, APOBEC3F, ZNF622, METAP2, EIF3K, and PKR). mRNAs of cells treated with siRNAs were extracted 48 h post transfection for RT-qPCR to evaluate the host factor knockdown efficiencies. Data were representative of two independent experiments. The mean values of biological duplicates were plotted with SD (two-way ANOVA and Šidák's multiple comparisons test: si-NT vs si-OAS3 *$p = 0.0118$; si-NT vs si-ZNF622 **$p = 0.0026$; si-NT vs si-NSF ***$p = 0.0006$; si-NT vs si-S100A9 ***$p = 0.0009$; si-NT vs si-G3BP1/G3BP2/CHCHD2/RBM8A/SBDS/SPCS3/KRTCAP2/APOBEC3F/METAP2/EIF3K/PKR ****$p < 0.0001$). (B) 293T cells were transfected with plasmids expressing 3xflag-tagged host factors (TRIM25, OAS3, SPCS3, APOBEC3F, eIF3k, and PKR) or empty vector control for 24 h, and later transfected with plasmid expressing CHIKV glycoproteins (E3-myc-E2-6K-E1). The cells were lysed and immunoprecipitated by anti-flag agarose beads. Immunoblot was probed to check for E2/E1 binding to these host factors. TRIM25-3xflag was transfected into 293T cells for immunoprecipitation control. Data were representative of three independent experiments. (C) 293T cells were transfected with plasmids expressing 3xflag-tagged host factors (SPCS3, eIF3k) or empty vector control for 24 h, followed by transfection with the plasmid expressing CHIKV E3-myc-E2-6K-E1. The cells were lysed for immunoprecipitation with Dynabeads Protein G conjugated with E2 antibody (CHK-48) (Fox et al, 2015). Immunoblot was probed for host factor (SPCS3, eIF3k) binding to E2. Data were representative of two independent experiments. (D) 293T cells were transfected with plasmids expressing 3xflag-tagged host factors (SPCS3, eIF3k) or empty vector control for 24 h, followed by transfection with the plasmid expressing CHIKV E3-myc-E2-6K-E1-3xflag. The cells were lysed for immunoprecipitation with Dynabeads Protein G conjugated with E1 antibody. Immunoblot was probed for host factor (SPCS3, eIF3k) binding to E1. Data were representative of two independent experiments.

　　　　　　　　　　　　　　　　