## [Peer Review File · The EMBO Journal]

Interaction of chikungunya virus glycoproteins with macrophage factors controls virion production.

Zhenlan Yao, Sangeetha Ramachandran, Serina Huang, Erin Kim, Yasaman Jami-Alahmadi, Prashant Kaushal, Mehdi Bouhaddou, James Wohlschlegel, and Melody Li

Corresponding author(s): Melody Li (manhingli@mednet.ucla.edu)

Review Timeline:

Transfer from Review Commons:	22nd Sep 23
Editorial Decision:	28th Sep 23
Appeal Received:	18th Apr 24
Editorial Decision:	19th Jun 24
Revision Received:	16th Jul 24
Accepted:	17th Jul 24

Review
COMMONS

Editor: Ioannis Papaioannou

Transaction Report: This manuscript was transferred to The EMBO JOURNAL following peer review at Review Commons.

Review #1

1. Evidence, reproducibility and clarity:

Evidence, reproducibility and clarity (Required)

Summary:

In this work Yao et al. show CHIK is able to infect macrophages in contrast to other arthritogenic alphaviruses RRV, ONNV, and SINV. They use a series of chimeric viruses made with ONNV, the closest species to CHIK, and determine the E2-E1 proteins are important viral determinants which allow CHIK to replicate in macrophages compared to ONNV. By comparing 397 CHIK sequences from infected patients, they identified 14 residues under pervasive and positive selection. Of these, 3 residues in E2 and 3 residues in E1 (amino acids) were different between CHIK and ONNV suggesting these residues contributed to the difference in macrophage tropism of CHIK compared to ONNV. The authors go on to determine what host factors the CHIK E2 protein is interacting with to presumably connect the viral and host determinants for CHIK infection in macrophages.

Major concerns:

1. The authors show one configuration of the E1-E2 heterodimer in Figure 4d. As shown, the E1 protein is exterior to the E2 protein and would suggest E1 is on the surface on the spike complex and virus surface. However, another configuration of the glycoproteins has E2 on the exterior of E1 and also on the exterior of the virus. The latter conformation is what has been observed in cryoEM studies of alphaviruses. The first configuration represents the E1-E2 between the three heterodimers which are important for spike assembly. The reason the orientation of the E2-E1 dimer is important is the authors speculate on the importance of the 6 CHIK residues not found in ONNV based on the structure, but the structural interpretation is, in my opinion, not correct.

2. Validation of E1 interaction with SPSC3 and eIF3k needs to be stronger. Some concerns/questions are listed below. A myc tag was inserted between E3 and E2. How efficiently does furin cleave E3 from E2 in this virus and how are viral titers of the myc-tagged virus compared to the non-tagged virus? I ask because is the IP looking at what is being pulled down by E2 or E3-myc-E2 that could be part of the spike polyprotein? The authors found E2 interacts with E3, E1 and a list of other host proteins. These results suggest several interactions including E2-host factor, E2-E1, E2-E3, E2-E1-host factor, E2-E3-E1, E2-E3-host factor. In figure 6d, and the

subsequent conclusions, the authors suggest E1 is interacting with the host factor and do not see E2 alone and very low amounts of E3-E2-6K-E1. based on how the IP was performed I am not sure how an interaction between E1 and SPSC3 alone, without E2, would be detected. I would also like to see a reciprocal pull down using E1 and also E2 to see if these host factors are pulled down.

3. If CHIK E1 is interacting with the host factors and that is antagonizing the antiviral response of SPSC3 (as one example), then what do pull downs using ONNV structural proteins look like? One would expect reduced interactions because the different amino acid causes a different E2-E1 dimer or attenuates the E1-host factor binding site.

4. E1 and E2 are thought to interact during polyprotein translation and the initial dimer forms in the ER. If E1 is interacting with SPSC3 in the ER, is E2 also present? Or is a population of E1 not interacting with E2 in order to inhibit SPSC3? I would love a model of how the authors see all these factors coming together for this new role of E1.

****Minor concerns:****

1. In Figure 1c, (-) RNA is shown but in the rest of the figures (+) RNA is shown. Show both or select one. I do find it interesting the (-) RNA levels are similar over time, even at 4 hours post transfection (early time). Related to this, ONNV has higher levels of (-) RNA but what is known about structural protein levels in ONNV and CHIK in macrophages? Are there comparable levels of CP and GP being produced?

2. Figure 2e and figure 3 have ONNV has the first bar followed by CHIK. In figure 1 and 2b, CHIK is first and then ONNV. helps the reader to have the controls in the same order.

3. Line 143-145 the authors discuss that when ONNV is the backbone and CHIK proteins are inserted the infection is more attenuated because of the E2 and E1 are from CHIK and ONNV, not the same virus (could also be E2-CP interactions are disrupted). However the chimeras made with the CHIK backbone (in Figure 2) have a mismatch between E2 and E1 as well.

4. When discussing the residues that were found in the FEL and MEME analysis, the authors start the amino acid numbering from CP and continue along the polyprotein. Usually when discussing amino acids in the structural proteins, each protein starts at amino acid 1. So V460 would be E2-V135. It would also be useful to know what the residues in ONNV were at these positions to see if amino acids changed in charge, size, bond forming potential, etc. Showing these residues in the E2-E1 conformation found in the virion would also allow one to find adjacent residues that could explain differences in spike assembly and potentially where/how E1 is binding to a host protein.

5. How effective is a non-attenuated CHIK strain in infecting macrophages? Could you make a SINV-La Reunion chimeric virus (which is BSL2) to see if a higher

percentage of macrophages are infected and is this potentially contributing to the increased pathogenesis of La Reunion? Also how different is 181/25 with a pathogenic strain in the E2 and E1 residues? and compared to ONNV?

6. When describing the last results section, "CHIK E1 binding proteins exhibit potent anti-CHIV activities" the authors use macrophages. In the rest of the text they consistently use THP-1 macrophages or human primary monocyte derived macrophages. The details of the cell type are extremely useful to the reader and having those in the last results section would be great.

7. The paper is well-written. There is a slight disconnect as the authors go from discussing results in Figure 4 to Figure 5.

****Referees cross-commenting****

I agree with R#2 that having some Particle:PFU data would add some data to determine why such differences in titers/infectivity.

I also see how this m/s could be split into two different m/s. One that focuses on the chimeric viruses and another that identifies the host factors important and goes in more depth with mechanism

2. Significance:

Significance (Required)

Strengths:

The authors have tackled an intriguing question: why do some alphaviruses infect macrophages and others do not. They have used a chimeric approach to very systematically identify the viral determinants E2 and E1 as being important in macrophage infection. Using AP-MS they identify host factors that interact with E2 (possibly E2 and E1, see comments above) but if their findings that E1 has a role in attenuating a host antiviral factor, this would be fantastic.

More and more examples of viral proteins having multiple roles during infection are in the literature. The idea that structural proteins also attenuate host antivirals is a developing field and vastly understudied. By fleshing out the results some more the authors might be onto something very important in alphavirus virology.

Limitations:

The study as it is presented is limited in the validation of host factors and their interacting partners. I have many questions about the methodology, validation, and model from this last section.

3. How much time do you estimate the authors will need to complete the suggested revisions:

Estimated time to Complete Revisions (Required)

(Decision Recommendation)

Between 3 and 6 months

Yes

Review #2

1. Evidence, reproducibility and clarity:

Evidence, reproducibility and clarity (Required)

****Summary:**** The authors utilize: 1) chimeric arthritogenic alphaviruses; evolution selection analyses with virus sequences isolated from human patients; and 3) mass spectrometry and proteomics to interrogate determinants of chikungunya virus (CHIKV) permissiveness in primary human macrophages and the human macrophage cell line, THP-1. The authors find that the vaccine strain, CHIKV 181/clone 25 replicates the most efficiently in primary monocyte-derived macrophages compared to

other arthritogenic alphaviruses. Using o'nyong o'nyong (ONNV) as a comparison, the authors generate several chimeric viruses with CHIKV structural proteins and ONNV non-structural proteins (and vice versa) and perform a series of E1 and E2 domain swap experiments. They determine that both CHIKV structural proteins, E2 and E1, are necessary to confer efficient virus production over ONNV in the absence of a difference in viral RNA production. The authors also identify a specific residue in E1 that appears to be important for efficient virus production in THP-1 macrophage cell lines. Finally, using mass spectrometry, the authors identify two host proteins, SPCS3 and eIF3k, that bind to CHIKV E1 structural protein and appear to act as antiviral host factors.

****Major comments:**** The authors elegantly demonstrate that CHIKV structural proteins confer an advantage over ONNV structural proteins in a step in the replication cycle downstream of virus RNA synthesis, possibly virion assembly. This point would be strengthened determining the particle-to-PFU ratio of the parental viruses and the chimeras. Presumably, the ratio would increase in the chimeras containing CHIKV structural proteins. Additionally, the authors should consider performing virion assembly blocking assays with a small molecule inhibitor to determine if this abrogates the virus production advantage of CHIKV structural proteins within the ONNV backbone. Finally, the authors should perform competition experiments with the chimeric viruses and ONNV in macrophages to determine if the chimeras can outcompete the parental ONNV strain. Based on their data, the chimeric viruses should outcompete. These experiments would likely take 3-4 weeks to complete.

The authors use both primary macrophages and macrophage cell lines as their in vitro model system and make one of their major points (listed in the title) that the determinants they identified in the CHIKV structural proteins convert macrophages into dissemination vessels; however, they do not show: 1) an in vivo model that the CHIKV-ONNV chimeras disseminate more efficiently than the parental ONNV; and 2) that these chimeras generate virus more efficiently specifically in macrophages. It would be useful to show that ONNV and CHIKV have equivalent virion production in other cell lines and that the advantage conferred by CHIKV structural proteins in the ONNV backbone is specific to macrophages. The authors should also change their title to reflect that dissemination is not directly being addressed in their study; the implications of their in vitro experimentation in a mammalian host would be more appropriate for the discussion.

OPTIONAL: The authors use CHIKV-ONNV chimeras but it would be interesting to test other chimeras to determine if CHIKV structural proteins confer the same advantage in the backbone of other arthritogenic alphaviruses. The study would also

be strengthened by using a pathogenic strain of CHIKV instead of the vaccine strain, as this is significantly attenuated in vivo.

In Figure 4, the authors identify residues in the CHIKV structural proteins that appear to be under positive selection in human subjects and generate point mutants in these residues with the corresponding ONNV residues. They find that one mutation, V1029I located in E1, completely abolishes virion production in THP-1 macrophage cell lines. However, in their previous chimeric experiments, they find that neither CHIKV E1 or E2 was sufficient to increase virus production in the ONNV backbone. The authors should address this discrepancy, otherwise they should consider moving the data in their point mutation experiments to a supplementary figure. While worthy of reporting, especially given the patient data, these experiments do not buttress the points made in the previous figures.

The authors conclude their manuscript with an assessment of several host proteins, namely SPCS3 and eIF3k, that were identified by mass spectrometry and whose knockdown results in increased virion production. The authors speculate about the role of these proteins but do not provide any mechanistic detail on how they might be playing a role. It is unclear that the putative antiviral role of these proteins involves steps downstream of virus replication, especially given that the authors speculate translation might be affected by eIF3k which, if the case, RNA synthesis should also be expected to be affected.

Overall, while the initial chimeric virus and domain swap approach is strong, the manuscript would benefit with a more thorough examination of virion assembly steps and a mechanistic link to virion production. Otherwise, the authors should revise the structure of their manuscript by de-emphasizing points about virion assembly and leave room for other mechanistic explanations of their chimeric data that more clearly link the host antiviral factor/E1 binding studies.

****Minor comments:**** In Figure 3e, the line under "with CHIKV E1" should be moved over to include the E2-II+E1 virus.

Figure 5a, 5b, and 6a should be replaced with higher resolution images.

2. Significance:

Significance (Required)

Strengths of the study include the initial chimeric virus and domain swap approach to determine factors that allow for the productive replication of chikungunya virus in macrophages compared to other arthritogenic alphaviruses. This approach yielded

useful insights and could be adapted to other viruses. The study is limited, however, by the lack of mechanistic detail linking the antiviral host factors identified which bind to the E1 structural protein, and the advantage conferred by CHIKV structural proteins in the ONNV backbone. The study would be greatly improved by structural studies of the chimeric viruses that directly demonstrate more efficient virion production and that knockdown of the identified factors specifically affects virion production. This point could be addressed either through additional experimentation or tempering of the authors' conclusions about the mechanism by which CHIKV structural proteins provide an advantage over those of ONNV.

The study advances knowledge in the field on what might advantage different pathogenic alphaviruses and explain differences in disease pathology. Additionally, the authors devise a simple and clever strategy that could be applied across different alphaviruses and would be useful to test in vivo in future studies. This study would be useful to a virology-specific audience.

3. How much time do you estimate the authors will need to complete the suggested revisions:

Estimated time to Complete Revisions (Required)

(Decision Recommendation)

Between 1 and 3 months

No

Review #3

1. Evidence, reproducibility and clarity:

Evidence, reproducibility and clarity (Required)

Review: In this manuscript the authors generated macrophages derived from the THP-1 cell line or human peripheral blood mononuclear cells stimulated with MCSF and infected them with alphaviruses some containing GFP expression cassettes. In Figure 1, they demonstrate that CHIKV infected these cells more robustly than RRV, SINV or the related ONNV. The authors generated an extensive array of CHIKV/ONNV chimeras to identify the viral proteins that dictate release from infected macrophages and narrowed it down to the envelop proteins E1 and E2. Fine mapping identified a couple of single mutations that affected macrophage infection outcomes. The authors then shifted their approach to identifying env protein interactors using a myc-tag pulldown methods followed by mass spectrometry. The assay identified a number of proteins including those involved in vesicular transport and interferon pathways. siRNA knockdown experiments were performed to identify interactors and many of them were shown to improve virus output.

Critique: Overall, the manuscript is well written but in its current state it is more like two different stories because the effects of envelop proteins and list of interactors are not brought together in on one story. A possible fix to this problem would be inclusion of ONNV and CHIKV containing env mutations that do and do not restore viral release from macrophages into the pulldown/association experiments shown in Figure 6. The other major issue is the lack of protein data for the viral mutants relative to WT ONNV and CHIKV and assessment of viral RNA in the supernatants to determine whether the block is release or an earlier event since viral RNA levels in the cell seems to be the same or at least normalized. Lastly, knockdown experiments indicate an effect of things like OAS3 or other innate immune modulators. There are no controls to demonstrate that these are specific to CHIKV infection or if knockdown would assist growth of ONNV as well.

****Other points to consider:****

1. The title does not fit the manuscript findings and should be modified.
2. It is unclear why the authors show results for SINV and RRV in Figure 1. Either these should be removed or the viruses should be carried throughout the experiments described in the Figure. Better yet would be to add additional alphaviruses to this analysis to determine if there are additional viruses that act similarly to CHIKV.

3. Is the data presented in Figure 1A significant?
4. The justification for inclusion of Figure 4A is lacking. It is unclear what this panel is supposed to be demonstrating.
5. There is little justification for the candidates assessed in
6. Extended data Figure 3 is very difficult to read due to the small font size.
7. Just to be clear, the blots shown in Figure 6D are different from those depicted in Extended data Figure 4b, because some of them look very similar.

2. Significance:

Significance (Required)

The study provides a fresh look at Alphavirus replication in macrophages. There are a number of issues that should be worked out that would enhance impact and interpretation of this study.

3. How much time do you estimate the authors will need to complete the suggested revisions:

Estimated time to Complete Revisions (Required)

(Decision Recommendation)

Between 1 and 3 months

Yes

Revision Plan

Manuscript number: RC-2023-02023

Corresponding author(s): Melody, Li

[The “revision plan” should delineate the revisions that authors intend to carry out in response to the points raised by the referees. It also provides the authors with the opportunity to explain their view of the paper and of the referee reports.]

The document is important for the editors of affiliate journals when they make a first decision on the transferred manuscript. It will also be useful to readers of the reprint and help them to obtain a balanced view of the paper.

*If you wish to submit a full revision, please use our "Full Revision" template. **It is important to use the appropriate template to clearly inform the editors of your intentions.**]*

1. General Statements [optional]

This section is optional. Insert here any general statements you wish to make about the goal of the study or about the reviews.

Our paper was very favorably reviewed by three expert reviewers at Review Commons, who were highly enthusiastic about our findings.

- **Reviewer #1** said that “*the paper is well written*”, and “*The authors have tackled an intriguing question: why do some alphaviruses infect macrophages and others do not. They have used a chimeric approach to very systematically identify the viral determinants E2 and E1 as being important in macrophage infection...*” and “*if their findings that E1 has a role in attenuating a host antiviral factor, this would be fantastic. More and more examples of viral proteins having multiple roles during infection are in the literature. The idea that structural proteins also attenuate host antivirals is a developing field and vastly understudied. By fleshing out the results some more the authors might be onto something very important in alphavirus virology.*”
- **Reviewer #2** said that “*the initial chimeric virus and domain swap approach is strong*”, “*The authors elegantly demonstrate that CHIKV structural proteins confer an advantage over ONNV structural proteins in a step in the replication cycle downstream of virus RNA synthesis*”, and “*The study advances knowledge in the field on what might advantage different pathogenic alphaviruses and explain differences in disease pathology. Additionally, the authors devise a simple and clever strategy that could be applied across different alphaviruses and would be useful to test in vivo in future studies.*”

Revision Plan

- **Reviewer #3** said that “*the manuscript is well written*”, and “*The study provides a fresh look at alphavirus replication in macrophages.*”

All three reviewers had suggestions to improve the clarity and accuracy of our manuscript, which we have now incorporated into our revision plan. We have performed additional experiments to strengthen the validation of the CHIKV glycoprotein interactors found by AP-MS and better connect the first and second parts of the manuscript. The proposed experiments/experimental results are detailed in the following point-by-point revision plan and briefly summarized here:

- **More accurate structural configuration of E2-E1 heterodimer consistent with previous cryoEM studies:** We addressed this by changing the configuration of E2-E1 heterodimer to present E2 on the exterior of E1, and additionally interrogating the positively selected residues in the complex trimer structure of E2-E1 heterodimers, which shed light on the roles of these residues in the context of heterodimer and trimer formation.
- **Further validation/model on E1 and host factor (SPCS3 and eIF3k) interactions in the absence of E2:** We plan to address this by performing reciprocal immunoprecipitation with CHIKV glycoproteins and immunofluorescence to validate the interaction of E1 with SPCS3/ eIF3k.
- **More thorough investigation of CHIKV structural proteins in the step of post viral RNA synthesis:** We have addressed this by further interrogating the production kinetics of CHIKV, ONNV, and the chimeras containing CHIKV glycoproteins through determining their particle-to-PFU ratios as well as treating infected cells with secretory pathway inhibitors. We found that CHIKV along with the chimeras generates significantly more infectious particles (8- to 20-fold lower in their particle-to-PFU ratios in comparison to ONNV) and is more dependent on the secretory pathway than ONNV.
- **Better connection of the first part on CHIKV determinant identification and the second part on related host factor identification through AP-MS:** To address this, we have already tested the binding of SPCS3 and eIF3k to CHIKV E1 with the positively selected site mutation that causes severe virion production defects in THP-1 derived macrophages, and found reduced viral binding to the host factors, suggesting attenuated E1-host factor binding site. To complement that experiment, we will also determine structural protein expression and processing of parental and E1 mutant CHIKV in eIF3k CRISPR knockout 293T cells to confirm the requirement of E1-eIF3k interaction for proper viral glycoprotein maturation and assembly. In addition, we plan to perform CORUM analysis to identify high confidence functional protein complexes using our 14 hits found in two independent mass spec experiments, which will provide mechanistic insights into how these identified cellular complexes and processes might modulate CHIKV infection.
- **Further validation of the host factors identified by AP-MS in blocking virion production of CHIKV:** We addressed this by evaluating CHIKV, ONNV, and SINV

Revision Plan

infections in 293T cells with CRISPR knockout of eIF3k, confirming its anti-CHIKV specificity.

Revision Plan

2. Description of the planned revisions

All figures referenced here are updated figures unless stated otherwise.

Insert here a point-by-point reply that explains what revisions, additional experimentations and analyses are planned to address the points raised by the referees.

Reviewer 1 major comments:

2. Validation of E1 interaction with SPSC3 and eIF3k needs to be stronger. Some concerns/questions are listed below. A myc tag was inserted between E3 and E2. How efficiently does furin cleave E3 from E2 in this virus and how are viral titers of the myc-tagged virus compared to the non-tagged virus? I ask because is the IP looking at what is being pulled down by E2 or E3-myc-E2 that could be part of the spike polyprotein? The authors found E2 interacts with E3, E1 and a list of other host proteins. These results suggest several interactions including E2-host factor, E2-E1, E2-E3, E2-E1-host factor, E2-E3-E1, E2-E3-host factor. In figure 6d, and the subsequent conclusions, the authors suggest E1 is interacting with the host factor and do not see E2 alone and very low amounts of E3-E2-6K-E1. based on how the IP was performed I am not sure how an interaction between E1 and SPSC3 alone, without E2, would be detected. I would also like to see a reciprocal pull down using E1 and also E2 to see if these host factors are pulled down.

We thank the reviewer for these concerns. Given the low viral protein expression in macrophages (Figure 1A), we need an efficient system to enrich for large amounts of CHIKV glycoproteins for identifying host interactors through mass spectrometry. Adding tag/reporter proteins, such as mCherry, between E3 and E2 have been used to label alphavirus glycoproteins in previous study², which is why we chose to use this myc tag labeling strategy coupled with myc Ab-conjugated agarose beads for AP-MS. However, like reviewer 1 speculated, inserting myc tag between E3 and E2 does attenuate CHIKV infectivity according to the reduced supernatant viral titers of 293T cells transfected with CHIKV/myc-E2 genomic RNA in comparison to those of cells transfected with unmodified CHIKV vaccine strain 181/clone 25 genomic RNA (shown below). Despite the attenuation, CHIKV/myc-E2 harvested from transfected 293T cells still reaches a titer over 10^8 pfu/ml, which allowed us to identify interactors by AP-MS.

We further analyzed the cleavage efficiency of glycoproteins by comparing the expression levels of E3-E2-6K-E1, E3-E2 (p62), E2, and E3 in 293T cells transfected with unmodified CHIKV or CHIKV/myc-E2 genomic RNA. We didn't detect any uncleaved forms of glycoproteins in cells transfected with either unmodified CHIKV or CHIKV/myc-E2 RNA when we probed with E2 antibody. However, probing with E3 antibody prior to longer exposure of the immunoblot showed higher E3-E2-6k-E1 and E3-E2 (p62) levels in cells

Revision Plan

transfected with CHIKV/myc-E2 RNA, suggesting that both mature E2 and E2-containing precursor polyproteins are available to be pulled down. Overall, the expression levels of mature E2 detected by E2 antibody are similar.

We thank reviewer 1 for providing a thorough dissection of all the possible interactions between the identified host factors and cleaved/uncleaved glycoproteins. This is a very interesting question. As reviewer 1 mentioned that E1 usually appears with E2 or E3-E2 in heterodimer forms, we were also surprised to find that E2 does not interact with either of the two host factors. To address this, we plan to conjugate E2 and E1 to protein A/G beads, respectively, for a reciprocal pulldown to validate CHIKV glycoprotein interactions with SPCS3 and eIF3k. Results from this experiment will be included in the fully revised manuscript.

Revision Plan

4. E1 and E2 are thought to interact during polyprotein translation and the initial dimer forms in the ER. If E1 is interacting with SPSC3 in the ER, is E2 also present? Or is a population of E1 not interacting with E2 in order to inhibit SPSC3? I would love a model of how the authors see all these factors coming together for this new role of E1.

We thank Reviewer 1 for proposing this interesting hypothesis. Given the unexpected absence of E2 in our validation of host factor-E1 pulldown, we speculate that a group of free E1 proteins with distinct function is interfering with host factors in the ER, which is a model worth further investigation and discussion. A great example of this is the alphavirus nonstructural protein 3 (nsP3) that plays essential roles in RNA replication, although depending on the alphavirus not all of the nsP3 in the cell colocalizes with dsRNA, suggesting there is a separate distinct pool of nsP3 outside of active viral replication complex that interacts with host factors in these observed larger cytoplasmic aggregates³. To address this, we plan to use laser confocal microscopy to observe the interactions between host factors (SPCS3, eIF3k), and CHIKV E2 and E1. We will include this result as well as our proposed model in the fully revised manuscript.

Reviewer 1 minor comments:

5. How effective is a non-attenuated CHIK strain in infecting macrophages? Could you make a SINV-La Reunion chimeric virus (which is BSL2) to see if a higher percentage of macrophages are infected and is this potentially contributing to the increased pathogenesis of La Reunion? Also how different is 181/25 with a pathogenic strain in the E2 and E1 residues? and compared to ONNV?

We thank Reviewer 1 for this question, which is also raised by Reviewer 2. In order to address this question, we plan to use the virulent CHIKV La Reunion strain to study the infection of THP-1 derived macrophages with non-attenuated CHIKV in BSL-3. We are getting trained in the BSL-3 facility and will soon be certified.

We thank Reviewer 1 for this insightful suggestion on investigating the conservation of these positively selected sites in different strains. We have aligned the sequences of ONNV and CHIKV strains from different lineages, including CHIKV vaccine strain 181/clone 25 and Thai strain AF15561 (the parental strain of CHIKV 181/clone 25). We found that the two positively selected sites with negative effects on virion production, E2-135 and E1-220 (sites 460 and 1029 in original manuscript version), are very conserved in either CHIKV or ONNV strains. CHIKV E2-135 is always valine (V) regardless of the lineages, while ONNV E2-135 is always leucine (L). CHIKV E1-220 is always V, while ONNV E1-220 is always isoleucine (I).

Revision Plan

We also analyzed the amino acid heterogeneity of E2-135 and E1-220 in 397 CHIKV patient sequences from NCBI Virus database. Most of the amino acids at these 2 sites are V. The counts of each amino acid at E2-135 and E1-220 is summarized in table below. This result suggests that valine residues at E2-135 and E1-220 are crucial for CHIKV fitness and strongly selected during viral evolution. The sequence alignment and table will be included and discussed in the fully revised manuscript.

	E2-135	E1-220
Valine (V)	394	392
Alanine (A)	1	3
Methionine (M)	1	0
Glutamic acid (E)	0	1
Glycine (G)	1	0
Isoleucine (I)	0	1

7. The paper is well-written. There is a slight disconnect as the authors go from discussing results in Figure 4 to Figure 5.

We thank Reviewer 1 for the comment regarding the disconnection of the last two figures in this paper which is also shared by the other reviewers. We have taken 3 approaches to address this comment: 1) We performed a pulldown of the host factors (SPCS3, eIF3k) identified in Figure 5 with CHIKV positively selected mutants examined in Figure 4 with deficient virion production. The result is presented in our response to Reviewer 3' s major comment #1, suggesting that the positively selected site in E1 is essential for CHIKV glycoprotein interaction with host factors. 2) To complement our first experiment, we will also determine structural protein expression and processing of parental and E1 mutant CHIKV in eIF3k CRISPR knockout 293T cells. 3) Finally, we plan to perform CORUM

Revision Plan

analysis to identify high confidence functional protein complexes using our 14 hits found in both mass spec experiments, which will provide mechanistic insights into how these identified cellular complexes and processes might modulate CHIKV infection.

Reviewer 2 major comments:

3. Finally, the authors should perform competition experiments with the chimeric viruses and ONNV in macrophages to determine if the chimeras can outcompete the parental ONNV strain. Based on their data, the chimeric viruses should outcompete.

We thank Reviewer 2 for this inspiring suggestion. The competition experiment is an innovative and informative way to evaluate whether CHIKV glycoproteins confer a selective advantage on virion production in THP-1 derived macrophages. We plan to infect THP-1 derived macrophages with ONNV and ONNV/CHIKV E2+E1 and detect the viral glycoproteins secreted in the supernatant by western blot, although there is a possibility that this experiment might not work due to superinfection exclusion. Given that there is no commercial antibody of ONNV available, we need to use tagged viruses for this competition experiment. We constructed ONNV/CHIKV myc-E2+E1 that has a myc tag at the N-terminus of CHIKV E2, and ONNV/HA-E2 that has a HA tag at the N-terminus of ONNV E2. Our first attempt at concentrating the viral progenies released by THP-1 derived macrophages infected with the two tagged viruses has not been successful. We performed sucrose gradient ultracentrifugation of the supernatant viral particles but the myc and HA tags were not detected in the expected sucrose layer. Next, we plan to use myc-Ab and HA-Ab conjugated beads to pull down the supernatant viral particles to detect the ratio of ONNV/CHIKV myc-E2+E1 and ONNV/HA-E2 secreted by THP-1 derived macrophages. This will determine whether ONNV containing CHIKV glycoproteins can outcompete ONNV in co-infected cells due to increased viral fitness.

4. The authors use both primary macrophages and macrophage cell lines as their in vitro model system and make one of their major points (listed in the title) that the determinants they identified in the CHIKV structural proteins convert macrophages into dissemination vessels; however, they do not show: 1) an in vivo model that the CHIKV-ONNV chimeras disseminate more efficiently than the parental ONNV; and 2) that these chimeras generate virus more efficiently specifically in macrophages. It would be useful to show that ONNV and CHIKV have equivalent virion production in other cell lines and that the advantage conferred by CHIKV structural proteins in the ONNV backbone is specific to macrophages. The authors should also change their title to reflect that dissemination is not directly being addressed in their study; the implications of their in vitro experimentation in a mammalian host would be more appropriate for the discussion.

Revision Plan

We acknowledge the limitations of the study, which include a lack of direct demonstration of in vivo dissemination. To address these concerns, we will include further discussion of our in vitro findings in the context of viral dissemination in mammalian hosts in the fully revised manuscript. We are also testing ONNV, CHIKV, Chimera I and ONNV/CHIKV E2+E1 infections in 293T cells to investigate whether the advantage conferred by CHIKV glycoproteins are macrophage specific.

We have also updated the title to accurately reflect the significance of this research: **“Chikungunya virus glycoprotein targeting of host factors increases viral fitness in human macrophage”**.

Reviewer 2 optional comments:

1.The authors use CHIKV-ONNV chimeras but it would be interesting to test other chimeras to determine if CHIKV structural proteins confer the same advantage in the backbone of other arthritogenic alphaviruses. The study would also be strengthened by using a pathogenic strain of CHIKV instead of the vaccine strain, as this is significantly attenuated in vivo.

We thank Reviewer 2 for this suggestion which is also suggested by Reviewer 1 in their minor comment #5. We plan to use virulent CHIKV La reunion strain and carry out infection experiments in BSL-3 to strengthen this study. We are getting trained in the BSL-3 facility and will be certified soon.

Reviewer 3 critique comments:

2.The other major issue is the lack of protein data for the viral mutants relative to WT ONNV and CHIKV and assessment of viral RNA in the supernatants to determine whether the block is release or an earlier event since viral RNA levels in the cell seems to be the same or at least normalized.

We thank Reviewer 3 for pointing out the insufficient clarification of the block leading to defective CHIKV mutant virion production. We previously detected E2 expression from 293T cells transfected with poly-glycoproteins (E3-myc-E2-6K-E1) containing E2-V135L (V460L in original manuscript version), E2-A164T (A489T in original manuscript version), E2-A246S (A571S in original manuscript version) and E1-V220I (V1029I in original manuscript version). We found that only E2-V135L mutation can lead to unexpected E2 cleavage (arrow pointed, shown below) as we mentioned but not shown in the original

Revision Plan

manuscript. This explains why E2-V135L mutation attenuates infectious CHIKV production.

The E2 expression of E1-V220I appears to be not affected in 293T cells transfected with poly-glycoproteins with E1-V220I (shown below). In addition, the E1-host factor binding result in our response to Reviewer 3's major comment #1 showed that E1 with the positively selected site mutation V220I can also be successfully expressed in 293T cells after transfection with poly-glycoprotein. Based on these current data, E1-V220I mutation likely abrogates virion production without affecting glycoprotein expression.

Our previous result of the ONNV particle-to-PFU ratio reveals that ONNV RNA is released but encapsidated in defective particles causing its attenuation in infected macrophages. Thus, even though the glycoproteins of E1-V220I can be expressed, the diminished virion production of CHIKV E1-V220I can still be ascribed to 1) blocked viral particle release and 2) production of defective particles like ONNV. Given that it is not feasible to obtain particle-to-PFU ratio of E1-V220I mutant which fails to form plaques, Reviewer 3's suggestion to assess the supernatant viral RNA will be a nice approach to address this question. To further address this concern, we plan to transfect THP-1 derived macrophages with CHIKV E1-V220I mutant RNA to detect the intracellular viral glycoprotein expression and supernatant viral RNA levels through western blot and TaqMan assay, respectively.

Reviewer 3 minor comments:

6. Extended data Figure 3 is very difficult to read due to the small font size.

Revision Plan

We apologize for the small font in Extended data Figure 3. We plan to replace Figure EV3 (Extended data 3 in unrevised version) with a CORUM protein-protein interaction network that centers on the significant hits identified by both AP-MS experiments, but includes hits from either one of the two experiments in these functional protein complexes. The figure will be more concise and centralized, and the font will be bigger.

Revision Plan

3. Description of the revisions that have already been incorporated in the transferred manuscript

Content changes in manuscript are highlighted in yellow.

Please insert a point-by-point reply describing the revisions that were already carried out and included in the transferred manuscript. If no revisions have been carried out yet, please leave this section empty.

Reviewer 1 major comments:

1. The authors show one configuration of the E1-E2 heterodimer in Figure 4d. As shown, the E1 protein is exterior to the E2 protein and would suggest E1 is on the surface on the spike complex and virus surface. However, another configuration of the glycoproteins has E2 on the exterior of E1 and also on the exterior of the virus. The latter conformation is what has been observed in cryoEM studies of alphaviruses. The first configuration represents the E1-E2 between the three heterodimers which are important for spike assembly. The reason the orientation of the E2-E1 dimer is important is the authors speculate on the importance of the 6 CHIK residues not found in ONNV based on the structure, but the structural interpretation is, in my opinion, not correct.

We thank reviewer 1 for pointing out the correct E2-E1 heterodimer configuration. To address this, we corrected the position of E2 and E1 in Figure 4 based on previous cryoEM study¹, keeping E2 always on the exterior in the E2-E1 heterodimer. We also replaced the Indian Ocean Lineage (IOL) E2-E1 structure¹ in the original Figure 4 with the CHIKV 181/clone 25 structure which was recently analyzed by Katherine Basore et al.². In a single E2-E1 heterodimer, all six unique CHIKV positive selection sites are located on the outside of the structure after correcting the configuration. In addition, we investigated two of the unique CHIKV positively selected sites that are important for virion production, E2-V135 (V460 in the original manuscript version) and E1-V220 (V1029 in the original manuscript version), in trimerized structure of E2-E1 heterodimers. We found that the E2-V135 and E1-V220 residues in one heterodimer are facing E2 of the neighboring heterodimer on either side. Interestingly, while V135 is embedded between the E2 proteins of two different heterodimers, E1-V220 is partially embedded by E1 and the neighboring E2 and partially exposed to the outside. This suggests that even though both E2-V135 and E1-V220 might be crucial for CHIKV E2-E1 trimerization, E1-V220 provides an additional docking site for host factor interactions. We thank review 1 again for this important comment leading to these new findings. We have updated **Figure 4F-4G** and the corresponding result section (**lines 201-209**) in this partially revised manuscript.

Revision Plan

Figure 4F

Figure 4G

Reviewer 1 minor comments:

1. In Figure 1c, (-) RNA is shown but in the rest of the figures (+) RNA is shown. Show both or select one. I do find it interesting the (-) RNA levels are similar over time, even at 4 hours post transfection (early time). Related to this, ONNV has higher levels of (-) RNA but what is known about structural protein levels in ONNV and CHIK in macrophages? Are there comparable levels of CP and GP being produced?

We thank Reviewer 1 for this comment. The (-) RNA is synthesized before the synthesis of subgenomic mRNA and therefore can reflect more accurately early viral replication and nonstructural protein functions. This is the reason why we consider the (-) RNA levels evaluated by specific nsP1 TaqMan probes to be more appropriate for determining early stage differences between ONNV and CHIKV replication in Figure 1 as the goal of that

Revision Plan

figure is to define the steps in CHIKV life cycle that are more efficient than those of ONNV in THP-1 derived macrophages. On the other hand, the (+) RNA evaluated by E1 primers that we used in the later figures monitors viral RNA synthesis over time in the reflection of genomic (+) RNA and subgenomic mRNA transcribed from (-) RNA templates. Similar levels of (+) RNA and contrasting virion titers really point the difference to the later stages of subgenomic mRNA translation, viral glycoprotein secretion, and assembly.

We have generated ONNV/myc-E2 reporter virus and assessed viral glycoprotein expression through flow cytometry using a FITC-conjugated anti-myc antibody in the THP-1 derived macrophages transfected with CHIKV/myc-E2 and ONNV/myc-E2 (shown below). The results show that the expression of ONNV glycoproteins is more inhibited than that of CHIKV glycoproteins, though both of their expression levels in macrophages seem to be suppressed. Since there is no commercial ONNV antibody available, we were unable to compare capsid expression levels between the two viruses. Overall, differences in the myc-tagged glycoprotein expression levels of the two viruses reveals ONNV defect in either structural protein translation or glycoprotein maturation.

2. Figure 2e and figure 3 have ONNV has the first bar followed by CHIK. In figure 1 and 2b, CHIK is first and then ONNV. helps the reader to have the controls in the same order.

We thank Reviewer 1 for this suggestion. We have changed the order of ONNV and CHIKV bars in figure 2E and figure 3 so the CHIKV bar consistently comes first in all the figures.

Revision Plan

3. Line 143-145 the authors discuss that when ONNV is the backbone and CHIK proteins are inserted the infection is more attenuated because of the E2 and E1 are from CHIK and ONNV, not the same virus (could also be E2-CP interactions are disrupted). However the chimeras made with the CHIK backbone (in Figure 2) have a mismatch between E2 and E1 as well.

We thank Reviewer 1 for this informative comment. We agree that the incompatible E2-E1 heterodimer formation may not be the only reason that causes attenuation of ONNV/CHIKV E1 and ONNV/CHIKV E2. There may be multiple factors contributing to the fitness of the chimeras, which requires more in-depth mechanistic investigations and is out of the scope of this study. We have now removed the explanation “potentially due to incompatible heterodimer formation between ONNV E2 and CHIKV E1” in **line 144**.

4. When discussing the residues that were found in the FEL and MEME analysis, the authors start the amino acid numbering from CP and continue along the polyprotein. Usually when discussing amino acids in the structural proteins, each protein starts at amino acid 1. So V460 would be E2-V135. It would also be useful to know what the residues in ONNV were at these positions to see if amino acids changed in charge, size, bond forming potential, etc. Showing these residues in the E2-E1 conformation found in the virion would also allow one to find adjacent residues that could explain differences in spike assembly and potentially where/how E1 is binding to a host protein.

We thank Reviewer 1 for this comment. We revised the amino acid numbers in the manuscript to start from the beginning of each structural protein. To look more into these residues in ONNV, we aligned CHIKV and ONNV from different lineages and compared the 6 positively selected sites (refer to our response to Reviewer 1’s minor comment #5). We found that E2-135 and E1-220 which are essential for CHIKV production are valines in all the aligned CHIKV strains. For the aligned ONNV strains, E2-135 are all leucines and E1-220 are all isoleucines. While valine, leucine and isoleucine are all amino acids with hydrophobic side chains, valine has the shortest side chain. The length of the side chains may lead to different hydrophobic properties that affect protein folding, which warrants further structural analysis.

6. When describing the last results section, "CHIKV E1 binding proteins exhibit potent anti-CHIV activities" the authors use macrophages. In the rest of the text they consistently use THP-1 macrophages or human primary monocyte derived macrophages. The details of the cell type are extremely useful to the reader and having those in the last results section would be great.

Revision Plan

We thank Reviewer 1 for pointing out the importance of cell type clarification in the last results section. We now consistently use “THP-1 derived macrophages” instead of “macrophages” in this section.

Reviewer 2’s major comments:

1. The authors elegantly demonstrate that CHIKV structural proteins confer an advantage over ONNV structural proteins in a step in the replication cycle downstream of virus RNA synthesis, possibly virion assembly. This point would be strengthened determining the particle-to-PFU ratio of the parental viruses and the chimeras. Presumably, the ratio would increase in the chimeras containing CHIKV structural proteins.

We thank Reviewer 2 for this comment. We agree that determining particle-to-PFU ratios of parental and chimeric viruses will strengthen this study. To obtain the particle-to-PFU ratio, we infected THP-1 derived macrophages with CHIKV, ONNV and chimeras containing CHIKV glycoproteins (Chimera I, and ONNV/CHIKV E2+E1) for 24 h. To quantify the secreted viral particles, we extracted viral RNA in the supernatant and detected (+) viral RNA through TaqMan assay with specific nsp1 probes. The released infectious virions were evaluated through plaque assay. The particle-to-PFU ratios are summarized in the table below. The results show that ONNV has the highest particle-to-PFU ratio (41398), suggesting defective ONNV genome encapsidated in particles leading to defective virion production. On the other hand, the particle-to-PFU ratio of CHIKV (747) is 55-fold lower than that of ONNV. Replacing E3-E2-6K-E1 of ONNV with CHIKV homologous proteins reduces the particle-to-PFU ratio by 8 fold to 4875. Replacing E2 and E1 of ONNV with the ones from CHIKV (ONNV/CHIKV E2+E1) reduces the particle-to-pfu ratio by 20 fold to 2017, suggesting that CHIKV glycoproteins enhance the infectivity of viral progenies produced by THP-1 derived macrophages. We have included the results in **Figure 3D-3E** in our partially revised manuscript and described in **lines 149-158**.

Revision Plan

Virus	Released particles (copy number)/ ml	Titer (pfu/ml)	Particle-to-PFU ratio (average)
ONNV	2.06E+10	4.50E+05	41398
	2.22E+10	6.00E+05	
CHIKV	1.27E+10	2.50E+07	747
	9.86E+09	1.00E+07	
Chimera I	2.28E+10	5.00E+06	4875
	1.82E+10	3.50E+06	
ONNV/CHIKV E2+E1	1.00E+11	4.30E+07	2017
	6.84E+10	4.00E+07	

2. Additionally, the authors should consider performing virion assembly blocking assays with a small molecule inhibitor to determine if this abrogates the virus production advantage of CHIKV structural proteins within the ONNV backbone.

We thank Reviewer 2 for this insightful comment. As the secretory pathway is commonly important for alphavirus glycoprotein maturation and assembly, it will be informative to interrogate CHIKV glycoprotein trafficking and assembly through this pathway using specific inhibitors, such as dihydropyridine FLI-06 and golgicide A. Golgicide A is a reversible inhibitor of the cis-Golgi GBF1, which leads to rapid disassembly of the Golgi and trans-Golgi network (TGN)⁴. FLI-06 is a new inhibitor that interferes with cargo recruitment to ER-exit sites and disrupts Golgi without depolymerizing microtubules or interfering GBF1⁵. We pretreated THP-1 derived macrophages with 10 μ M FLI-06 or golgicide A for 30 mins prior to infection with CHIKV, ONNV, Chimera I, or ONNV/ CHIKV E2+E1. After 1 hour of virus adsorption in PBS with 1% FBS in the absence of the inhibitors, the cells were treated with the inhibitors at the same concentration (10 μ M) in complete medium for 24 h. The plaque assay result shows that all the viruses are sensitive to secretory pathway inhibition, however, the production of viruses containing CHIKV glycoproteins is significantly more attenuated by FLI-06 and golgicide A. This suggests that CHIKV glycoproteins-mediated trafficking and assembly is more heavily dependent on the host secretory pathway. We will include this result in the fully revised manuscript.

Revision Plan

Reviewer 2 optional comments:

2. In Figure 4, the authors identify residues in the CHIKV structural proteins that appear to be under positive selection in human subjects and generate point mutants in these residues with the corresponding ONNV residues. They find that one mutation, V1029I located in E1, completely abolishes virion production in THP-1 macrophage cell lines. However, in their previous chimeric experiments, they find that neither CHIKV E1 or E2 was sufficient to increase virus production in the ONNV backbone. The authors should address this discrepancy, otherwise they should consider moving the data in their point mutation experiments to a supplementary figure. While worthy of reporting, especially given the patient data, these experiments do not buttress the points made in the previous figures.

We thank Reviewer 2 for this insightful comment. According to previous studies, E2 and E1 always interact with each other from the step of the formation of single heterodimer in the ER to heterodimer trimerization before viral particle assembly. Although the E1-V220 site (previously called V1029) on the exterior of a single E2-E1 heterodimer appears to not be engaged in the E2-E1 interaction E1-V220 is partially exposed and protruding into the groove formed by E1 and the E2 of neighboring heterodimer, accessible to host

Revision Plan

factors. As such, mutating CHIKV E1-V220 to the ONNV residue (E1-V220I) may not only disrupt E2-E1 trimerization but also interfere viral glycoprotein interaction with host factors (presented in our response to Reviewer 1's major comment #1). Similarly, solely swapping E2 or E1 with CHIKV substitute in the ONNV backbone would also affect the interaction between neighboring E2 and E1 in trimerized spike, which may explain why neither ONNV/CHIKV E2 or ONNV/CHIKV E1 rescues virion production in THP-1 derived macrophages. We have included this in the partially revised discussion section **lines 296-313**.

3. The authors conclude their manuscript with an assessment of several host proteins, namely SPCS3 and eIF3k, that were identified by mass spectrometry and whose knockdown results in increased virion production. The authors speculate about the role of these proteins but do not provide any mechanistic detail on how they might be playing a role. It is unclear that the putative antiviral role of these proteins involves steps downstream of virus replication, especially given that the authors speculate translation might be affected by eIF3k which, if the case, RNA synthesis should also be expected to be affected.

We thank Reviewer 2 for this comment. We acknowledge that we have yet a full mechanistic understanding of how SPCS3 and eIF3k impact virion production. We plan to investigate their antiviral roles in our follow-up studies. For our partial revision, we have constructed several single eIF3k knockout (KO) clones of 293T cells. The eIF3k sgRNA we designed targets exon 3 which would eliminate expression of all 3 splice isoforms of eIF3k (see below for KO schematic and sequence verification of CRISPR KO). Unfortunately, we failed to obtain single clones of 293T cells with SPCS3 complete KO, consistent with a previous study by Rong Zhang et al⁶ that were unable to recover SPCS3 KO clones likely due to the importance of SPCS3 in cell survival. We infected an eIF3k KO clone (clone 9) with CHIKV vaccine strain 181/clone 25, ONNV SG650, and SINV Toto1101. Interestingly, we found that the antiviral activity of eIF3k is specific to CHIKV as CRISPR KO of eIF3k increases CHIKV production by 2.5 fold but not ONNV or SINV production (shown below). We have included this in the partially revised manuscript in **line 272-282 (Figure 7B-7D)**.

We presume that Reviewer 2's inference of eIF3k's potential effects on viral RNA synthesis is based on our speculation of its antiviral role in viral translation, which may affect viral nonstructural gene expression. We would like to clarify that eIF3k is not an initiation factor traditionally needed for cap-dependent translation. It is also not clear what translation process (nonstructural polyprotein translation from viral genomic RNA or structural polyprotein translation from viral subgenomic mRNA) involves eIF3k if it indeed

Revision Plan

affects viral protein expression. Notably, previous SINV studies imply that alphavirus structural polyprotein translation may employ unique mechanisms without the requirement of several crucial initiation factors^{4,5}. It will be interesting to see whether eIF3k participates in viral subgenomic mRNA translation as that would affect viral glycoprotein expression leading to reduced virion production. We have now included additional discussion on eIF3k antiviral mechanisms in the partially revised manuscript in **lines 345-353**.

Figure 7B

Figure 7C

Wild-type	CACCCTGTGCAAGTGCATGATCGA-----CCAGGCACATGTATCCTTCCAGCACT
KO allele 1	CACCCTGTGCAAGTGCATGTGCAAGTGCAGGATACCAGGCACATGTATCCTTCCAGCACT
KO allele 2	CACCCTGT-----A-----CCAGGCACATGTATCCTTCCAGCACT

Figure 7D

Revision Plan

4. Overall, while the initial chimeric virus and domain swap approach is strong, the manuscript would benefit with a more thorough examination of virion assembly steps and a mechanistic link to virion production. Otherwise, the authors should revise the structure of their manuscript by de-emphasizing points about virion assembly and leave room for other mechanistic explanations of their chimeric data that more clearly link the host antiviral factor/E1 binding studies.

We thank the reviewer for these positive comments and suggestions. We have addressed this by further interrogating the production kinetics of CHIKV, ONNV, and the chimeras containing CHIKV glycoproteins through determining their particle-to-PFU ratios as well as treating infected cells with secretory pathway inhibitors (refer to our responses to Reviewer 2 major comments #1 and #2). We have also included additional discussion on eIF3k antiviral mechanisms specifically on how it may affect other steps of the viral life cycle in the partially revised manuscript in **lines 345-353** (refer to our response to Reviewer 2 optional comment #3).

Reviewer 3's critique comments:

1. Overall, the manuscript is well written but in its current state it is more like two different stories because the effects of envelope proteins and list of interactors are not brought together in one story. A possible fix to this problem would be inclusion of ONNV and CHIKV containing env mutations that do and do not restore viral release from macrophages into the pulldown/association experiments shown in Figure 6.

We thank Reviewer 3 for the insightful suggestions to better connect the first (CHIKV determinants) and second (CHIKV glycoprotein interactors) parts of the manuscript. In response to the Reviewer's comment, we tested the binding of SPCS3 and eIF3k to CHIKV E1 with E1-V220I (V1029I in original manuscript version) mutation (shown below) which was shown to abrogate virion production in THP-1 derived macrophages in Figure 4E. We transfected plasmids expressing 3XFLAG-tagged SPCS3/eIF3k or empty vector for 24 h followed by transfection with plasmids expressing either the parental CHIKV vaccine strain 181/clone 25 poly-glycoproteins (E3-myc-E2-6K-E1) or poly-glycoproteins with the E1-V220I mutation. Interestingly, we found that mutating CHIKV E1-V220 to the homologous ONNV residue reduces the binding to either SPCS3 or eIF3k. This result strongly suggests that the positively selected E1-V220 is located in the interaction interface between E1 and SPCS3/eIF3k, confirming the genetic conflict between E1 and these host factors to be one of the major drivers of CHIKV evolution observed at site E1-V220. We have included this result in partially revised manuscript in **Figure 7A** and in **lines 265-271**.

Figure 7A

3. Lastly, knockdown experiments indicate an effect of things like OAS3 or other innate immune modulators. There are no controls to demonstrate that these are specific to CHIKV infection or if knockdown would assist growth of ONNV as well.

We also thank Reviewer 3 for the suggestion to check whether the identified host factors specifically target CHIKV or inhibit the infection of ONNV as well. We previously tried but were facing some issues. Since only a small fraction of macrophages can be infected with CHIKV and even a smaller fraction can be infected with ONNV (Figure 1A), it is hard to elucidate the roles of these identified host factors in ONNV infection by siRNA knockdown. We decided to take a more rigorous approach to investigate the antiviral specificity of identified host factors, especially understudied SPCS3 and eIF3k, to different alphaviruses by generating complete knockout 293T single cell clones. Despite the fact that we did not successfully generate SPCS3 complete KO, we obtained an eIF3k KO

Revision Plan

single cell clone and infected it with CHIKV, ONNV and SINV (refer to our response to Reviewer 2 optional comment #3). We found that eIF3k only has antiviral activity against CHIKV with almost no effects on ONNV or SINV infection. We have included this in our partially revised manuscript in **line 272-282 (Figure 7B-7D)**.

Reviewer 3 minor comments:

1. The title does not fit the manuscript findings and should be modified.

We thank Reviewer 3 for this important comment, which was also brought up by Reviewer 2. We have now changed our title to **“Chikungunya virus glycoprotein targeting of host factors increases viral fitness in human macrophage”**, which more accurately reflects the significance of our research.

3. Is the data presented in Figure 1A significant?

We thank Reviewer 3 for this question. We infected both THP-1 derived macrophages and primary monocyte derived macrophages with EGFP-expressing alphaviruses each in duplicates for two independent times. The general low expression of EGFP in all virus-infected groups refrains us from drawing conclusions based on statistically significant differences observed with MFI, hence we chose to show representative scatter plots in the original manuscript. To address Reviewers 3's question, we plotted the infected cell (EGFP+) based on the percentages of the experimental duplicates (shown below), and found CHIKV infection to be the most significantly different from that of the other alphaviruses in primary monocyte derived macrophage. The numbers above the bar charts are the mean percentages of EGFP+ cells.

Revision Plan

4. The justification for inclusion of Figure 4A is lacking. It is unclear what this panel is supposed to be demonstrating.

This is an excellent suggestion as the host factors identified by AP-MS not only contain interactors of CHIKV mature E2 but also those of uncleaved E2-containing precursor polyproteins. We modified Figure 4A to reflect all E2/E2-containing poly-glycoproteins present in CHIKV-infected cells (shown below).

5. There is little justification for the candidates assessed in

We understand Reviewer 3's concern. Due to the nature of mass spectrometry studies which predict protein-protein interactions rather than direct functional validation, we acknowledge that we may miss some host candidates that have anti- or pro-CHIKV activities. Although justification of hit selection from mass spectrometry datasets is more difficult than that from CRISPR KO screen datasets, we set up specific criteria to identify host protein candidates with the greatest potential to functionally interact with CHIKV glycoproteins. Most of the proteins we chose to validate (Figure 6a) were identified in both of our independent AP-MS experiments, which both pass through a P-value threshold of 0.05 and log2 fold change of 0.

7. Just to be clear, the blots shown in Figure 6D are different from those depicted in Extended data Figure 4b, because some of them look very similar.

We thank Reviewer 3 for this question. In Figure 6D, we expressed CHIKV glycoproteins through transfecting CHIKV genomic RNA into 293T cells, while, in Figure 4B, we expressed CHIKV glycoproteins through transfecting poly-glycoprotein plasmid (pcDNA3.1-E3-myc-E2-6K-E1) into 293T cells, which are complementary approaches to express CHIKV glycoproteins to validate their interactions with identified host factors. We have now added schematics to illustrate the different experimental strategies above the figures in this partially revised manuscript (shown below).

Figure 6D

Figure EV4B

4. Description of analyses that authors prefer not to carry out

Please include a point-by-point response explaining why some of the requested data or additional analyses might not be necessary or cannot be provided within the scope of a revision. This can be due to time or resource limitations or in case of disagreement about the necessity of such additional data given the scope of the study. Please leave empty if not applicable.

Reviewer 1 major comments:

3.If CHIK E1 is interacting with the host factors and that is antagonizing the antiviral response of SPSC3 (as one example), then what do pull downs using ONNV structural proteins look like? One would expect reduced interactions because the different amino acid causes a different E2-E1 dimer or attenuates the E1-host factor binding site.

We thank Reviewer 1 for this insightful suggestion. We agree that it would be informative to examine the interactions between ONNV glycoproteins and identified host factors (SPCS3 and eIF3k). Unfortunately, there is no commercial ONNV glycoprotein antibody available making this experiment unfeasible. Interestingly, we did observe reduced interactions between the host factors SPSC3 and eIF3k and the CHIKV E1-V220I mutant (V1029I in original manuscript version) where the positively selected site in E1 was mutated to the homologous ONNV residue (please refer to our response to Reviewer 3's major comment #1). This result suggests that the ONNV glycoproteins likely have an attenuated E1-host factor binding site as the reviewer speculated.

Reviewer 3 minor comments:

2. It is unclear why the authors show results for SINV and RRV in Figure 1. Either these should be removed or the viruses should be carried throughout the experiments described in the Figure. Better yet would be to add additional alphaviruses to this analysis to determine if there are additional viruses that act similarly to CHIKV.

We apologize for the confusion caused by including SINV and RRV results in Figure 1. We intended to show the superiority of CHIKV in infecting primary monocyte derived macrophages among arthritogenic alphaviruses, which we speculate may provide the molecular basis for macrophage-mediated CHIKV dissemination and disease. We would like to keep the SINV and RRV infection results in Figure 1 to highlight the relative susceptibility of macrophages to CHIKV. To echo the additional alphaviruses tested in Figure 1 and bring the story full circle, we included the result of SINV infection of eIF3k CRISPR KO 293T cells in Figure 7B-7D. These results uncover inhibitory activities of eIF3k that are specific to CHIKV.

Revision Plan

References cited in the revision plan:

1. Voss, J. E. *et al.* Glycoprotein organization of Chikungunya virus particles revealed by X-ray crystallography. *Nature* **468**, 709–712 (2010).
2. Jose, J., Tang, J., Taylor, A. B., Baker, T. S. & Kuhn, R. J. Fluorescent Protein-Tagged Sindbis Virus E2 Glycoprotein Allows Single Particle Analysis of Virus Budding from Live Cells. *Viruses* **7**, 6182–6199 (2015).
3. Götte, B., Liu, L. & McInerney, G. M. The Enigmatic Alphavirus Non-Structural Protein 3 (nsP3) Revealing Its Secrets at Last. *Viruses* **10**, 105 (2018).
4. Saenz, J. B. *et al.* Golgicide A reveals essential roles for GBF1 in Golgi assembly and function. *Nat. Chem. Biol.* **5**, 157–165 (2009).
5. Krämer, A. *et al.* Small molecules intercept Notch signaling and the early secretory pathway. *Nat. Chem. Biol.* **9**, 731–738 (2013).
6. Zhang, R. *et al.* A CRISPR screen defines a signal peptide processing pathway required by flaviviruses. *Nature* **535**, 164–168 (2016).

Dear Prof. Li,

Thank you for submitting your manuscript (EMBOJ-2023-115683) to The EMBO Journal from Review Commons, along with the reports of the referees who evaluated it. I have now read the manuscript and the reports carefully and discussed them with the other members of our editorial team. In addition, I have also consulted an expert external advisor regarding the potential suitability of your manuscript for our journal. I regret to say that our conclusion was that we cannot offer publication in The EMBO Journal.

We acknowledge that in this study you use evolutionary selection analyses to construct chimeric alphaviruses and identify specific domains of two chikungunya virus glycoproteins that are involved in efficient virion production in infected macrophages. You further provide evidence indicating that one of these viral glycoproteins binds to host factors that exhibit inhibitory activities against production of the virus. Your study suggests that viral structural proteins can evolve to counteract host restriction factors, which we recognize that will be of interest to the virology community.

We also note, however, that the understanding of the underlying mechanisms is rather limited, which was raised both by the referees and our additional advisor. Notably, it remains unclear if and how the viral glycoproteins antagonize the host restriction factors. In addition, the presented data do not explain how eIF3k functions as an antiviral factor. Taking into account these considerations as well as the remaining concerns raised by the referees and the input from our advisor, we think that the advance provided is not sufficiently striking for publication in The EMBO Journal, which normally publishes papers that provide in-depth novel mechanistic understanding of biological phenomena. We have therefore decided not to consider the manuscript further.

I am sorry to have to disappoint you on this occasion, but I would like to thank you for your interest in our journal and for the opportunity to consider your manuscript. I wish you every success with the publication of these results in a more suitable journal.

Yours sincerely,

Rev_Com_number: RC-2023-02023

New_manu_number: EMBOJ-2023-115683

Corr_author: Li

Title: Chikungunya virus glycoprotein targeting of host factors increases viral fitness in human macrophage.

Full Revision

Resubmission: Previous ID: EMBOJ-2023-115683

Corresponding author(s): Melody Li

[Please use this template only if the submitted manuscript should be considered by the affiliate journal as a full revision in response to the points raised by the reviewers.]

*If you wish to submit a preliminary revision with a revision plan, please use our "Revision Plan" template. **It is important to use the appropriate template to clearly inform the editors of your intentions.**]*

1. General Statements [optional]

This section is optional. Insert here any general statements you wish to make about the goal of the study or about the reviews.

Our paper was very favorably reviewed by three expert reviewers at Review Commons, who were highly enthusiastic about our findings.

- **Reviewer #1:** *“The authors have tackled an intriguing question: why do some alphaviruses infect macrophages and others do not...The idea that structural proteins also attenuate host antivirals is a developing field and vastly understudied. By fleshing out the results some more the authors might be onto something very important in alphavirus virology.”*
- **Reviewer #2:** *“The study advances knowledge in the field on what might advantage different pathogenic alphaviruses and explain differences in disease pathology. Additionally, the authors devise a simple and clever strategy that could be applied across different alphaviruses and would be useful to test in vivo in future studies.”*
- **Reviewer #3:** *“The manuscript is well written”, and “The study provides a fresh look at alphavirus replication in macrophages.”*

All three reviewers had suggestions to improve the clarity and accuracy of our manuscript. Both the Editor and Reviewer 2 also commented that additional mechanistic insights on the host restriction factors would add to the study, specifically how the viral glycoproteins interact with/antagonize the host restriction factors, and how eIF3k functions as an antiviral factor. We have now addressed and incorporated all the comments, which have significantly improved the manuscript. We performed several experiments to strengthen the original findings, connect the interactions between the viral E1 glycoprotein and macrophage factors with the positive selection site, and elucidate the antiviral mechanisms of newly identified host restriction factors, which are presented in two entirely new figures (Figures 7 and 8). This significant body of work has been included in the fully revised manuscript and is detailed in the following point-by-point revision plan and briefly summarized here:

- **Enhanced infection of macrophages with the BSL-3 virulent strains of CHIKV:** We validated that the pathogenic strains of CHIKV (La Reunion and Asian strains) also infect THP-1 derived macrophages significantly more efficiently compared to ONNV strain SG650.
- **More accurate structural configuration of E2-E1 heterodimer indicates evolutionary selection sites are important for E2-E1 trimer assembly:**
 1. We replaced the original E2-E1 heterodimer structure (CHIKV Indian Ocean Lineage) with the most current E2-E1 structure (CHIKV vaccine strain 181/clone 25) reported by Basore et al¹.
 2. We kept the E2-E1 heterodimer configuration consistent with the observation from previous cryoEM studies to present E2 on the exterior of E1.
 3. We showed that E2-135 and E1-220 are in critical sites for E2-E1 trimer formation.
- **Validation of viral E1 glycoprotein interactions with host factors (SPCS3 and eIF3k) in the absence of E2:**
 1. We confirmed SPCS3 and eIF3k specifically bind to E1 without engagement of E2 through reciprocal IP.
 2. Use of confocal laser scanning with Airyscan detector to increase resolution showed that E2 is predominantly localized to the plasma membrane, while E1 is more localized to the cytoplasm.
- **More thorough investigation of CHIKV structural proteins in the step of post viral RNA synthesis in macrophages:**
 1. We showed that CHIKV and the chimeras containing CHIKV glycoproteins have lower particle-to-PFU ratios (by 8- to 20-fold lower) in comparison to ONNV when infecting macrophages.
 2. The production of CHIKV and the chimeras containing CHIKV glycoproteins is more sensitive to secretory pathway inhibitors, suggesting their greater dependence on the host secretory pathway.
- **Solidifying the role of CHIKV glycoprotein in virion production to the evolutionary arms race with SPCS3 and eIF3k:** Mutation of CHIKV E1-220 into the ONNV homologous residue (E1-V220I) that destroys CHIKV production in macrophages significantly attenuates E1 interaction with SPCS3/eIF3k. This suggests that E1-V220I is evolutionarily selected by antiviral host factors such as SPCS3 and eIF3k.
- **Deepening the understanding of eIF3k anti-CHIKV mechanism:**
 1. We showed translocation of eIF3k from the nucleus to cytoplasm and increased colocalization of eIF3k and SPCS3 upon CHIKV infection through Airyscan microscopy.
 2. We generated eIF3k CRISPR KO cells lines and confirmed anti-CHIKV specificity of eIF3k.

3. We confirmed that the anti-CHIKV activity of eIF3k is translation-independent with no effects on viral nonstructural and structural protein expression.
 4. We showed that the HAM domain of eIF3k mediates its antiviral activity.
- **Highlighting the key protein complexes that interact with CHIKV glycoproteins through CORUM analysis:** We refined our CORUM network analysis to emphasize high confidence functional protein complexes using the 14 hits found in our two independent mass spec experiments, further supporting the involvement of SPCS3- and eIF3k-containing complexes in CHIKV infection.

This section is mandatory. Please insert a point-by-point reply describing the revisions that were already carried out and included in the transferred manuscript.

Reviewer 1 major comments:

1. The authors show one configuration of the E1-E2 heterodimer in Figure 4d. As shown, the E1 protein is exterior to the E2 protein and would suggest E1 is on the surface on the spike complex and virus surface. However, another configuration of the glycoproteins has E2 on the exterior of E1 and also on the exterior of the virus. The latter conformation is what has been observed in cryoEM studies of alphaviruses. The first configuration represents the E1-E2 between the three heterodimers which are important for spike assembly. The reason the orientation of the E2-E1 dimer is important is the authors speculate on the importance of the 6 CHIK residues not found in ONNV based on the structure, but the structural interpretation is, in my opinion, not correct.

We thank Reviewer 1 for this excellent comment on pointing out the correct E2-E1 heterodimer configuration. To address this, we corrected the position of E2 and E1 in Figure 4 based on previous cryoEM study¹, keeping E2 consistently on the exterior in the E2-E1 heterodimer. We also replaced the Indian Ocean Lineage (IOL) E2-E1 structure² in the original Figure 4 with the CHIKV vaccine strain 181/clone 25 structure, which was recently analyzed by Katherine Basore et al.². In a single E2-E1 heterodimer, all six unique CHIKV positively selected sites are located on the outside of the structure after correcting the configuration. In addition, we investigated two of the unique CHIKV positively selected sites that are important for virion production, E2-V135 (V460 in the original manuscript version) and E1-V220 (V1029 in the original manuscript version), in trimerized structure of E2-E1 heterodimers. We found that the E2-V135 and E1-V220 residues in one

heterodimer are facing E2 of the neighboring heterodimer on either side. Interestingly, while V135 is embedded between the E2 proteins of two different heterodimers, E1-V220 is partially embedded by E1 and the neighboring E2 and partially exposed to the outside. This suggests that even though both E2-V135 and E1-V220 might be crucial for CHIKV E2-E1 trimerization, E1-V220 provides an additional docking site for host factor interactions. We thank review 1 again for this important comment leading to these new findings. We have updated **Figure 4G-4H** and the corresponding Results section (**lines 249-257**) in this fully revised manuscript.

Figure 4

G

H

- Validation of E1 interaction with SPSC3 and eIF3k needs to be stronger. Some concerns/questions are listed below. A myc tag was inserted between E3 and E2. How efficiently does furin cleave E3 from E2 in this virus and how are viral titers of

the myc-tagged virus compared to the non-tagged virus? I ask because is the IP looking at what is being pulled down by E2 or E3-myc-E2 that could be part of the spike polyprotein? The authors found E2 interacts with E3, E1 and a list of other host proteins. These results suggest several interactions including E2-host factor, E2-E1, E2-E3, E2-E1-host factor, E2-E3-E1, E2-E3-host factor. In figure 6d, and the subsequent conclusions, the authors suggest E1 is interacting with the host factor and do not see E2 alone and very low amounts of E3-E2-6K-E1. based on how the IP was performed I am not sure how an interaction between E1 and SPCS3 alone, without E2, would be detected. I would also like to see a reciprocal pull down using E1 and also E2 to see if these host factors are pulled down.

We thank the reviewer for these concerns. Given the low viral protein expression in macrophages (Figure 1A), we need an efficient system to enrich for large amounts of CHIKV glycoproteins for identifying host interactors through mass spectrometry. Adding tag/reporter proteins, such as mCherry, between E3 and E2 has been used to label alphavirus glycoproteins in previous study³, which is why we chose to use this myc tag labeling strategy coupled with myc Ab-conjugated agarose beads for AP-MS.

For Reviewer 1's first two questions, as speculated, inserting myc tag between E3 and E2 does attenuate CHIKV infectivity according to the reduced supernatant viral titers of 293T cells transfected with CHIKV/myc-E2 genomic RNA in comparison to those of cells transfected with unmodified CHIKV vaccine strain 181/clone 25 genomic RNA (shown below as parental CHIKV). Despite the attenuation, CHIKV/myc-E2 harvested from transfected 293T cells still reaches a titer over 10^8 pfu/ml, which allowed us to identify interactors by AP-MS. We also analyzed the cleavage efficiency of glycoproteins by comparing the expression levels of E3-E2-6K-E1, E3-E2 (p62), E2, and E3 in 293T cells transfected with unmodified CHIKV or CHIKV/myc-E2 genomic RNA. We didn't detect any uncleaved forms of the glycoproteins in cells transfected with either unmodified CHIKV or CHIKV/myc-E2 RNA when we probed with E2 antibody. However, probing with E3 antibody prior to longer exposure of the immunoblot showed higher E3-E2-6k-E1 and E3-E2 (p62) levels in cells transfected with CHIKV/myc-E2 RNA. Together, these results suggest that both mature E2 and E2-containing precursor polyproteins are more available to be pulled down in CHIKV/myc-E2 transfected cells probably due to lower cleavage efficiency. Overall, the expression levels of mature E2 detected by E2 antibody are similar.

We thank Reviewer 1 for providing a thorough dissection of all the possible interactions between the identified host factors and cleaved/uncleaved glycoproteins. As Reviewer 1 mentioned that E1 usually appears with E2 or E3-E2 in heterodimer forms, we were also surprised to find that E2 does not interact with either of the two host factors. To address this, we did both E2 and E1 reciprocal IP. For E2 reciprocal IP, we transfected 293T cells with plasmids expressing E3-myc-E2-6k-E1 and 3xflag-SPCS3/eIF3k and pulled down E2 with protein G Dynabeads conjugated with E2 antibody. For E1 reciprocal IP, we transfected 293T cells with plasmids expressing E3-myc-E2-6k-E1-3xflag and 3xflag-SPCS3/eIF3k and pulled down E1 with protein G Dynabeads conjugated with E1 antibody. In the second experiment, we were able to detect E1 and host factors using the flag antibody following the E1 IP. Again, we found that both SPCS3 and eIF3k consistently bind to E1 but not E2. We included these results in **Figure EV5C-5D** and Results section **(line 323-329)**.

Figure EV5

C Reciprocal E2 immunoprecipitation

D Reciprocal E1 immunoprecipitation

3. If CHIKV E1 is interacting with the host factors and that is antagonizing the antiviral response of SPSC3 (as one example), then what do pull downs using ONNV structural proteins look like? One would expect reduced interactions because the different amino acid causes a different E2-E1 dimer or attenuates the E1-host factor binding site.

We thank Reviewer 1 for this insightful suggestion. We agree that it would be informative to examine the interactions between ONNV glycoproteins and identified host factors (SPCS3 and eIF3k). Unfortunately, there is no commercial ONNV glycoprotein antibody available making this experiment unfeasible. Interestingly, we did observe reduced interactions between the host factors SPCS3 and eIF3k and the CHIKV E1-V220I mutant (V1029I in original manuscript version) where the positively selected site in E1 was mutated to the homologous ONNV residue (please refer to our response to Reviewer 3's major comment #1). This result suggests that the ONNV glycoproteins likely have an attenuated E1-host factor binding site as the reviewer speculated. We included this result as **Figure 7C**.

4. E1 and E2 are thought to interact during polyprotein translation and the initial dimer forms in the ER. If E1 is interacting with SPSC3 in the ER, is E2 also present? Or is a population of E1 not interacting with E2 in order to inhibit SPSC3? I would love a model of how the authors see all these factors coming together for this new role of E1.

We thank Reviewer 1 for proposing this interesting and novel hypothesis. The unexpected absence of E2 in both host factor and E1/E2 reciprocal pulldowns is contradictory to the prevailing model of E2 and E1 always acting together as heterodimers. It is possible that there is a group of free E1 proteins particularly interfering with host factors in the ER, which is a model worth further investigation and discussion. A great example of this is the alphavirus nonstructural protein 3 (nsP3) that plays essential roles in RNA replication, although depending on the alphavirus not all of the nsP3 in the cell colocalizes with dsRNA, supporting a separate distinct pool of nsP3 outside of active viral replication complex that interacts with host factors⁴.

To examine the free E1 model, we infected 293T cells that overexpressed 3xflag-SPCS3/eIF3k with CHIKV vaccine strain 181/clone 25 and analyzed the colocalization between viral glycoproteins and host factors by Airyscan microscopy. Through Airyscan, which has a higher resolution than traditional laser confocal microscope, we found that some of the E2 and E1 proteins colocalize but not all. The majority of E2 is found on the plasma membrane, while E1 mostly borders the nucleus, potentially localizing to the ER compartment. We selected four cells as regions of interest (ROI) and analyzed Pearson coefficient correlation of E1 vs

SPCS3/eIF3k and E1 vs E2 through Image J. The Pearson coefficient correlation value of E1 vs SPCS3/eIF3k is generally higher than that of E1 vs E2, suggesting that there is a separate distinct pool of E1 in the cytoplasm interacting with host restriction factors. We included these results in Figure 7A-7B and Results section (line 330-341).

Figure 7

Reviewer 1 minor comments:

Full Revision

1. In Figure 1c, (-) RNA is shown but in the rest of the figures (+) RNA is shown. Show both or select one. I do find it interesting the (-) RNA levels are similar over time, even at 4 hours post transfection (early time). Related to this, ONNV has higher levels of (-) RNA but what is known about structural protein levels in ONNV and CHIK in macrophages? Are there comparable levels of CP and GP being produced?

We thank Reviewer 1 for this comment. The (-) RNA is synthesized before the synthesis of subgenomic mRNA and therefore can reflect more accurately early viral replication and nonstructural protein functions. This is the reason why we consider the (-) RNA levels evaluated by specific nsP1 TaqMan probes to be more appropriate for determining early stage differences between ONNV and CHIKV replication in Figure 1 as that figure aims to define the steps in CHIKV life cycle that are more efficient than those of ONNV in THP-1 derived macrophages. On the other hand, the (+) RNA evaluated by E1 primers that we used in the later figures monitors viral RNA synthesis over time in the reflection of genomic (+) RNA and subgenomic mRNA transcribed from (-) RNA templates. Similar levels of (+) RNA and contrasting virion titers point the difference to the later stages of subgenomic mRNA translation, viral glycoprotein secretion, and assembly.

We have generated ONNV/myc-E2 reporter virus and assessed viral glycoprotein expression through flow cytometry using a FITC-conjugated anti-myc antibody in the THP-1 derived macrophages transfected with CHIKV/myc-E2 and ONNV/myc-E2 (shown below). Both of CHIKV and ONNV translation levels in macrophages seem to be suppressed. The results below show that the expression of ONNV glycoproteins is more inhibited than that of CHIKV glycoproteins. Since the anti-CHIKV activity of eIF3k is translation independent, it is likely that additional macrophage factors restrict CHIKV or ONNV expression. Since there is no commercial ONNV antibody available, we were unable to compare capsid expression levels between the two viruses. Overall, differences in the myc-tagged glycoprotein expression levels of the two viruses reveals ONNV defect in either structural protein translation or glycoprotein maturation.

2. Figure 2e and figure 3 have ONNV as the first bar followed by CHIK. In figure 1 and 2b, CHIK is first and then ONNV. helps the reader to have the controls in the same order.

We thank Reviewer 1 for this suggestion. We have changed the order of ONNV and CHIKV bars in **Figure 2E** and **Figure 3** so the CHIKV bar consistently comes first in all the figures.

3. Line 143-145 the authors discuss that when ONNV is the backbone and CHIK proteins are inserted the infection is more attenuated because of the E2 and E1 are from CHIK and ONNV, not the same virus (could also be E2-CP interactions are disrupted). However the chimeras made with the CHIK backbone (in Figure 2) have a mismatch between E2 and E1 as well.

We thank Reviewer 1 for this informative comment. We agree that the incompatible E2-E1 heterodimer formation may not be the only reason that causes attenuation of ONNV/CHIKV E1 and ONNV/CHIKV E2. There may be multiple factors contributing to the fitness of the chimeras, such as disruption of E2-CP interactions as the reviewer suggested, which requires more in-depth mechanistic investigations and is out of the scope of this study. We have now removed the explanation “potentially due to incompatible heterodimer formation between ONNV E2 and CHIKV E1” in **line 156-157**.

4. When discussing the residues that were found in the FEL and MEME analysis, the authors start the amino acid numbering from CP and continue along the polyprotein. Usually when discussing amino acids in the structural proteins, each protein starts at amino acid 1. So V460 would be E2-V135. It would also be useful to know what the residues in ONNV were at these positions to see if amino acids changed in charge, size, bond forming potential, etc. Showing these residues in the E2-E1 conformation found in the virion would also allow one to find adjacent residues that could explain differences in spike assembly and potentially where/how E1 is binding to a host protein.

We thank Reviewer 1 for this comment. We revised the amino acid numbers in the manuscript to start from the beginning of each structural protein. To look more into these residues in ONNV, we aligned CHIKV and ONNV from different lineages and compared the 6 positively selected sites (refer to our response to Reviewer 1's minor comment #5 and Figure EV2D-E). We found that E2-135 and E1-220, which are essential for CHIKV production, are valines in all the aligned CHIKV strains. For the aligned ONNV strains, E2-135 are all leucines and E1-220 are all isoleucines. While valine, leucine, and isoleucine are all amino acids with hydrophobic side chains, valine has the shortest side chain. The length of the side chains may lead to different hydrophobic properties that affect protein folding, which warrants further biochemical and structural analysis.

5. How effective is a non-attenuated CHIK strain in infecting macrophages? Could you make a SINV-La Reunion chimeric virus (which is BSL2) to see if a higher percentage of macrophages are infected and is this potentially contributing to the increased pathogenesis of La Reunion? Also how different is 181/25 with a pathogenic strain in the E2 and E1 residues? and compared to ONNV?

We thank Reviewer 1 for this question, which is also raised by Reviewer 2. In order to address this question, we compared the virulent CHIKV La Reunion and Asian strains to ONNV SG650 strain in their abilities to infect THP-1 derived macrophages. We found that the two pathogenic strains also infect macrophages better (Figure EV1A).

We thank Reviewer 1 for this insightful suggestion on investigating the conservation of these positively selected sites in different strains. We have aligned the sequences of ONNV and CHIKV strains from different lineages, including CHIKV vaccine strain 181/clone 25 and Thai strain AF15561 (the parental strain of CHIKV 181/clone 25). We found that the two positively selected sites with negative effects on virion production, E2-135 and E1-220 (sites 460 and 1029 in original manuscript version), are very conserved in either CHIKV or ONNV strains. CHIKV E2-135 is always valine (V) regardless of the

lineages, while ONNV E2-135 is always leucine (L). CHIKV E1-220 is always V, while ONNV E1-220 is always isoleucine (I).

Figure EV2

We also analyzed the amino acid heterogeneity of E2-135 and E1-220 in 397 CHIKV patient sequences from NCBI Virus database. Most of the amino acids at these 2 sites are V. The number of sequences with each amino acid at E2-135 and E1-220 is summarized in the table below. These results suggest that valine residues at E2-135 and E1-220 are crucial for CHIKV fitness and strongly selected during viral evolution. The sequence alignments and table are included as **Figure EV2C-2E** and discussed in the fully revised manuscript **(line 225-231, and line 244-248).**

	No. of sequences at E2-135	No. of sequences at E1-220
Valine (V)	394	392
Alanine (A)	1	3
Methionine (M)	1	0
Glutamic acid (E)	0	1
Glycine (G)	1	0
Isoleucine (I)	0	1

6. When describing the last results section, "CHIKV E1 binding proteins exhibit potent anti-CHIV activities" the authors use macrophages. In the rest of the text they consistently use THP-1 macrophages or human primary monocyte derived macrophages. The details

Full Revision

of the cell type are extremely useful to the reader and having those in the last results section would be great.

We thank Reviewer 1 for pointing out the importance of cell type clarification in the last Results section. We now consistently use “THP-1 derived macrophages” instead of “macrophages” in this section.

7. The paper is well-written. There is a slight disconnect as the authors go from discussing results in Figure 4 to Figure 5.

We thank Reviewer 1 for the comment regarding the disconnect between Figures 4 and 5 in this paper which is also shared by the other reviewers. We have taken the following approaches to address this comment:

1. We confirmed that mutations of the positively selected residues (E2-V135L, E1-V220I), which resulted in deficient virion production in Figure 4, do not affect CHIKV nonstructural or structural protein expression in THP-1 derived macrophages (**Figure 4F**).
2. We performed immunoprecipitation of the host restriction factors (SPCS3, eIF3k) identified in Figure 5 with CHIKV glycoprotein containing E1-V220I mutation, which caused abolishment of CHIKV production in Figure 4D. The result is presented in our response to Reviewer 3's major comment #1 and included in the manuscript as **Figure 7C**, showing that the E1-V220I mutation significantly attenuates E1 binding to SPCS3 and eIF3k. Given that both SPCS3 and eIF3k showed anti-CHIKV activity in Figure 6, this result suggests that SPCS3 and eIF3k likely drove the positive selection of E1 at this site.

Overall, we proposed that the positively selected sites found in Figure 4 have uncovered the molecular interface for the evolutionary arms race between CHIKV and the host restriction factors identified in Figure 5.

****Referees cross-commenting****

I agree with R#2 that having some Particle:PFU data would add some data to determine why such differences in titers/infectivity.

I also see how this m/s could be split into two different m/s. One that focuses on the chimeric viruses and another that identifies the host factors important and goes in more depth with mechanism.

We thank Reviewer 1 for all the great suggestions. We also debated whether we should present this as one bigger story or two stories. With additional experiments (now

included in new **Figures 7 and 8**, refer to our responses to the Editor's concern and Reviewer 2's optional comment #3), we feel that the virus and host sides of this evolutionary arms race story are significantly better connected and advance our understanding of a previously under-appreciated function of viral glycoproteins in modulating host innate immunity.

Reviewer #1 (Significance (Required)):

Strengths:

The authors have tackled an intriguing question: why do some alphaviruses infect macrophages and others do not. They have used a chimeric approach to very systematically identify the viral determinants E2 and E1 as being important in macrophage infection. Using AP-MS they identify host factors that interact with E2 (possibly E2 and E1, see comments above) but if their findings that E1 has a role in attenuating a host antiviral factor, this would be fantastic.

More and more examples of viral proteins having multiple roles during infection are in the literature. The idea that structural proteins also attenuate host antivirals is a developing field and vastly understudied. By fleshing out the results some more the authors might be onto something very important in alphavirus virology.

Limitations:

The study as it is presented is limited in the validation of host factors and their interacting partners. I have many questions about the methodology, validation, and model from this last section.

We thank Reviewer 1 for their enthusiasm about our study and providing many insightful comments to address the limitations. To address Reviewer and Editor's concerns, we used different methodologies to further validate host factor SPCS3 and eIF3k interactions with E1 in **Figure 7**. We also investigated eIF3k anti-CHIKV mechanism in eIF3k CRISPR KO cells in **Figure 8**. We uncovered that 1) E1 associating with the host restriction factors exists as a separate distinct pool from the E1 associating with E2, 2) positive selection of E1 drives stronger interaction with the host factors, and 3) the antiviral activity of eIF3k is specific to CHIKV and dependent on the HAM-domain of eIF3k.

Reviewer 2 summary:

The authors utilize: 1) chimeric arthritogenic alphaviruses; evolution selection analyses with virus sequences isolated from human patients; and 3) mass spectrometry and proteomics to interrogate determinants of chikungunya virus (CHIKV) permissiveness in

primary human macrophages and the human macrophage cell line, THP-1. The authors find that the vaccine strain, CHIKV 181/clone 25 replicates the most efficiently in primary monocyte-derived macrophages compared to other arthritogenic alphaviruses. Using o'nyong o'nyong (ONNV) as a comparison, the authors generate several chimeric viruses with CHIKV structural proteins and ONNV non-structural proteins (and vice versa) and perform a series of E1 and E2 domain swap experiments. They determine that both CHIKV structural proteins, E2 and E1, are necessary to confer efficient virus production over ONNV in the absence of a difference in viral RNA production. The authors also identify a specific residue in E1 that appears to be important for efficient virus production in THP-1 macrophage cell lines. Finally, using mass spectrometry, the authors identify two host proteins, SPCS3 and eIF3k, that bind to CHIKV E1 structural protein and appear to act as antiviral host factors.

Reviewer 2 major comments:

1. The authors elegantly demonstrate that CHIKV structural proteins confer an advantage over ONNV structural proteins in a step in the replication cycle downstream of virus RNA synthesis, possibly virion assembly. This point would be strengthened determining the particle-to-PFU ratio of the parental viruses and the chimeras. Presumably, the ratio would increase in the chimeras containing CHIKV structural proteins.

We thank Reviewer 2 for this excellent comment. We agree that determining particle-to-PFU ratios of parental and chimeric viruses will strengthen this study. To obtain the particle-to-PFU ratio, we infected THP-1 derived macrophages with CHIKV, ONNV and chimeras containing CHIKV glycoproteins (Chimera I from Figure 2A, and ONNV/CHIKV E2+E1 from Figure 3A) for 24 h. To quantify the secreted viral particles, we extracted viral RNA in the supernatant and detected (+) viral RNA through TaqMan qPCR assay with specific viral nsP1 probes. The released infectious virions were evaluated through plaque assay. The particle-to-PFU ratios are summarized in the table below. The results show that ONNV has the highest particle-to-PFU ratio (41398). On the other hand, the particle-to-PFU ratio of CHIKV (747) is significantly lower than that of ONNV by 55-fold. Replacing E3-E2-6K-E1 of ONNV with CHIKV poly-glycoprotein significantly reduces the particle-to-PFU ratio of ONNV by 8-fold to 4875. Replacing E2 and E1 of ONNV with the homologous proteins from CHIKV (ONNV/CHIKV E2+E1) significantly reduces the particle-to-pfu ratio by 20-fold to 2017. The particle-to-PFU ratio results imply that CHIKV might be more superior in the steps that requires glycoproteins, including viral assembly or exocytosis. This again emphasizes that CHIKV glycoproteins enhance the infectivity of viral progenies produced by THP-1 derived macrophages. We have included the results in **Figure 3D-3E** in our fully revised manuscript and described in **lines 165-171**.

Virus	Released particles (copy number)/ ml	Titer (pfu/ml)	Particle-to-PFU ratio (average)
ONNV	2.06E+10	4.50E+05	41398
	2.22E+10	6.00E+05	
CHIKV	1.27E+10	2.50E+07	747
	9.86E+09	1.00E+07	
Chimera I	2.28E+10	5.00E+06	4875
	1.82E+10	3.50E+06	
ONNV/CHIKV E2+E1	1.00E+11	4.30E+07	2017
	6.84E+10	4.00E+07	

2. Additionally, the authors should consider performing virion assembly blocking assays with a small molecule inhibitor to determine if this abrogates the virus production advantage of CHIKV structural proteins within the ONNV backbone.

We thank Reviewer 2 for this insightful comment. As the secretory pathway is commonly important for alphavirus glycoprotein maturation and assembly, it will be informative to interrogate CHIKV glycoprotein trafficking and assembly through this pathway using specific inhibitors, such as dihydropyridine FLI-06 and golgicide A. Golgicide A leads to rapid disassembly of the Golgi and trans-Golgi network (TGN)⁵. FLI-06 is a new inhibitor that interferes with cargo recruitment to ER-exit sites and disrupts Golgi without depolymerizing microtubules or interfering GBF1⁶. We pretreated THP-1 derived macrophages with 10 μ M FLI-06 or golgicide A for 30 mins prior to infection with ONNV, CHIKV, Chimera I, or ONNV/ CHIKV E2+E1. After 1 hour of virus adsorption in PBS with 1% FBS in the absence of the inhibitors, the cells were treated with the inhibitors at the same concentration (10 μ M) in complete medium for 24 h before viral supernatant was harvested for quantification through plaque assay. The plaque assay result shows that all the viruses are sensitive to secretory pathway inhibition, however, the production of viruses containing CHIKV glycoproteins (CHIKV, Chimera I, and ONNV/CHIKV E2+E1) is significantly more attenuated by FLI-06 and golgicide A treatment. This suggests that CHIKV glycoproteins-mediated trafficking and assembly is more heavily dependent on the host secretory pathway. We included this result in **Figure EV1B** and Results section **(line 171-180)**.

Figure EV1B

3. Finally, the authors should perform competition experiments with the chimeric viruses and ONNV in macrophages to determine if the chimeras can outcompete the parental ONNV strain. Based on their data, the chimeric viruses should outcompete.

We thank Reviewer 2 for this great suggestion. We agree that the competition experiment is an innovative and informative way to evaluate whether CHIKV glycoproteins confer a selective advantage on virion production in THP-1 derived macrophages. Unfortunately, there were several obstacles that prevented us from successfully performing this experiment.

Initially, we struggled with distinguishing between ONNV wild-type and chimeric virus. We thought about infecting THP-1 derived macrophages with ONNV and ONNV/CHIKV E2+E1 and detecting ONNV vs CHIKV E1 RNA through qPCR in the supernatant. However, this approach would lead to a false result, as the ONNV genome can be packaged in viral particles assembled by CHIKV E2 and E1, and the ONNV/CHIKV E2+E1 genome can be packaged in viral particles assembled by ONNV E2 and E1. Given

these concerns, we decided to evaluate the levels of the ONNV and ONNV/CHIKV E2+E1 glycoproteins in the supernatant from infected THP-1 derived macrophages. We were faced with two problems when attempting to detect viral proteins in the supernatant: 1) Inefficient concentration of ONNV and ONNV/CHIKV E2+E1 particles for immunoblotting; 2) Lack of commercially available antibodies for ONNV E2 and E1 protein detection. As a result, we needed to construct and use tagged viruses: ONNV/CHIKV myc-E2+E1 that has a myc tag at the N-terminus of CHIKV E2, and ONNV/HA-E2 that has a HA tag at the N-terminus of ONNV E2. We performed sucrose gradient ultracentrifugation of the supernatant viral particles but the myc and HA tags were not detected in the expected sucrose layer. Instead, we planned to pull down supernatant viral particles with myc-Ab and HA-Ab conjugated beads. Surprisingly, we found that the HA tag makes ONNV infection of macrophages much more efficient than the parental untagged strain. The change in the infection phenotype of tagged ONNV may lead to a false result of the competition experiment. The superinfection exclusion⁷ of alphavirus may also add difficulties to performing alphavirus coinfection in macrophages. Due to all these technical limitations, we have not been able to carry out the competition experiment, although we plan to in future studies.

4. The authors use both primary macrophages and macrophage cell lines as their in vitro model system and make one of their major points (listed in the title) that the determinants they identified in the CHIKV structural proteins convert macrophages into dissemination vessels; however, they do not show: 1) an in vivo model that the CHIKV-ONNV chimeras disseminate more efficiently than the parental ONNV; and 2) that these chimeras generate virus more efficiently specifically in macrophages. It would be useful to show that ONNV and CHIKV have equivalent virion production in other cell lines and that the advantage conferred by CHIKV structural proteins in the ONNV backbone is specific to macrophages. The authors should also change their title to reflect that dissemination is not directly being addressed in their study; the implications of their in vitro experimentation in a mammalian host would be more appropriate for the discussion.

We thank Reviewer 2 for this comment. For the first concern, we acknowledge the limitations of the study, which include a lack of direct demonstration of in vivo viral dissemination. To correct this, we added discussions on the limitations of our research and the potential of using the ONNV-CHIKV chimeric viruses for in vivo infection to address macrophage-mediated dissemination of CHIKV in the fully revised Discussion section **(Line 504-515)**.

For the second concern, we tested ONNV, CHIKV, Chimera I, Chimera III, and ONNV/CHIKV E2+E1 infections in 293T cells to investigate whether the advantage conferred by CHIKV glycoproteins are macrophage specific. As shown in Figure 2B and 3B, CHIKV, Chimera I, and ONNV/CHIKV E2+E1 produce similarly high titers in contrast

to ONNV and Chimera III. Here, we found that these viral infections are generally less restricted in 293T cells ($\sim 10^8$ pfu/ml), and there is no significant difference of Chimera I or Chimera III productions, compared to CHIKV. (The non significance is not annotated in the graph). Interestingly, the infection of 293T cells with ONNV/CHIKV E2+E1 is significantly more productive than that with the parental CHIKV. These results clearly demonstrate that ONNV infection is not as attenuated in 293T cells as in macrophages, and hence the requirement of CHIKV structural proteins is highly specific to macrophage infection. We included this result in **Figure EV3A** and added Result Section on this in **line 181-188**.

Figure EV3A

We have also updated the title to accurately reflect the significance of this research: **“Genetic conflict of alphavirus glycoproteins with macrophage factors drives virion production.”**

Reviewer 2 optional comments:

1. The authors use CHIKV-ONNV chimeras but it would be interesting to test other chimeras to determine if CHIKV structural proteins confer the same advantage in the backbone of other arthritogenic alphaviruses. The study would also be strengthened by using a pathogenic strain of CHIKV instead of the vaccine strain, as this is significantly attenuated in vivo.

We thank Reviewer 2 for this suggestion which is also suggested by Reviewer 1 in their minor comment #5. We recently got the approval for conducting BSL-3 work and carried out infection of THP-1 derived macrophages with virulent CHIKV La reunion strain (LR2006 OPY1) and Asian strain (AF15561). The result showed that both La Reunion and Asian strains infect THP-1 derived macrophages more efficiently than ONNV, suggesting CHIKV generally infects macrophages better. We included this in **Figure EV1A** and Results section **(line 107-110)**.

Figure EV1A

2. In Figure 4, the authors identify residues in the CHIKV structural proteins that appear to be under positive selection in human subjects and generate point mutants in these residues with the corresponding ONNV residues. They find that one mutation, V1029I located in E1, completely abolishes virion production in THP-1 macrophage cell lines. However, in their previous chimeric experiments, they find that neither CHIKV E1 or E2 was sufficient to increase virus production in the ONNV backbone. The authors should address this discrepancy, otherwise they should consider moving the data in their point mutation experiments to a supplementary figure. While worthy of reporting, especially

given the patient data, these experiments do not buttress the points made in the previous figures.

We thank Reviewer 2 for this insightful comment. According to previous studies, E2 and E1 always interact with each other from the step of single heterodimer formation in the ER to heterodimer trimerization before viral particle assembly. Although the E1-V220 site (previously called V1029) on the exterior of a single E-E1 heterodimer appears to not be engaged in the E2-E1 interaction, E1-V220 is partially exposed and protruding into the groove formed by E1 and the E2 of neighboring heterodimer after trimerization, accessible to host factors. As such, mutating CHIKV E1-V220 to the ONNV residue (E1-V220I) may not only disrupt E2-E1 trimerization but also interfere with viral glycoprotein interaction with host factors (presented in our response to Reviewer 1's major comment #1). Similarly, solely swapping E2 or E1 with CHIKV substitute in the ONNV backbone would also affect the interaction between neighboring E2 and E1 in trimerized spike, which may explain why neither ONNV/CHIKV E2 or ONNV/CHIKV E1 rescues virion production in THP-1 derived macrophages. We have included this in the fully revised Discussion section **lines 421-437**.

3. The authors conclude their manuscript with an assessment of several host proteins, namely SPCS3 and eIF3k, that were identified by mass spectrometry and whose knockdown results in increased virion production. The authors speculate about the role of these proteins but do not provide any mechanistic detail on how they might be playing a role. It is unclear that the putative antiviral role of these proteins involves steps downstream of virus replication, especially given that the authors speculate translation might be affected by eIF3k which, if the case, RNA synthesis should also be expected to be affected.

We thank Reviewer 2 for this comment. We acknowledge that we have yet a full mechanistic understanding of how SPCS3 and eIF3k impact virion production. This is also a concern from the Editor. In this fully revised manuscript, we generated eIF3k CRISPR knockout (KO) 293T cells and investigated the anti-CHIKV mechanism of eIF3k in the new **Figure 8**.

We successfully generated two eIF3k CRISPR KO single cell clones in 293T cells that are verified by genomic DNA sequencing and immunoblotting: clones 7 and 9 (**Figure 8A**). Unfortunately, we failed to obtain single cell clones in 293T cells with complete SPCS3 KO, consistent with a previous study by Rong Zhang et al⁸, where the authors were also unable to recover SPCS3 KO clones likely due to the importance of SPCS3 in cell survival. We infected one of the eIF3k KO clones (clone 9) with CHIKV vaccine strain

Full Revision

181/clone 25, ONNV strain SG650, and SINV strain Toto1101. Interestingly, we found that the antiviral activity of eIF3k is specific to CHIKV as CRISPR KO of eIF3k increases CHIKV production (by 2.5-fold) but not ONNV or SINV production (shown below). We have included this in the fully revised manuscript in **lines 364-372 (Figure 8A-8B)**.

We presume that Reviewer 2's inference of eIF3k's potential effects on viral RNA synthesis is based on our speculation of its antiviral role in viral translation, which may affect viral nonstructural gene expression. In Figure 6C, we found that eIF3k knockdown doesn't affect CHIKV RNA replication according to the qPCR result (gray bar). The results of **Figure 8C** and **8D** also revealed that the anti-CHIKV activity of eIF3k is translation-independent, as restoration of eIF3k in eIF3k KO cells does not inhibit viral nonstructural protein (nsP3) or subgenomic promoter-mediated E2, E1 and EGFP expression. Based on these results, eIF3k neither affects CHIKV RNA replication nor translation. We included these results in **Figure 8C-8D** and Result Section **line 376-384**.

We would like to clarify that eIF3k is not an initiation factor traditionally needed for cap-dependent translation. eIF3k is a nonessential component of eIF3 complex and can be easily dissociated from the complex. eIF3k is even absent from the genomes of some species. In addition to its cytoplasmic localization, eIF3k also strongly localizes to the nucleus suggesting there might be biological function of eIF3k beyond protein synthesis. We have included this additional discussion in the fully revised manuscript in **lines 492-498**.

In order to probe deeper into the antiviral mechanism of eIF3k, we further investigated the role of different eIF3k protein domains in CHIKV virion production. eIF3k consists of two domains, HEAT repeat-like HAM (HEAT analogous motif) domain, a winged-helix-like WH domain, and a C-terminal long tail region (**Figure 8E**). Additionally, the hydrophobic residues of the four helices from the HAM domain and the first helix from the WH domain form a hydrophobic core. We constructed pcDNA constructs with N-terminally 3xflag-tagged eIF3k mutants with the tail truncation (HAM+WH), partial WH and tail truncation (Core), and WH and tail truncation (HAM) (**Figure 8F**). We found that overexpression of both the HAM and WH domains, and the HAM domain alone inhibit CHIKV infection, suggesting that the HAM domain alone is sufficient to confer this antiviral phenotype (**Figure 8H**). We included these findings in the Results section (**line 385-399**). Taken together, these results indicate that the eIF3k HAM domain mediates specific and translation-independent inhibition of CHIKV virion production.

Figure 8

4. Overall, while the initial chimeric virus and domain swap approach is strong, the manuscript would benefit with a more thorough examination of virion assembly steps and a mechanistic link to virion production. Otherwise, the authors should revise the structure of their manuscript by de-emphasizing points about virion assembly and leave room for other mechanistic explanations of their chimeric data that more clearly link the host antiviral factor/E1 binding studies.

We thank the reviewer for these positive comments and constructive suggestions. We have addressed this by further interrogating the production kinetics of CHIKV, ONNV, and the chimeras containing CHIKV glycoproteins through determining their particle-to-PFU ratios as well as treating infected cells with secretory pathway inhibitors (refer to our responses to Reviewer 2 major comments #1 and #2). We have also demonstrated that eIF3k inhibits CHIKV virion production in a manner that is independent of changes in viral RNA synthesis and translation (**Figures 6C and 8**) and included additional discussion on eIF3k antiviral mechanism specifically on how it may affect other steps of the viral life cycle in the fully revised manuscript in **lines 478-503** (refer to our response to Reviewer 2 optional comment #3).

Reviewer 2 minor comments:

1. In Figure 3e, the line under "with CHIKV E1" should be moved over to include the E2-II+E1 virus.

We thank Reviewer 2 for pointing out this issue. We have corrected it (now **Figure 3G**) in our fully revised manuscript.

2. Figure 5a, 5b, and 6a should be replaced with higher resolution images.

We are sorry for the low resolution and small fonts of the images in Figure 5. In the fully revised manuscript, we adjusted the font size and resolution of Figure 5 to make it more readable. We also moved original Figure 5B and 5C to Figure EV3 and replaced them with a protein fold change histogram (**new Figure 5B**) and a CORUM protein-protein interaction network (**new Figure 5C**) to better present the proteomics results.

Reviewer 3 critique comments:

1. Overall, the manuscript is well written but in its current state it is more like two different stories because the effects of envelope proteins and list of interactors are not brought together in one story. A possible fix to this problem would be inclusion of ONNV and CHIKV containing env mutations that do and do not restore viral release from macrophages into the pulldown/association experiments shown in Figure 6.

We thank Reviewer 3 for the insightful suggestions to better connect the first (CHIKV determinants) and second (CHIKV glycoprotein interactors) parts of the manuscript. In response to the Reviewer's comment, we tested the binding of SPCS3 and eIF3k to

CHIKV E1 with the V220I (V1029I in original manuscript version) mutation (shown below), which we previously demonstrated to abrogate virion production in THP-1 derived macrophages in Figure 4D. We transfected plasmids expressing 3xflag-tagged SPCS3/eIF3k or empty vector for 24 h followed by transfection with plasmids expressing either the parental CHIKV vaccine strain 181/clone 25 poly-glycoproteins (E3-myc-E2-6K-E1) or poly-glycoproteins with the E1-V220I mutation. Interestingly, we found that mutating CHIKV E1-V220 to the homologous ONNV residue reduces the binding to both SPCS3 and eIF3k. This result strongly suggests that the positively selected E1-V220 is located in the interaction interface between E1 and SPCS3/eIF3k, which means that E1-V220I is evolutionarily selected by host restriction factors, including SPCS3 and eIF3k. We have included this result in the fully revised manuscript in **Figure 7C** and in **lines 342-349**.

Figure 7C

2. The other major issue is the lack of protein data for the viral mutants relative to WT ONNV and CHIKV and assessment of viral RNA in the supernatants to determine whether

the block is release or an earlier event since viral RNA levels in the cell seems to be the same or at least normalized.

We thank Reviewer 3 for the great suggestion on further characterizing the viral life cycle stages at which virion production of the CHIKV mutants is blocked. In our original manuscript, we showed that the E2-V135L mutation (V460L in original manuscript version) significantly decreases virus titers and intracellular (+) vRNA levels in viral RNA transfected macrophages while the E1-V220I mutation (V1029I in original manuscript version) completely abrogates virion production without affecting viral RNA replication (Figure 4D).

To address this question, we transfected viral RNA of the parental CHIKV vaccine strain 181/clone 25, E2-V135L mutant (V460L in original manuscript version), and E1-V220I mutant (V1029I in original manuscript version) into THP-1 derived macrophages. We then evaluated the intracellular protein expression and supernatant viral copy numbers of the E2-V135L and E1-V220I mutants to determine whether attenuated virion production is caused by a block in viral translation or release, respectively. It appears that both the E2-V135L and E1-V220I mutations moderately decrease E2 and E1 protein levels, respectively. However, both the E2 protein level of E2-V135L and E1 protein level of E1-V220I is robust in viral RNA-transfected macrophages, which implies that the defect in CHIKV mutant production from macrophages may be due to deficiency in downstream viral life cycle stages such as virion assembly or exit. We included this result in the fully revised manuscript as **Figure 4F** and result section **(line 241-244)**.

Figure 4F

The supernatant viral copy number quantified by TaqMan qPCR assay with specific probe against viral nsP1 showed that there is a slight decrease of mutant viral RNA E2-V135L and E1-V220I (see below), nevertheless, the decrease is not significant in comparison to parental CHIKV. Generally, E2-V135L and E1-V220I mutant transfection of macrophages still results in high viral copy numbers in the supernatant around 2×10^{10} . However, we are not completely convinced by the TaqMan assay result, as it's very difficult to remove the residual viral RNA in the supernatant that was not successfully transfected into the cells. Although we washed the macrophages extensively with PBS after transfection, we are concerned that the high copy numbers of E2-V135L and E1-V220I present in the supernatant may be due to residual untransfected RNA. Given that, we didn't include the TaqMan result in the fully revised manuscript.

3. Lastly, knockdown experiments indicate an effect of things like OAS3 or other innate immune modulators. There are no controls to demonstrate that these are specific to CHIKV infection or if knockdown would assist growth of ONNV as well.

We thank Reviewer 3 for the suggestion to check whether the identified host factors specifically target CHIKV or inhibit the infection of ONNV as well. We previously tried but were facing some issues. Since only a small fraction of macrophages can be infected with CHIKV and even a smaller fraction can be infected with ONNV (Figure 1A), it is hard to elucidate the roles of these identified host factors in ONNV infection by siRNA knockdown. We decided to take a more rigorous approach to investigate the antiviral specificity of identified host factors, especially the understudied SPCS3 and eIF3k, to different alphaviruses by generating complete knockout 293T single cell clones. Despite the fact that we did not successfully generate SPCS3 complete KO, we obtained an eIF3k KO single cell clone and infected it with CHIKV, ONNV and SINV (refer to our response to

Full Revision

Reviewer 2 optional comment #3). We found that eIF3k only has antiviral activity against CHIKV with almost no effects on ONNV or SINV infection. We have included this in our fully revised manuscript in **lines 364-372 (Figure 8A-8B)**.

Reviewer 3 minor comments:

1. The title does not fit the manuscript findings and should be modified.

We thank Reviewer 3 for this important comment, which was also brought up by Reviewer 2. We have now changed our title to “**Genetic conflict of alphavirus glycoproteins with macrophage factors drives virion production.**”, which more accurately reflects the significance of our research.

2. It is unclear why the authors show results for SINV and RRV in Figure 1. Either these should be removed or the viruses should be carried throughout the experiments described in the Figure. Better yet would be to add additional alphaviruses to this analysis to determine if there are additional viruses that act similarly to CHIKV.

We apologize for the confusion caused by including SINV and RRV results in Figure 1. We intended to show the superiority of CHIKV in infecting primary monocyte derived macrophages among arthritogenic alphaviruses, which we speculate may provide the molecular basis for macrophage-mediated CHIKV dissemination and disease. We decided to keep the SINV and RRV infection results in Figure 1 to highlight the relative susceptibility of primary monocyte derived macrophages to CHIKV. To echo the additional alphaviruses tested in Figure 1 and bring the story full circle, we included the results on SINV infection of eIF3k CRISPR KO 293T cells in **Figure 8B**. These results confirm the CHIKV-specific inhibitory activities of eIF3k.

3. Is the data presented in Figure 1A significant?

We thank Reviewer 3 for this question. To address this question, we plotted the values for EGFP-positive cells (percent infected cells) based on the experimental duplicates (shown below), and found CHIKV infection to be the most significantly different from that of the other alphaviruses in primary monocyte derived macrophages. The numbers above the bar charts are the mean values of EGFP+ cells .

4. The justification for inclusion of Figure 4A is lacking. It is unclear what this panel is supposed to be demonstrating.

We thank Reviewer 3 for this question. In Figure 4A, we wanted to show the bioinformatic analysis pipeline for detecting the positively selected residues in CHIKV glycoprotein sequences, which we further interrogated in Figure 4D. It's an important piece of information for Figure 4 and the follow-up investigations surrounding the positive selection sites of E2 and E1. We have included additional description and explanation of Figure 4A in our Results section in this fully revised manuscript (Line 211-218).

5. There is little justification for the candidates assessed in

We understand Reviewer 3's concern. Due to the nature of mass spectrometry studies which predict protein-protein interactions rather than direct functional validation, we acknowledge that we may miss some host candidates that have anti- or pro-CHIKV activities. Although justification of hit selection from mass spectrometry datasets is more difficult than that from CRISPR KO screen datasets, we set up specific criteria to identify host protein candidates with the greatest potential to functionally interact with CHIKV glycoproteins. Most of the proteins we chose to validate (Figure 6A) were identified in two independent AP-MS experiments and pass the P-value < 0.05 and the log2 fold change >0.

6. Extended data Figure 3 is very difficult to read due to the small font size.

We apologize for the small font in Extended data Figure 3. This problem was also mentioned by Reviewer 2. We removed original Extended data Figure 3 and included a new CORUM protein-protein interaction network in **Figure 5C**, which centers on the significant hits (colored in red) identified by two independent AP-MS experiments and includes relevant protein complex components detected in either independent experiment. The figure is now more concise and centralized, and the font is bigger.

7. Just to be clear, the blots shown in Figure 6D are different from those depicted in Extended data Figure 4b, because some of them look very similar.

We thank Reviewer 3 for this question. In **Figure 6D**, we expressed CHIKV glycoproteins through transfecting CHIKV genomic RNA into 293T cells, while, in **Figure EV5B** (original Extended data Figure 4b), we expressed CHIKV glycoproteins through transfecting a poly-glycoprotein plasmid (pcDNA3.1-E3-myc-E2-6K-E1) into 293T cells, which are complementary approaches to express CHIKV glycoproteins to validate their interactions with identified host factors. We have now added schematics to illustrate the different experimental strategies above the figures in this fully revised manuscript (shown below).

Figure 6D

Figure EV5B

1. Basore, K. *et al.* Cryo-EM Structure of Chikungunya Virus in Complex with the Mxra8 Receptor. *Cell* **177**, 1725-1737.e16 (2019).

2. Voss, J. E. *et al.* Glycoprotein organization of Chikungunya virus particles revealed by X-ray crystallography. *Nature* **468**, 709–712 (2010).
3. Jose, J., Tang, J., Taylor, A. B., Baker, T. S. & Kuhn, R. J. Fluorescent Protein-Tagged Sindbis Virus E2 Glycoprotein Allows Single Particle Analysis of Virus Budding from Live Cells. *Viruses* **7**, 6182–6199 (2015).
4. Götte, B., Liu, L. & McInerney, G. M. The Enigmatic Alphavirus Non-Structural Protein 3 (nsP3) Revealing Its Secrets at Last. *Viruses* **10**, 105 (2018).
5. Saenz, J. B. *et al.* Golgicide A reveals essential roles for GBF1 in Golgi assembly and function. *Nat Chem Biol* **5**, 157–165 (2009).
6. Krämer, A. *et al.* Small molecules intercept Notch signaling and the early secretory pathway. *Nat Chem Biol* **9**, 731–738 (2013).
7. Karpf, A. R., Lenches, E., Strauss, E. G., Strauss, J. H. & Brown, D. T. Superinfection exclusion of alphaviruses in three mosquito cell lines persistently infected with Sindbis virus. *J Virol* **71**, 7119–7123 (1997).
8. Zhang, R. *et al.* A CRISPR screen defines a signal peptide processing pathway required by flaviviruses. *Nature* **535**, 164–168 (2016).

Dear Melody,

Thank you again for the submission of your revised manuscript to The EMBO Journal. Two of the original referees who previously assessed the original version of your manuscript for Review Commons have been asked to re-assess the revised manuscript, and we have now received their comments, which I have already shared with you (included again below). They are both satisfied with the revision, they mention that the findings are now likely to be of interest to a general audience, and they support publication of the manuscript. There is only one minor request for clarification (by referee #1), which we would need from you to address in a final version of your manuscript.

From the editorial side, there are also some changes and corrections that we need from you before we can proceed with acceptance of the manuscript:

- The manuscript (including legends of main Figures, EV Figures and Tables following the list of References) needs to be uploaded in Word format (text highlighting can now be removed), while the Figures must be uploaded as individual, high-resolution Figure files. For more information about Figures, please visit: <https://bit.ly/EMBOPressFigurePreparationGuideline>.
- Please note that "equal contribution" is limited to co-first or co-last authors, it is not possible for the authors whose names appear in the middle positions of the author list. Instead, we use CRediT to specify the contributions of each author in the journal submission system. Please feel free to use the free text box to provide more detailed descriptions during submission. See also our guide to authors for more information: <https://www.embopress.org/page/journal/14602075/authorguide#authorshipguidelines>.
- All funding information should be entered in our manuscript handling system during resubmission, and it should be identical to that listed in your Acknowledgements section. The following item is currently missing from our online system: "Sydney Finegold 1110 Post-Doctoral Fellow Award; NIH GM089778".
- Please provide a list of up to 5 keywords after the Abstract of your revised manuscript.
- Please change the heading "Main" to "Introduction".
- Please change the heading "Methods" to "Materials and Methods".
- Please use our "Structured Methods" format to organize the information on reagents, tools and methods, part of which is now in the Appendix. Please find more instructions and templates in our guide: <https://www.embopress.org/page/journal/14602075/authorguide#textformat>.
- The literature citations in the References list should be provided in alphabetical order, and only the names of the first 10 co-authors should be listed followed by "et al." for publications with more than 10 co-authors. Please find more information on our reference format here: <https://www.embopress.org/page/journal/14602075/authorguide#referencesformat>.
- A conflict-of-interest statement (with the heading: "Disclosure and competing interests statement") is mandatory. Please review our policy here: (<https://www.embopress.org/page/journal/14602075/authorguide#conflictsofinterest>).
- Please include in your resubmission a completed author checklist, which you can download from our author guide (<https://www.embopress.org/page/journal/14602075/authorguide>). This checklist will also be part of the Peer Review File that will be published online along with your article.
- Your Appendix Table should be renamed to "Dataset EV1" with the appropriate callout in the manuscript and its legend included as a separate tab in the same Excel file.
- The Appendix must be provided as a single file in PDF format. The title of the study should be provided on its first page, followed by a Table of Contents including page numbers. The "Supplementary information" section should be removed from the main manuscript file.
- Please include in your resubmission the Source Data that have been requested by our Source Data coordinator, along with a completed Source Data checklist.
- Please note that EMBO press papers are accompanied online by:
 - A) a short (2 sentences) summary of the findings and their significance,
 - B) 2-5 short bullet points highlighting the key results, and
 - C) a synopsis image in .jpg or .png format that is exactly 550 pixels wide and 300-600 pixels high (the height is variable within this range). You can either show a model or key data in the synopsis image. Please note that the text needs to be legible at the

final size.

Please upload this information along with your revised manuscript (the text for A and B should be provided in a separate Word file).

- Please make sure that all deposited datasets (listed in your Data availability statement) are publicly available.
- Please define the annotated p values ***/** as well as provide the exact p-values for the same in the legend of figure 7d; 8b; as appropriate.
- Please note that the exact p values are not provided in the legends of figures 1b-c; 2b-c, e; 3b-c, e, g; 4d; 6b-c; EV 1a-b; EV 3a-c; EV 5a.
- Please indicate the statistical test used for data analysis in the legends of figures 4c; 5d; 6a; 7d; 8b; EV 2b; EV 4a.
- Please note that information related to n is missing in the legends of figures 7a-b, d; 8b; EV 3b-c; EV 5a.
- Please note that the error bars are not defined in the legends of figures 8b, h; EV 3b-c; EV 5a.
- Please change the section order of your manuscript as follows: title page with complete author information, abstract, keywords, introduction, results, discussion, materials & methods, data availability section, acknowledgements, disclosure and competing interests statement, references, main figure legends, tables, expanded figure legends.

Please also note that as part of the EMBO publications' Transparent Editorial Process, The EMBO Journal publishes online a Peer Review File along with each accepted manuscript. This File will be published in conjunction with your paper and will include the referee reports, your point-by-point response and all pertinent correspondence relating to the manuscript. You can opt out of this by letting the editorial office know (contact@embojournal.org). If you do opt out, the Peer Review File link will point to the following statement: "No Peer Review File is available with this article, as the authors have chosen not to make the review process public in this case."

We look forward to seeing a final version of your manuscript as soon as possible. Please use this link to submit your revision: <https://emboj.msubmit.net/cgi-bin/main.plex>

Best regards,

Ioannis

Referee #1:

This manuscript is a resubmission and the authors have very nicely nicely address the initial set of comments as well as added some additional experiments to strengthen the manuscript.

Minor issue: line 175 the authors say FLI-06 and Golgicide block the ER/Gogli and Golgi respectively. These drugs block specific pathways, and I would say what those are. Blocking the ER or Golgi doesn't make sense.

Referee #2:

The authors identify the Chikungunya virus (CHIKV) structural proteins E1 and E2 as factors that confer the ability of the virus to replicate in a macrophage cell line, THP-1. They find that this appears specific to macrophages as they do not observe the

advantage conferred by CHIKV structural proteins in 293T cells. The authors do a better job in this draft of connecting the two main findings: E2/E1 licensing of replication in macrophages and identification of interacting partners of E2/E1. The authors reconcile the discrepancy in not observing E2 in their pulldowns by suggesting that the binding partners interact with a pool of free E1 and show differential subcellular localization of E2 vs E1, which lends credence to the two proteins having a different suite of interactors. The authors further show mechanistic relevance of the E1 binding partners they identified which is also a significant improvement from the initial draft.

While not required for this publication, because eIF3k is a ubiquitously expressed protein, it would be interesting to determine what specifically about eIF3k in the context of macrophages is allowing it to act as a negative regulator of replication such that E1 has adapted to elude this putative antiviral activity. Understanding cell type-specific posttranslational modification of eIF3k would be useful.

The manuscript findings would be of interest to a general virology and cell biology audience. The findings lay groundwork for future studies aimed at understanding how arthritogenic alphaviruses (namely CHIKV) use macrophages as vehicles for dissemination and persistence in vivo.

No further comments. The authors have successfully addressed major concerns raised previously.

Rev_Com_number: RC-2023-02023
New_manu_number: EMBOJ-2023-115683R-Q
Corr_author: Li
Title: Genetic conflict of alphavirus glycoproteins with macrophage factors drives virion production.

All editorial and formatting issues were resolved by the authors.

Dear Melody,

Congratulations on an excellent manuscript! I am very pleased to inform you that it has been accepted for publication in The EMBO Journal. Many thanks for your thorough responses to the referee concerns and for addressing the editorial requests.

If you have any questions, please do not hesitate to contact the Editorial Office. Thank you for your contribution to The EMBO Journal. Working with you has been a pleasure!

Best wishes,

Ioannis

Rev_Com_number: RC-2023-02023
New_manu_number: EMBOJ-2023-115683R1
Corr_author: Li
Title: Genetic conflict of alphavirus glycoproteins with macrophage factors drives virion production.